# ALS-associated RNA-binding proteins promote *UNC13A* transcription through REST downregulation

Yasuaki Watanabe [1,2✉], Naoki Suzuki [1,3], Tadashi Nakagawa[2,4], Masaki Hosogane [2],
Tetsuya Akiyama[1], Naotoshi Kageyama[2], Yukino Funayama[1,2], Hitoshi Warita [1], Satoru Morimoto[5,6],
Hideyuki Okano[5,6], Masashi Aoki[1] & Keiko Nakayama [2,7✉]

## Abstract

**Amyotrophic lateral sclerosis (ALS) is a neurodegenerative disease characterized by selective loss of motor neurons. Although multiple pathophysiological mechanisms have been identified, no comprehensive understanding of these heterogeneous processes has been achieved. The ALS-associated RNA-binding protein (RBP) TDP-43 has previously been shown to stabilize *UNC13A* mRNA by preventing cryptic exon inclusion. Here, we show that the ALS-associated RBPs MATR3, FUS, and hnRNPA1 regulate *UNC13A* expression by targeting the transcriptional repressor REST. These RBPs bind to and downregulate *REST* mRNA to promote *UNC13A* transcription. Loss of any of these RBPs in cultured cells or in iPSC-derived motor neurons carrying the ALS-causing FUS P525L mutation leads to REST overexpression, and the same is observed in motor neurons of individuals with familial or sporadic ALS. The functional convergence of four RBPs on the regulation of *UNC13A* expression underscores the important role of this process for synaptic integrity, and its association with ALS pathogenesis could be relevant for the development of new therapeutic agents.**

**Keywords** ALS; REST; UNC13A; Cryptic Exon; FUS
**Subject Categories** Chromatin, Transcription & Genomics; Molecular Biology of Disease; Neuroscience

## Introduction

Amyotrophic lateral sclerosis (ALS) is a rapidly progressive neurodegenerative disease characterized by the degeneration and loss of motor neurons. The identification of several dozen genes that contribute to disease risk or act as causative agents has greatly advanced understanding of ALS pathophysiology (Brown and Al-Chalabi, 2017; Suzuki et al, 2023; Watanabe et al, 2020). These genes manifest a wide range of predicted functions, and multiple mechanistic hypotheses for their roles in ALS have been proposed, but a comprehensive understanding of these genetically diverse pathways remains elusive. Among the genes mutated in ALS, however, several—including those for TDP-43 (TAR DNA-binding protein–43), MATR3 (matrin 3), FUS (fused in sarcoma), and hnRNPA1 (heterogeneous nuclear ribonucleoprotein A1)—encode RNA-binding proteins (RBPs) that share an important role in RNA metabolism (Johnson et al, 2014; Kim et al, 2013; Van Deerlin et al, 2008; Vance et al, 2009; Xue et al, 2020).

Whereas nuclear loss and cytoplasmic accumulation of TDP-43 are the most prominent features of the pathology of ALS and frontotemporal dementia (Arai et al, 2006; Neumann et al, 2006), mutations in the COOH-terminal region of FUS result in its mislocalization to the cytoplasm and nuclear depletion, both of which are considered to be central to ALS pathogenesis (Akiyama et al, 2019; Kwiatkowski et al, 2009; Suzuki et al, 2010; Suzuki et al, 2012). Such mislocalization of FUS has been observed in sporadic as well as familial cases of ALS (Tyzack et al, 2019). Similarly, the disappearance of nuclear hnRNPA1 and MATR3 has been documented not only in familial ALS but in occasional sporadic cases (Honda et al, 2015; Johnson et al, 2014; Tada et al, 2018). These pathological features suggest that the nuclear function of various RBPs is important for the maintenance of motor neurons, and that the loss of such RBP function may lead to convergent impairment of RNA metabolism mediated by these proteins. However, it has remained unclear whether these RBPs regulate a common pathway or target the same RNAs.

In its role as a splicing regulator, TDP-43 suppresses the insertion of cryptic exons during pre-mRNA splicing (Ling et al, 2015). If TDP-43 is lost from the nucleus, these cryptic exons remain unspliced, often leading to instability and subsequent degradation of mRNAs via nonsense-mediated decay (NMD). The mRNAs for STMN2 (stathmin 2) and UNC13A (Unc-13 homolog A), both of which are key targets of TDP-43, are especially vulnerable to such instability (Brown et al, 2022; Klim et al, 2019; Ma et al, 2022; Melamed et al, 2019). STMN2 dysfunction is associated with both deficient axonal regeneration and the dying-

---

[1]Department of Neurology, Graduate School of Medicine, Tohoku University, Sendai, Miyagi 980-8575, Japan. [2]Division of Cell Proliferation, ART, Graduate School of Medicine, Tohoku University, Sendai, Miyagi 980-8575, Japan. [3]Department of Rehabilitation Medicine, Graduate School of Medicine, Tohoku University, Sendai, Miyagi 980-8575, Japan. [4]Department of Clinical Pharmacology, Faculty of Pharmaceutical Sciences, Sanyo-Onoda City University, Sanyo-Onoda 756-0884, Japan. [5]Keio University Regenerative Medicine Research Center (KRM), Kawasaki, Kanagawa 210-0821, Japan. [6]Division of Neurodegenerative Disease Research, Tokyo Metropolitan Institute for Geriatrics and Gerontology, Itabashi, Tokyo 173-0015, Japan. [7]Research Infrastructure Management Center, Institute of Science Tokyo, Bunkyo-ku, Tokyo 113-8510, Japan. ✉E-mail: yasuaki.watanabe.b8@tohoku.ac.jp; keiko.nakayama.e4@tohoku.ac.jp

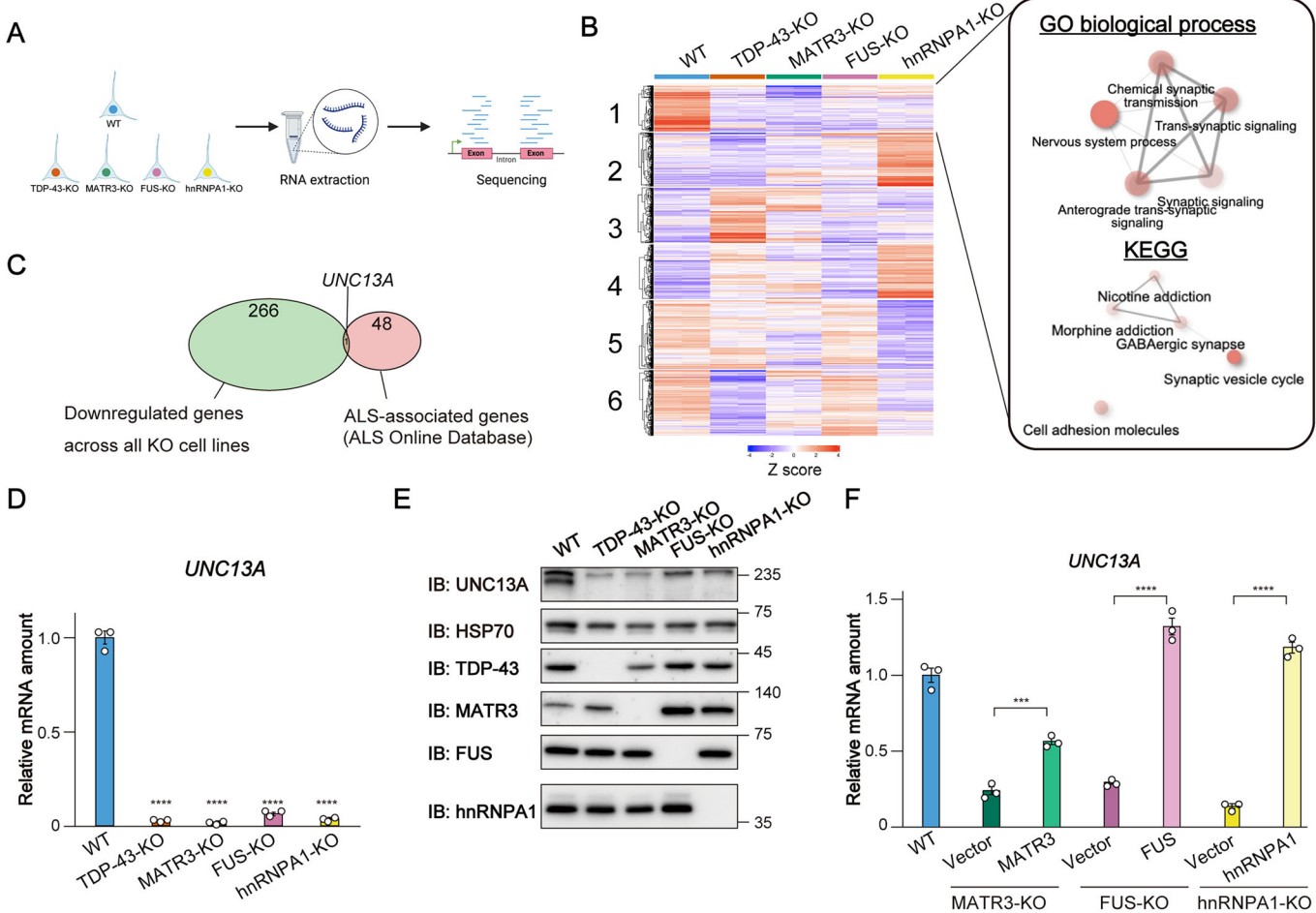

**Figure 1.** *UNC13A* expression is downregulated in RBP-KO cell lines.

(A) Experimental workflow for RNA-seq analysis of SH-SY5Y cell lines deficient in TDP-43, MATR3, FUS, or hnRNPA1. Illustrations were generated with Biorender.com. (B) The results of expression analysis for RNA-seq data from WT as well as TDP-43-, MATR3-, FUS-, and hnRNPA1-KO cell lines. The RNA-seq analysis was performed with biologically independent duplicates. The top 2000 most differentially expressed genes were classified into six groups by k-means clustering with the use of iDEP version 2.01 (http://bioinformatics.sdstate.edu/idep) (Ge et al, 2018), and the data were mean-centered for each gene (left). The heatmap color key represents Z-score normalized expression values. Genes in cluster 1 were subjected to Gene Ontology (GO) biological process and Kyoto Encyclopedia of Genes and Genomes (KEGG) pathway analysis, and the associated processes and pathways were visualized as a network (right). Color intensity indicates the false discovery rate (FDR) value, representing statistical significance, whereas circle size indicates fold enrichment. (C) Venn diagram showing the overlap between genes in cluster 1 (B) and ALS-related genes classified in the ALS Online Database (ALSod: http://alsod.iop.kcl.ac.uk) as definitive ALS genes, clinical modifiers, and genes with strong to moderate genetic evidence for ALS (Wroe et al, 2008), which are listed in Appendix Table S1. (D) RT-qPCR analysis of *UNC13A* mRNA in the four RBP-KO cell lines and WT cells. Data are means ± SEM from three biological replicates. ****$P < 0.0001$ (one-way ANOVA followed by Tukey's post hoc test); exact $P$ values: WT vs TDP-43-KO, $P = 2.1e{-}11$; WT vs MATR3-KO, $P = 1.9e{-}11$; WT vs FUS-KO, $P = 2.9e{-}11$; WT vs hnRNPA1-KO, $P = 2.3e{-}11$. (E) Immunoblot (IB) analysis of UNC13A, TDP-43, MATR3, FUS, and hnRNPA1 in the four RBP-KO cell lines and WT cells. HSP70 was examined as a loading control. (F) RT-qPCR analysis of *UNC13A* mRNA in WT cells and in three RBP-KO cell lines complemented with a corresponding doxycycline-inducible RBP vector (or the empty vector as a control) and treated with doxycycline. Data are means ± SEM from three biological replicates. ***$P < 0.001$, ****$P < 0.0001$ (Student's *t* test); exact $P$ values: MATR3-KO, $P = 0.00072$; FUS-KO, $P = 5.9e{-}05$; hnRNPA1-KO, $P = 1.2e{-}05$. See also Appendix Figs. S1, S2, Dataset EV1, and Appendix Table S1. Source data are available online for this figure.

back mechanism of motor neuron death in mice (López-Erauskin et al, 2024). UNC13A is essential for the formation and maintenance of synaptic vesicles and neurotransmitter release at synapses (Augustin et al, 1999; Willemse et al, 2023). Single-nucleotide polymorphisms in the noncoding region of *UNC13A* that are associated with increased ALS risk have been found to give rise to the inclusion of a cryptic exon that destabilizes its mRNA (Akiyama et al, 2022; van Es et al, 2009).

We have now uncovered convergent mechanisms by which four ALS-associated RBPs regulate the expression of *UNC13A*. Whereas TDP-43 stabilizes *UNC13A* mRNA by blocking insertion of a

cryptic exon, the loss of MATR3, FUS, or hnRNPA1 gives rise to the transcriptional repression of *UNC13A* mediated by repressor element-1 silencing transcription factor (REST). These findings thus identify *UNC13A* as a key convergence point downstream of these ALS-associated RBPs, suggesting that the dysfunction of these RBPs contributes to a unified pathogenic pathway characterized by loss of UNC13A expression. Our results represent a significant step forward in understanding the heterogeneous nature of ALS by elucidating how dysfunctions of diverse ALS-associated RBPs converge to impair gene expression control relevant to synaptic function.

# Results

## *UNC13A* expression is downregulated in RBP-KO cell lines

To investigate the possible operation of a common pathway triggered by the dysfunction of RBPs that are genetically and pathologically associated with ALS, we first generated SH-SY5Y human neuroblastoma cell lines deficient in either TDP-43, MATR3, FUS, or hnRNPA1 with the use of the CRISPR/Cas9 system (Appendix Fig. S1A–D). These RBP-knockout (KO) cells tended to exhibit a rounded morphology and lack cytoplasmic processes compared with WT cells (Fig. EV1A). Although TDP-43 plays a crucial role in neuronal survival (Iguchi et al, 2009; Sephton et al, 2010), we were still able to maintain TDP-43-KO cell cultures despite high expression of cleaved poly(ADP-ribose) polymerase (PARP), an apoptotic marker (Fig. EV1B). RNA-sequencing (seq) analysis of the wild-type (WT) and four RBP-KO cell lines revealed transcripts regulated by these RBPs (Fig. 1A). Application of k-means clustering analysis to the RNA-seq data identified a specific cluster of commonly downregulated genes in the RBP-KO cell lines that was prominently associated with neural processes and synaptic function (Fig. 1B; Dataset EV1), with other clusters being primarily related to nonneuronal processes (Appendix Fig. S2). The identification of synaptic signaling pathways as significantly enriched Gene Ontology (GO) terms is consistent with previous findings that synaptic dysfunction is an early feature of ALS (Nishimura and Arias, 2021; Vinsant et al, 2013). Further analysis of the cluster of commonly downregulated genes embedded within these neuronal pathways identified *UNC13A*—a gene implicated in the synaptic vesicle cycle—as the only gene overlapping with the 48 genes categorized in ALS online database (ALSoD) as having statistical support, clinical relevance, or strong to moderate genetic evidence for ALS pathogenesis (van Es et al, 2009; Wroe et al, 2008) (Fig. 1C; Dataset EV1; Appendix Table S1).

To validate our RNA-seq data, we performed reverse transcription and quantitative polymerase chain reaction (RT-qPCR) analysis. This analysis confirmed a substantial reduction in the amount of *UNC13A* mRNA in all four RBP-KO cell lines (Fig. 1D). In contrast, *STMN2* mRNA, another key target of TDP-43, was depleted only in TDP-43-KO cells (Appendix Fig. S1E). Immunoblot analysis revealed that the UNC13A protein was also depleted in MATR3-, FUS-, and hnRNPA1-KO cell lines as well as in TDP-43-KO cells, even though TDP-43 protein was present in these RBP-KO cell lines (Fig. 1E). The reintroduction of the respective RBP cDNA into MATR3-, FUS-, or hnRNPA1-KO cell lines restored *UNC13A* mRNA abundance (Fig. 1F; Appendix Fig. S1F–H), indicating that the loss of *UNC13A* mRNA was not due to off-target effects of each RBP gene knockout. Collectively, these findings suggested that ALS-associated RBPs commonly regulate synapse-related gene expression, including that of *UNC13A*.

## TDP-43 stabilizes *UNC13A* mRNA by blocking cryptic splicing, whereas MATR3, FUS, and hnRNPA1 promote *UNC13A* transcription

Nuclear depletion of TDP-43 has been associated with the downregulation of *UNC13A* expression attributable to cryptic exon inclusion and consequent mRNA instability (Brown et al, 2022; Ma et al, 2022). To investigate whether the loss of *UNC13A* mRNA observed in cells depleted of other ALS-associated RBPs was also dependent on cryptic exon inclusion, we performed RT-qPCR analysis to detect the cryptic exon included in mature *UNC13A* mRNA in response to TDP-43 depletion. With the use of primers designed specifically for the detection of this exon (Koike et al, 2023; Ma et al, 2022), we found it to be present in *UNC13A* transcripts of TDP-43-KO cells, but not in those of MATR3-, FUS-, or hnRNPA1-KO cells (Fig. 2A; Appendix Fig. S3A). This observation indicated that the downregulation of *UNC13A* expression apparent in these latter cell lines is not dependent on cryptic exon insertion.

To confirm that *UNC13A* mRNA containing the cryptic exon is degraded via the NMD pathway in our TDP-43-KO cell model, we examined the effect of NMD inhibition on *UNC13A* mRNA stability. RT-PCR analysis with primers designed to amplify regions including the cryptic exon revealed the presence of PCR products containing this exon in TDP-43-KO cells (Fig. 2B). The amount of these PCR products was increased by knockdown of UPF1 (Fig. 2B; Appendix Fig. S3B), a key component of the NMD pathway. Furthermore, treatment of TDP-43-KO cells with the NMD inhibitor cycloheximide restored *UNC13A* mRNA levels, as confirmed by RT-qPCR analysis targeting both cryptic and canonical exons (Fig. 2C,D; Appendix Fig. S3C). These findings confirmed that *UNC13A* mRNA is degraded as a result of the inclusion of a cryptic exon and subsequent NMD in the absence of TDP-43.

In contrast, inhibition of NMD in MATR3-, FUS-, or hnRNPA1-KO cells did not significantly affect *UNC13A* mRNA abundance (Fig. 2E; Appendix Fig. S3D). To compare nascent *UNC13A* transcript levels between the four RBP-KO cell lines and WT cells, we exposed the cells to 4-thiouridine (4-EU) to allow its incorporation into the newly synthesized transcriptome. Whereas 4-EU-labeled nascent *UNC13A* transcript levels were similar in TDP-43-KO cells and WT cells, they were significantly reduced in the other three RBP-KO cell lines (Fig. 2F,G). Further RT-qPCR analysis with primers targeting intronic regions of *UNC13A* did not show a reduction in the amount of *UNC13A* pre-mRNA in TDP-43-KO cells compared with WT cells, but a significant reduction was observed in MATR3-, FUS-, and hnRNPA1-KO cell lines (Appendix Fig. S3E,F). These findings indicated that, under normal physiological conditions, ALS-associated RBPs regulate *UNC13A* mRNA abundance through distinct mechanisms. Specifically, in the absence of TDP-43, *UNC13A* mRNA is destabilized as a result of the inclusion of a cryptic exon and degraded via the NMD pathway. In contrast, in the absence of MATR3, FUS, or hnRNPA1, *UNC13A* transcription is disrupted (Fig. 2H).

## REST is upregulated and binds to the *UNC13A* promoter in RBP-KO cells

The dysregulation of *UNC13A* transcription in MATR3-, FUS-, and hnRNPA1-KO cells did not likely reflect a direct effect of these proteins on gene transcription, given their established roles as RBPs (Xue et al, 2020). We therefore hypothesized that these RBPs influence a specific transcription factor that regulates *UNC13A* mRNA synthesis. To identify such a transcription factor, we performed a meta-analysis of chromatin immunoprecipitation (ChIP)-seq data sets from the ENCODE database (ENCODE

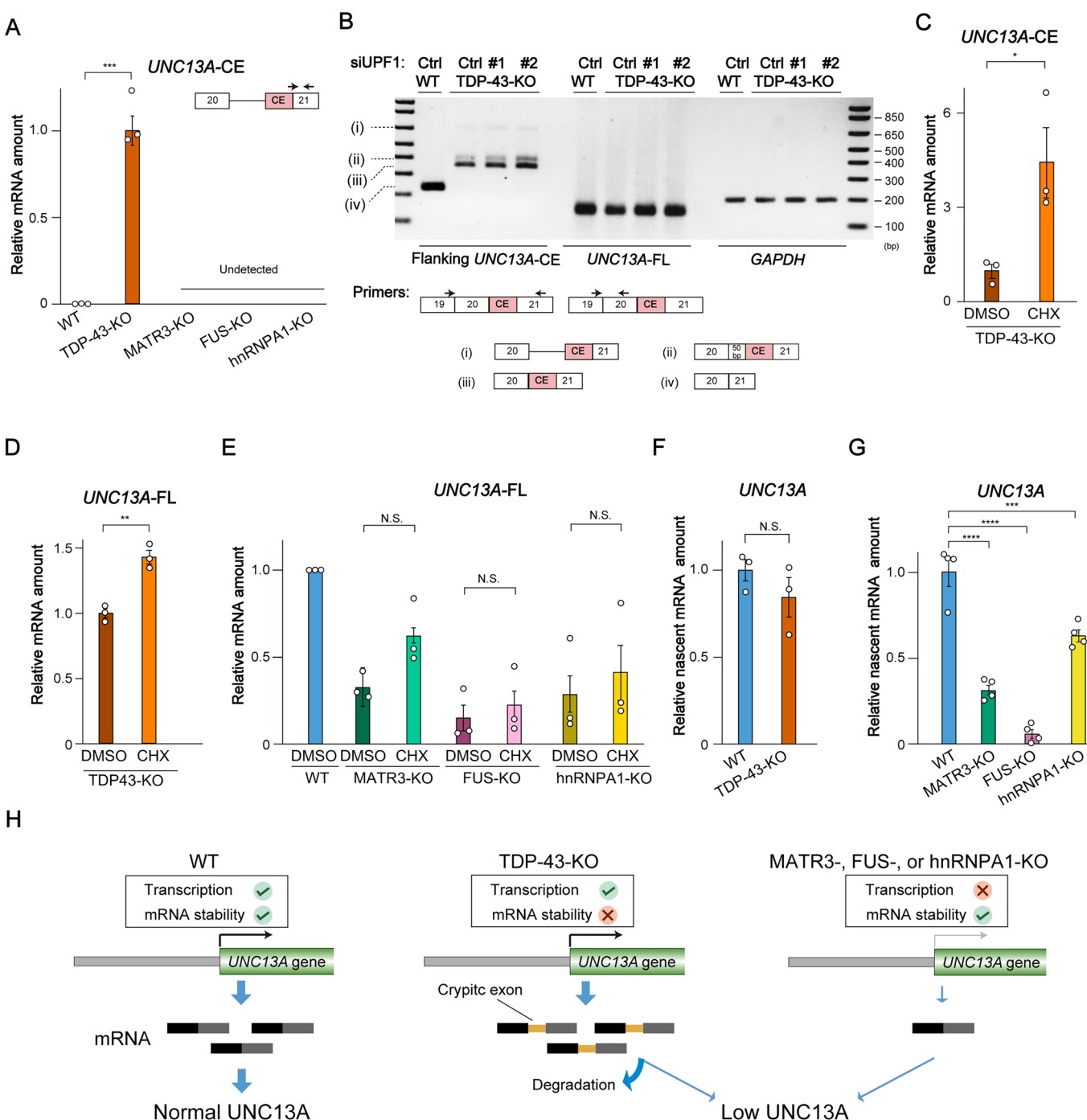

Project Consortium, 2012; Oki et al, 2018). Sixteen data sets showed significant peaks indicative of transcription factor binding at the *UNC13A* promoter region, with five proteins being enriched in this region (Fig. 3A; Appendix Table S2). Among these five proteins, REST (also known as neuron-restrictive silencer factor, NRSF) has been shown to suppress neuronal gene expression in nonneural tissues, a key function required for proper activation of these genes only in appropriate cells (Andrés et al, 1999; Lunyak and Rosenfeld, 2005). Disruption of this REST-mediated regulatory mechanism has been implicated in neurodegenerative disease (Hwang and Zukin, 2018). Suppressor of Integration 3A (SIN3A)

is a core component of the SIN3/HDAC complex and was one of the four other enriched proteins at the *UNC13A* promoter. Since SIN3A is known to be recruited by REST to silence neuronal genes (Huang et al, 1999), we decided to focus future analyses on REST. The ChIP-seq data revealed that REST binds to the *UNC13A* promoter in both neuroblastoma and nonneuronal cell lines, and we confirmed this binding pattern in SH-SY5Y cells by ChIP-qPCR analysis (Appendix Fig. S4A,B).

For subsequent analysis of RNA-seq data, we focused on genes with transcript levels downregulated as for *UNC13A* in MATR3-, FUS-, and hnRNPA1-KO cells in order to search for specific

◀ **Figure 2. TDP-43 stabilizes *UNC13A* mRNA by blocking cryptic splicing, whereas MATR3, FUS, and hnRNPA1 promote *UNC13A* transcription.**

(A) RT-qPCR analysis of *UNC13A* transcripts including the cryptic exon (CE) in WT cells and the four RBP-KO cell lines. The PCR primer sequences were referenced from Ma et al (2022), and their locations are illustrated in the upper right corner. The forward primer was designed to span the junction between the canonical exon and the cryptic exon. Data are means ± SEM from three biological replicates. ***$P < 0.001$ (Student's *t* test); exact *P* value = 0.00030. (B) RT-PCR analysis of WT or TDP-43-KO cells transfected with either a GC duplex (negative control, Ctrl) or two different small interfering RNAs (siRNAs #1 or #2) for *UPF1*. The PCR primer sequences were referenced from Ma et al (2022), and their locations are shown below the gel images; they flanked the CE of *UNC13A* or recognized a region of *UNC13A* mRNA unaffected by CE inclusion (FL). *GAPDH* was examined as an internal control. Among the PCR products amplified with the primers flanking the CE of *UNC13A*, (i–iii) indicate different intron retention patterns for products containing the CE, whereas (iv) indicates a product lacking the CE. (C–E) RT-qPCR analysis of *UNC13A*-CE mRNA in TDP-43-KO cells (C), *UNC13A*-FL mRNA in TDP-43-KO cells (D), and *UNC13A*-FL mRNA in WT, MATR3-KO, FUS-KO, and hnRNPA1-KO cells (E) after treatment with either cycloheximide (CHX, 100 μg/ml) or dimethyl sulfoxide (DMSO) vehicle for 6 h. Data are means ± SEM from three biological replicates. *$P < 0.05$, **$P < 0.01$; N.S., not significant (Student's *t* test); exact *P* values: (C) $P = 0.038$; (D) $P = 0.0022$. (F, G) RT-qPCR analysis of nascent *UNC13A*-FL mRNA in WT and TDP-43-KO cells (F) as well as in WT, MATR3-KO, FUS-KO, and hnRNPA1-KO cells (G) that had been labeled with 4-EU. Data are means ± SEM from three (F) or four (G) biological replicates. ***$P < 0.001$, ****$P < 0.0001$; N.S., not significant (F, Student's *t* test; G, one-way ANOVA followed by Tukey's post hoc test); exact *P* values for (G): WT vs MATR3-KO, $P = 1.8e{-}6$; WT vs FUS-KO, $P = 1.0e{-}7$; WT vs hnRNPA1-KO, $P = 0.00074$. (H) Proposed mechanisms for the regulation of *UNC13A* mRNA abundance in WT, TDP-43-KO, and the other three types of RBP-KO cells. See also Appendix Fig. S3. Source data are available online for this figure.

transcriptional regulators in these three cell lines. We identified common transcription factor binding motifs within 300 bp of the transcription start site (TSS) of such genes. These motifs included two complementary sequences targeted by REST (Fig. 3B; Appendix Table S3), further emphasizing the extensive regulatory influence of this transcription factor. Neuron-restrictive silencer element (NRSE) serves as a binding site for REST, recruiting corepressors to suppress transcription. While the canonical NRSE is highly conserved and functions as a strong REST binding site, noncanonical variants contain sequence differences but can interact with REST and contribute to transcriptional repression (Johnson et al, 2007). We noticed the presence of a noncanonical NRSE motif for REST binding in the *UNC13A* promoter region (Fig. 3C).

On the basis of these findings, we hypothesized that REST might be upregulated in RBP-KO cells compared with WT cells and thereby repress transcription of *UNC13A* and other target genes. Indeed, RNA-seq and RT-qPCR analyses confirmed that *REST* mRNA abundance was substantially higher in MATR3-, FUS-, and hnRNPA1-KO cells than in WT cells (Fig. 3D,E). In addition, the amount of REST protein was increased in these three RBP-KO cell lines but not in TDP-43-KO cells (Fig. 3F; Appendix Fig. S4C), suggesting that upregulation of REST might play a key role in the control of *UNC13A* and other gene expression in the absence of MATR3, FUS, or hnRNPA1.

We also investigated whether genes whose transcription is regulated by REST are included among the commonly downregulated genes in MATR3-, FUS-, and hnRNPA1-KO cells. We defined potential REST target genes as genes with a REST binding site within ±1 kb of the TSS by cross-referencing ENCODE data sets (ENCODE Project Consortium, 2012; Oki et al, 2018). We identified 38 genes that were both potential targets of REST and downregulated in the three RBP-KO cell lines (Appendix Fig. S4D; Table EV1). GO analysis of these genes identified several including *UNC13A* as being associated with synapse-related functions (Appendix Fig. S4E), suggesting that dysfunction of MATR3, FUS, and hnRNPA1 commonly induces synaptic pathology as a result of dysregulation of REST. In addition, 13 genes associated with the GO term "neuron projection" were identified (Appendix Fig. S4E). Indeed, the number of cells with cytoplasmic processes was lower in MATR3-, FUS-, and hnRNPA1-KO cells than in WT cells (Fig. EV1A), whereas knockdown of REST increased the proportion of these cells (Fig. EV2A,B). These findings suggest that REST regulates not only synapse-related genes but also a set of genes involved in neuronal morphology in the three RBP-KO cell lines.

## REST inhibits *UNC13A* transcription in RBP-KO cells

To assess the impact of REST upregulation on *UNC13A* promoter activity, we performed a luciferase reporter assay with three different constructs: a promoter-less luciferase (mock) construct as a control, a canonical construct containing the entire *UNC13A* promoter region, and a modified (ΔR) construct that contains a version of the *UNC13A* promoter lacking a 6-bp sequence that is required for REST binding and possesses a high PhyloP score indicative of a high level of evolutionary conservation (Fig. 4A,B). The promoter activity of the ΔR construct was significantly increased compared with that of the WT (canonical) construct in HEK293T cells, suggesting that endogenous REST represses *UNC13A* transcription (Fig. 4C). Furthermore, whereas knockdown of REST mediated by the CRISPR/Cas9 system increased the activity of the canonical construct, it had no significant effect on that of the ΔR construct (Fig. 4D,E). These data suggested that REST represses *UNC13A* transcription through its interaction with the gene promoter.

The activity of the canonical construct was significantly lower in MATR3-, FUS-, and hnRNPA1-KO cell lines compared with WT cells (Fig. 4F), consistent with the observation that *UNC13A* transcription is downregulated in these three RBP-KO cell lines (Fig. 2G; Appendix Fig. S3E). Conversely, the activity of the ΔR construct was increased in the three RBP-KO cell lines, with the ΔR/canonical activity ratio far exceeding that for WT cells (Fig. 4G), suggesting that the inability of REST to bind to the mutated *UNC13A* promoter results in a large increase in *UNC13A* transcriptional activity in MATR3-, FUS-, and hnRNPA1-KO cells, in which REST is overexpressed relative to WT cells.

To confirm that this regulation of *UNC13A* transcription by REST is reflected in the amount of *UNC13A* mRNA in WT and RBP-KO cell lines, we depleted *REST* mRNA in the cells by ~90% by siRNA transfection (Appendix Fig. S5A). This intervention resulted in a significant increase in *UNC13A* expression, which achieved similar levels in all tested cell lines (Fig. 4H). Together, these data suggested that the downregulation of *UNC13A* transcription apparent in MATR3-, FUS-, and hnRNPA1-KO cell lines is attributable to REST overexpression. In contrast, such REST knockdown had no significant effect on *UNC13A* expression in TDP-43-KO cells (Appendix Fig. S5B,C), consistent with our observation that *UNC13A* mRNA is destabilized and degraded as a result of cryptic exon insertion in these cells (Fig. 2A–D).

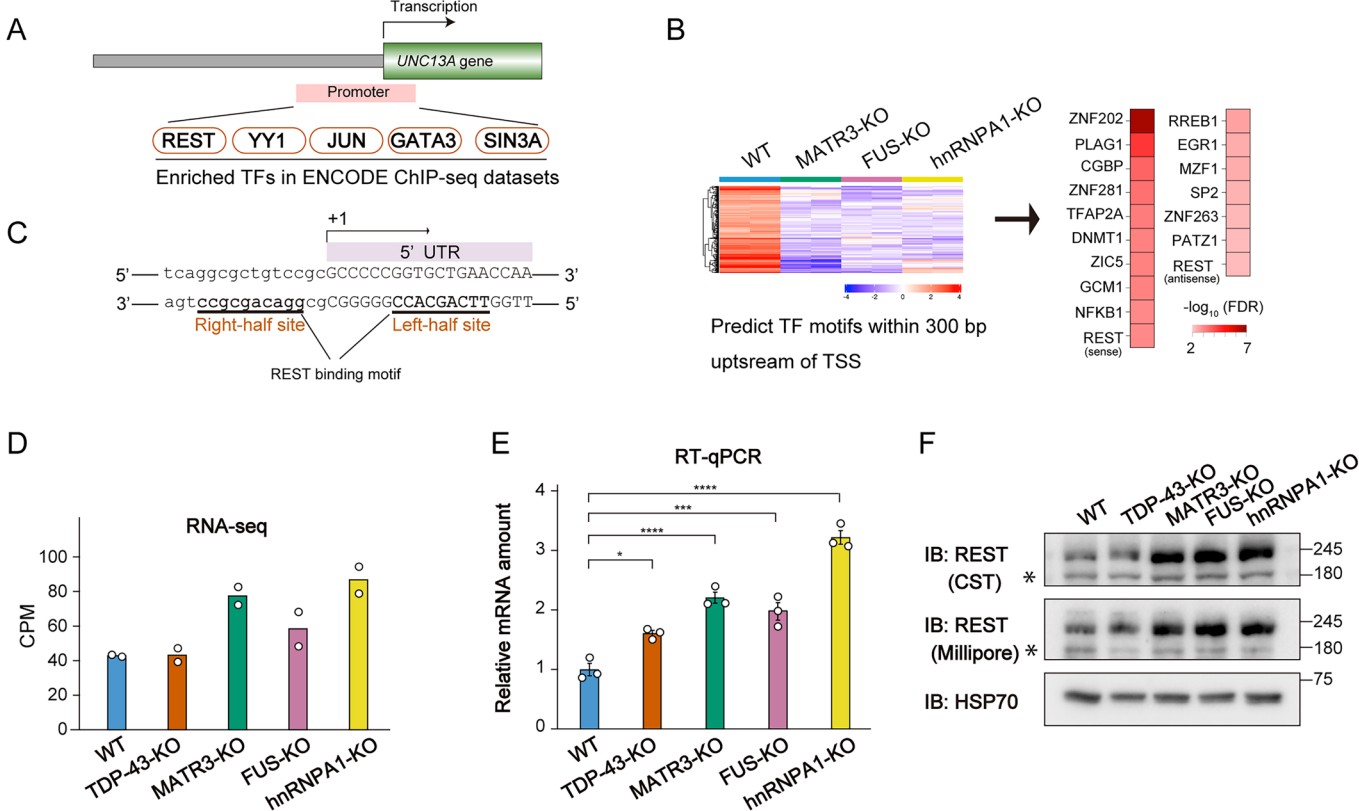

**Figure 3.   REST is upregulated and binds to the *UNC13A* promoter in RBP-KO cells.**

(A) Identification of transcription factors (TFs) that bind to the *UNC13A* promoter with the use of ChIP-Atlas (Oki et al, 2018). The *UNC13A* promoter region was aligned with ChIP-seq data sets from ENCODE that are specific to neuronal cells in order to identify enriched transcription factors. The enrichment score threshold was set at 500, and the analyzed promoter region was a 339-bp ENCODE candidate cis-regulatory element (cCRE) corresponding to chr19:17688234-17688572 in Hg38. The five identified transcriptional factors are shown. UTR, untranslated region. (B) Motif analysis for transcription factors enriched in the promoter regions (300 bp upstream of the TSS) of genes commonly downregulated in MATR3-, FUS-, and hnRNPA1-KO cell lines. The downregulated genes were identified by k-means clustering (left). Transcription factors with an FDR of <0.01 are shown in a heatmap based on the FDR values (right). (C) REST binding motif within the *UNC13A* promoter region. Bases corresponding to the noncanonical motif on the antisense strand are underlined. (D, E) *REST* mRNA abundance in four RBP-KO cell lines and WT cells as determined by RNA-seq (D) or RT-qPCR (E) analysis. The RNA-seq data are means from biological duplicates and are presented as counts per million (CPM), and the RT-qPCR data are means ± SEM from three biological replicates. *P < 0.05, ***P < 0.001, ****P < 0.0001 (one-way ANOVA followed by Tukey's post hoc test); exact P values for (E): WT vs TDP-43-KO, P = 0.017; WT vs MATR3-KO, P = 9.3e−05; WT vs FUS-KO, P = 0.00053; WT vs hnRNPA1-KO, P = 3.0e−07. (F) Immunoblot analysis of REST in four RBP-KO cell lines and in WT cells. The REST protein was detected at a position corresponding to ~210 kDa with two different antibody preparations. Asterisks indicate nonspecific bands. Quantitative data are presented in Appendix Fig. S4C. See also Appendix Fig. S4, Appendix Tables S2, S3, and Table EV1. Source data are available online for this figure.

## MATR3, FUS, and hnRNPA1, but not TDP-43, bind to *REST* mRNA

On the basis of our finding that *REST* mRNA abundance is increased in MATR3-, FUS-, and hnRNPA1-KO cell lines (Fig. 3D,E), we investigated the mechanism by which these three RBPs regulate *REST* expression. We found that the level of nascent *REST* mRNA was unchanged in the three RBP-KO cell lines compared with WT cells (Fig. 5A), suggesting that the increase in the amount of *REST* mRNA in the RBP-KO cells was not due to an effect on transcription. We next performed an RNA immunoprecipitation (RIP) assay to determine whether each of the three RBPs and TDP-43 interacts with *REST* mRNA. TDP-43 and MATR3 are known to interact with *TARDBP* mRNA, which encodes TDP-43 (Avendaño-Vázquez et al, 2012; Ayala et al, 2011; Boehringer et al, 2017). As expected, both proteins bound

to *TARDBP* mRNA, serving as a positive control in our RIP assay (Fig. 5B,C). Under these RIP conditions, TDP-43 did not bind to *REST* mRNA, whereas MATR3 did. Furthermore, FUS and hnRNPA1 were found to bind to *REST* mRNA (Fig. 5D,E), suggesting that the *REST* mRNA abundance was regulated posttranscriptionally by MATR3, FUS and hnRNPA1, but not TDP-43, through binding to and consequent destabilization of *REST* mRNA.

Although these RIP assays were optimized for protein-RNA interactions using the reducing agent dithiothreitol in the buffer, co-immunoprecipitation experiments without reducing agents revealed that FUS and hnRNPA1 interact with MATR3 as well as each other (Fig. EV3A,B), consistent with previous proteomic studies (Iradi et al, 2018; Yamaguchi and Takanashi, 2016). These findings suggest that MATR3, FUS, and hnRNPA1 may cooperatively regulate *REST* mRNA stability.

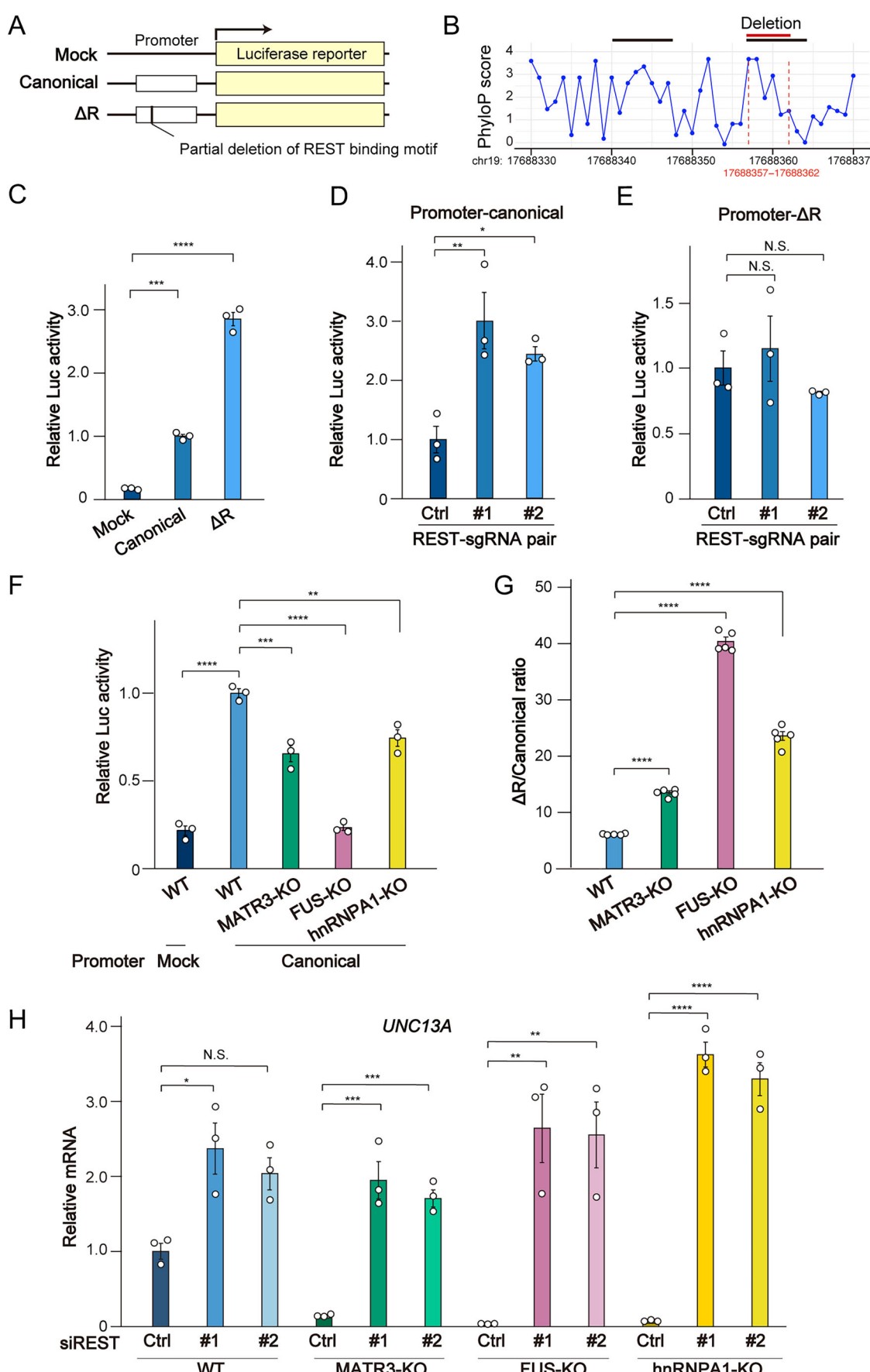

**Figure 4.   REST inhibits *UNC13A* transcription in RBP-KO cells.**

(A) Schematic diagram of mock, canonical, and ΔR luciferase reporter constructs for the *UNC13A* promoter. The ΔR construct lacks a 6-bp sequence essential for REST binding. (B) Line graph showing PhyloP scores for evolutionary conservation of the *UNC13A* promoter region. The black bars indicate the REST binding motif, and the red bar indicates the region deleted in the ΔR construct. (C) Relative luciferase (Luc) activity for HEK293T cells transfected with the firefly luciferase constructs shown in (A) as well as with a vector for Renilla luciferase. The firefly/Renilla luciferase activity ratio was measured 2 days after transfection. Data are means ± SEM from three biological replicates. ***$P < 0.001$, ****$P < 0.0001$ (one-way ANOVA followed by Tukey's post hoc test); exact $P$ values: Mock vs Canonical, $P = 0.00027$; Mock vs ΔR, $P = 4.5e{-}07$. (D, E) Relative luciferase activity for the canonical (D) and ΔR (E) promoter constructs in HEK293T cells previously transfected with two different pairs of Cas9-single-guide RNA (sgRNA) vectors targeting REST or with a control vector (Ctrl). Data are means ± SEM from three biological replicates. *$P < 0.05$, **$P < 0.01$; N.S., not significant (one-way ANOVA followed by Tukey's post hoc test); exact $P$ values for (D): Ctrl vs #1, $P = 0.0095$; Ctrl vs #2, $P = 0.039$. (F) Relative luciferase activity for the mock and canonical promoter constructs in WT and RBP-KO cell lines. Data are means ± SEM from three biological replicates. **$P < 0.01$, ***$P < 0.001$, ****$P < 0.0001$ (one-way ANOVA followed by Tukey's post hoc test); exact $P$ values: WT vs Mock, $P = 1.4e{-}07$; WT vs MATR3-KO, $P = 0.00025$; WT vs FUS-KO, $P = 1.7e{-}07$; WT vs hnRNPA1-KO, $P = 0.0025$. (G) The ratio of ΔR/canonical promoter construct luciferase activity in WT and RBP-KO cell lines. Statistical significance for the comparison of ΔR/canonical activity ratios between WT and each RBP-KO cell line was assessed. Data are means ± SEM from five biological replicates. ****$P < 0.0001$ (one-way ANOVA followed by Tukey's post hoc test); exact $P$ values: WT vs MATR3-KO, $P = 5.8e{-}07$; WT vs FUS-KO, $P = 2.3e{-}14$; WT vs hnRNPA1-KO, $P = 2.1e{-}12$. (H) RT-qPCR analysis of *UNC13A* mRNA in WT, MATR3-KO, FUS-KO, and hnRNPA1-KO cell lines transfected with a GC duplex (negative control) or two different siRNAs targeting *REST*. Data are means ± SEM from three biological replicates. *$P < 0.05$, **$P < 0.01$, ***$P < 0.001$, ****$P < 0.0001$; N.S., not significant (one-way ANOVA followed by Tukey's post hoc test); exact $P$ values: WT Ctrl vs #1, $P = 0.016$; WT Ctrl vs #2, $P = 0.051$; MATR3-KO Ctrl vs #1, $P = 0.00045$; MATR3-KO Ctrl vs #2, $P = 0.00099$; FUS-KO Ctrl vs #1, $P = 0.0053$; FUS-KO Ctrl vs #2, $P = 0.0063$; hnRNPA1-KO Ctrl vs #1, $P = 9.50e{-}06$; hnRNPA1-KO Ctrl vs #2, $P = 1.71e{-}05$. See also Appendix Fig. S5. Source data are available online for this figure.

## FUS IDRs are essential for the regulation of *REST* mRNA

Among the three RBPs that influence REST expression, we next focused on FUS in order to further investigate the molecular mechanism of *REST* mRNA regulation, given that depletion of FUS attenuated transcriptional regulation of *UNC13A* to a greater extent than did that of the other RBPs (Figs. 2G and 4F; Appendix Fig. S3E) and that *FUS* is more frequently mutated in individuals with ALS compared with the other RBP genes (Akiyama et al, 2016; Renton et al, 2014).

To identify which portions of the RNA-binding domain (amino acids 212–526) of FUS influence *REST* mRNA stability, we generated expression vectors for full-length (FL) FUS and deletion mutants lacking the glycine-rich, RRM, RGG1, RGG2, or ZnF domains (ΔGly, ΔRRM, ΔRGG1, ΔRGG2, and ΔZnF, respectively) (Fig. 6A) and introduced these constructs into FUS-KO cells (Fig. 6B). Although the expression level of FL in FUS-KO cells was relatively low compared with that of the endogenous protein in WT cells, its expression in FUS-KO cells effectively reduced the abundance of *REST* mRNA to a level similar to that apparent in WT cells (Fig. 6B,C). Whereas, like FL, the ΔRRM and ΔZnF mutants suppressed the amount of *REST* mRNA and rescued *UNC13A* expression in FUS-KO cells, the ΔGly, ΔRGG1, and ΔRGG2 mutants had no such effects (Fig. 6D,E). The domains deleted in the latter three mutants showed high PONDR scores, suggestive of the presence of intrinsically disordered regions (IDRs) (Fig. 6A). Given that IDRs contribute to phase separation (Molliex et al, 2015), the inability of these three mutants lacking IDRs to compensate for the loss of endogenous FUS implicated phase separation in the regulation of *REST* mRNA by FUS.

The 27 tyrosine residues in the $NH_2$-terminal prion-like domain (PrLD, amino acids 1–239) of FUS mediate phase separation through interaction with arginine residues in the IDRs of the RNA-binding domain (Qamar et al, 2018; Wang et al, 2018). To investigate the role of FUS phase separation in the regulation of *REST* mRNA, we substituted these 27 tyrosine residues with serine (27YS mutant) to impair such separation (Fig. 6F). To prevent loss of the 27YS mutant from the nucleus as a result of an inability to interact with chromatin (Reber et al, 2021), we also incorporated

the SV40 nuclear localization signal into the mutant protein (27YS-NLS mutant). Expression of the 27YS and 27YS-NLS mutants in FUS-KO cells revealed that, unlike the FL protein, they were predominantly found in the soluble fraction of cell lysates, consistent with previous findings that phase separation precedes the formation of the formation of insoluble aggregates (Appendix Fig. S6A) (Qamar et al, 2018; Reber et al, 2021). Furthermore, the 27YS and 27YS-NLS mutants showed reduced formation of cytoplasmic granules that colocalize with stress granules compared to the FL protein, suggesting impaired phase separation capacity (Fig. 6G; Appendix Fig. S6B). Neither mutant attenuated *REST* expression or restored *UNC13A* expression in the FUS-KO cells (Fig. 6H,I). These results thus underscored the essential role of FUS IDRs and potential contribution of phase separation in the regulation of *REST* mRNA stability by FUS.

Furthermore, although it remains unclear whether MATR3 and hnRNPA1 regulate *REST* mRNA through a mechanism similar to that of FUS, most ALS-associated mutations in these three RBPs are located within their IDRs (Beijer et al, 2021; Kapeli et al, 2017; Malik and Barmada, 2021). This observation raises the possibility that MATR3 and hnRNPA1 may also contribute to *REST* regulation through IDR-dependent mechanisms in the context of ALS (Figs. 6A and EV4A,B).

## REST is overexpressed in spinal motor neurons of individuals with ALS

Our data showed that *UNC13A* expression is repressed due to the overexpression of *REST* in SH-SY5Y cells depleted of MATR3, FUS, or hnRNPA1. To validate the relevance of this mechanism in *FUS*-mutated ALS pathophysiology, we examined induced motor neurons (iMNs) derived from induced pluripotent stem cells (iPSCs) carrying a mutation in the COOH-terminal NLS of FUS (FUS P525L/+), which is associated with juvenile-onset, severe ALS (Conte et al, 2012). This mutation has been shown to result in a loss of its nuclear function, likely due to cytoplasmic mislocalization of FUS (Marrone et al, 2019; Sun et al, 2015). Consistent with this, we observed partial redistribution of FUS to the cytoplasm in FUS P525L/+ iMNs (Appendix Fig. S7A). As expected, *REST* mRNA

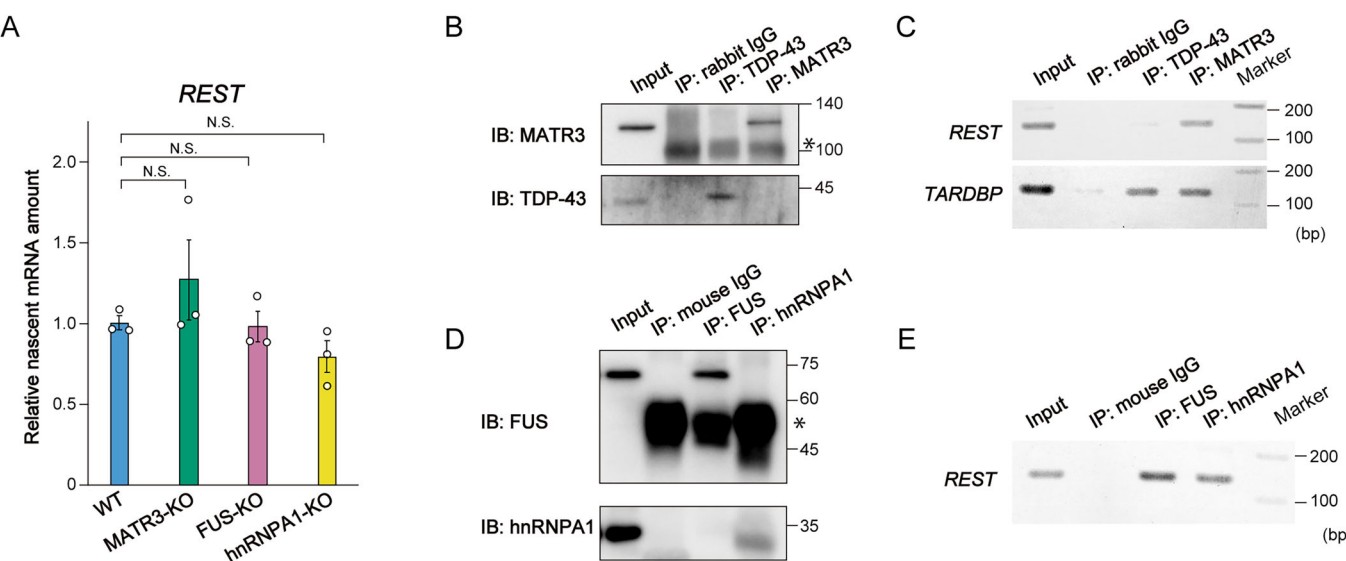

**Figure 5. MATR3, FUS, and hnRNPA1, but not TDP-43, bind to *REST* mRNA.**

(A) RT-qPCR analysis of nascent *REST* mRNA abundance in WT, MATR3-KO, FUS-KO, and hnRNPA1-KO cell lines. Data are means ± SEM from three biological replicates. N.S., not significant (one-way ANOVA followed by Tukey's post hoc test). (B) Enrichment of TDP-43 and MATR3 by immunoprecipitation from WT cell lysate. The cell lysate (Input) as well as the immunoprecipitate (IP) obtained with antibodies to TDP-43, MATR3 or with control immunoglobulin G (IgG) were subjected to immunoblot analysis with antibodies to TDP-43 or MATR3. An asterisk indicates the dimer of the IgG heavy chain. (C) Detection of *REST* mRNA by RT-PCR analysis of the samples obtained as in (B). (D) Enrichment of FUS and hnRNPA1 by immunoprecipitation from WT cell lysate. Asterisk indicates the monomer of the IgG heavy chain. (E) Detection of *REST* mRNA by RT-PCR analysis of the samples obtained as in (D). Source data are available online for this figure.

levels were elevated, whereas *UNC13A* mRNA levels were reduced in FUS P525L/+ iMNs relative to WT cells (Fig. 7A,B). Furthermore, protein expression analysis revealed similar changes in REST and UNC13A levels (Appendix Fig. S7B). These results indicate that the FUS-dependent regulation of REST and UNC13A observed in SH-SY5Y cells is recapitulated in a disease-relevant motor neuron model.

To extend our analysis to a broader pathophysiological context of ALS, including both familial and sporadic cases, we analyzed the transcriptomic data from Answer ALS platform, which provides RNA-seq profiles of iPSC-derived motor neurons from hundreds of ALS and non-ALS controls (Baxi et al, 2022). Correlation analysis revealed an inverse relationship between *REST* and *UNC13A* expression in both ALS and non-ALS groups, with no significant difference in the correlation coefficient between the two groups (Fisher's correlation test, $P = 0.197$) (Fig. EV5A,B). This indicates that REST-mediated repression of *UNC13A* is a conserved mechanism in motor neurons, regardless of ALS status. Nevertheless, contrary to our expectations, when comparing *REST* and *UNC13A* expression levels between ALS and non-ALS groups, no significant differences were observed (Fig. EV5C,D). We reasoned that this might be due to the limitations of bulk transcriptomic approaches in detecting subtle, cell-type-specific dysregulation of RBPs, including defects in their function or localization that are relevant to ALS pathology.

To assess the limitations of bulk analyses using iPSC-derived motor neurons, particularly their inability to fully capture disease-specific changes in mature motor neurons, we examined *REST* and *UNC13A* expression with publicly available RNA-seq data for motor neurons isolated from the lumbar region of individuals with sporadic ALS by laser capture microdissection. The symptoms of

these individuals were first apparent rostrally and progressed in a descending manner, indicating that the motor neurons in the lumbar anterior horn were relatively preserved (Krach et al, 2018). We found that *REST* expression was increased in the spinal motor neurons of these samples compared with those of control individuals without ALS (Fig. 7C).

The attenuation of *STMN2* expression, previously identified as a potential pathological indicator of TDP-43 mislocalization (to the cytoplasm rather than the nucleus) (Prudencio et al, 2020), was apparent in the ALS samples of the RNA-seq data set (Appendix Fig. S7C), suggestive of TDP-43 pathology. However, no correlation was detected between *STMN2* and *REST* expression levels (Appendix Fig. S7D), suggesting that *REST* overexpression was independent of TDP-43 pathology. The expression level of *UNC13A* tended to be decreased in the individuals with sporadic ALS, but this difference did not achieve statistical significance (Fig. 7D). Although the RNA-seq protocol specifically targeted motor neurons, the samples likely also included other cell types such as astrocytes and microglia (Krach et al, 2018). Given that *UNC13A* expression is minimal in glial cells (Uhlén et al, 2015), such contamination might have masked a potentially significant difference in *UNC13A* mRNA levels between non-ALS and ALS motor neurons.

We also examined REST protein expression in spinal motor neurons of individuals with ALS by immunohistochemical staining. In control individuals, FUS was localized predominantly to the nucleus (Fig. 7E). In contrast, in an individual with ALS associated with a mutation in the NLS of FUS (FUS-ALS), FUS was localized mostly to the cytoplasm, where it formed aggregates (Fig. 7F). Of note, the number of motor neurons expressing REST was significantly increased in FUS-ALS (Fig. 7G–I), suggestive of a

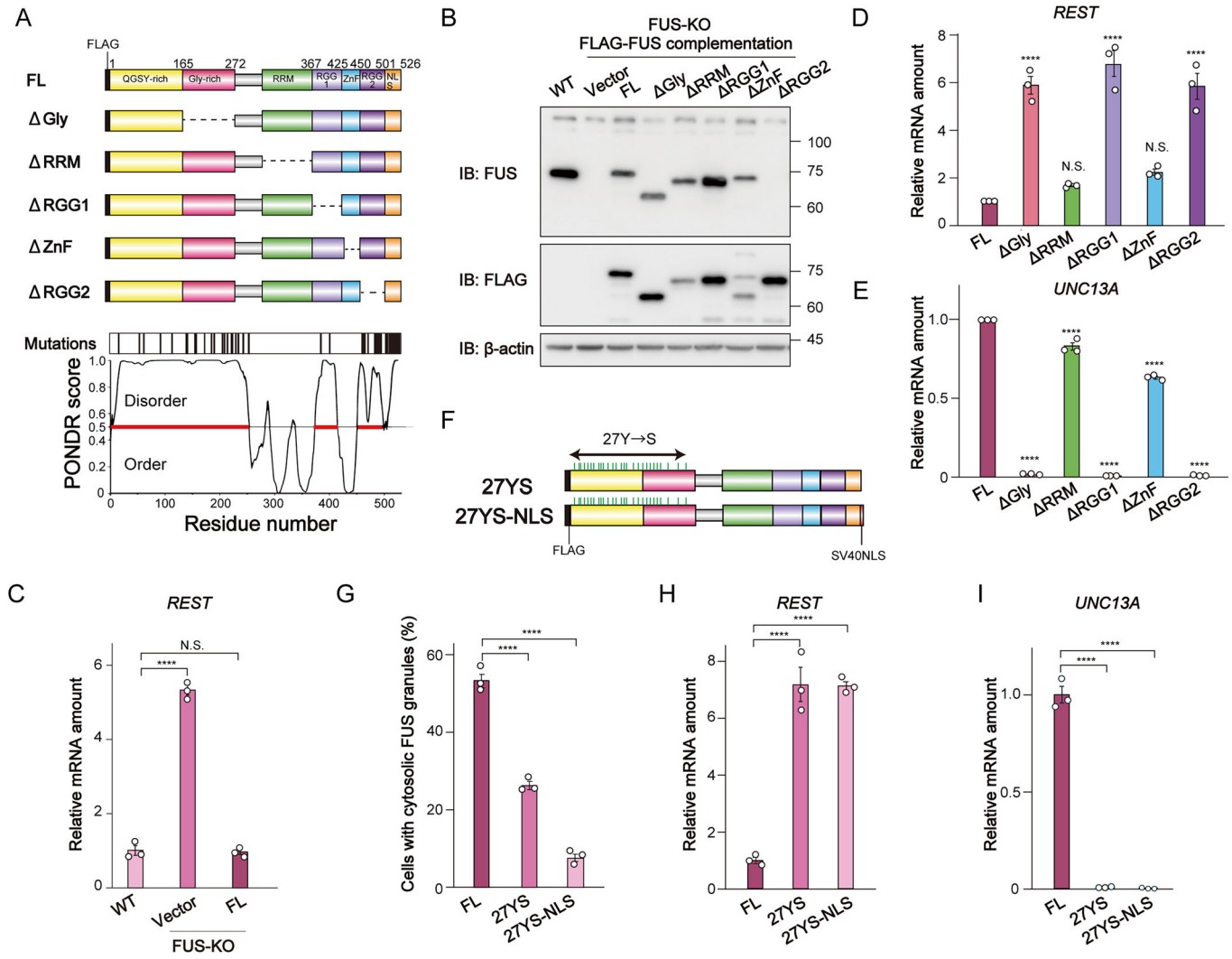

**Figure 6. FUS IDRs are essential for the regulation of *REST* mRNA.**

(A) Schematic representation of FLAG epitope-tagged human FUS mutant constructs (top). Dashed lines indicate deleted domains. FL full length, Gly glycine-rich domain, RRM RNA recognition motif, RGG Arg-Gly-Gly, ZnF zinc finger. The middle schematic illustrates the locations of ALS-associated mutations in FUS, referenced from Kapeli et al, 2017. Disorder prediction for FUS residues by PONDR (http://www.pondr.com) is shown at the bottom. (B) Immunoblot analysis of endogenous FUS in WT cells and of ectopic FLAG-tagged FL or deletion mutant forms of FUS expressed in FUS-KO cells. β-actin was examined as a loading control. (C) RT-qPCR analysis of *REST* mRNA in WT cells as well as in FUS-KO cells expressing FUS-FL or harboring the empty vector. Data are means ± SEM from three biological replicates. ****$P < 0.0001$; N.S., not significant (one-way ANOVA followed by Tukey's post hoc test); exact $P$ value: WT vs Vector, $P = 6.2e{-}07$. (D and E) RT-qPCR analysis of *REST* (D) and *UNC13A* (E) mRNAs in FUS-KO cells expressing FL or deletion mutant forms of FUS. Data are means ± SEM from three independent experiments. ****$P < 0.0001$; N.S., not significant (one-way ANOVA followed by Tukey's post hoc test); exact $P$ values for (D): FL vs ΔGly, $P = 6.6e{-}06$; FL vs ΔRGG1, $P = 1.1e{-}06$; FL vs ΔRGG2, $P = 7.3e{-}06$; exact $P$ values for (E): FL vs ΔGly, $P = 2.8e{-}14$; FL vs ΔRRM, $P = 1.5e{-}06$; FL vs ΔRGG1, $P = 2.8e{-}14$; FL vs ΔZnF, $P = 2.1e{-}10$; FL vs ΔRGG2, $P = 2.8e{-}14$. (F) Schematic representation of FUS mutants. The 27 NH$_2$-terminal tyrosines are all replaced by serine in 27YS, whereas 27YS-NLS also possesses the NLS of SV40 at its COOH-terminus. (G) Quantification of the percentage of cells with cytoplasmic FUS granules co-localized with G3BP after sodium arsenite treatment (1 mM, 30 min) in U2OS cells expressing FL, 27YS, or 27YS-NLS forms of FUS. At least 90 cells were analyzed per experiment. Data are mean ± SEM from three independent experiments. ****$P < 0.0001$ (one-way ANOVA followed by Tukey's post hoc test); exact $P$ values: FL vs 27YS, $P = 1.8e{-}05$; FL vs 27YS-NLS, $P = 1.0e{-}06$. See also Appendix Fig. S6B for representative images. (H, I) RT-qPCR analysis of *REST* (H) and *UNC13A* (I) mRNAs in FUS-KO cells expressing FL, 27YS, or 27YS-NLS forms of FUS. Data are means ± SEM from three biological replicates. ****$P < 0.0001$ (one-way ANOVA followed by Tukey's post hoc test); exact $P$ values for (H): FL vs 27YS, $P = 5.0e{-}05$; FL vs 27YS-NLS, $P = 5.3e{-}05$; exact $P$ values for (I): FL vs 27YS, $P = 5.5e{-}07$; FL vs 27YS-NLS, $P = 5.3e{-}07$. See also Appendix Fig. S6. Source data are available online for this figure.

pathological change linked to FUS dysfunction and involving REST overexpression. We also observed an increase in the number of REST-positive motor neurons in individuals with sporadic ALS compared with control individuals (Fig. 7I,J; Appendix Fig. S7E). Importantly, REST overexpression was observed even in an

individual with sporadic ALS lacking TDP-43 pathology (Fig. 7J; Appendix Fig. S7F), suggesting that REST overexpression is independent of such pathology. Together, these findings thus indicated that REST overexpression is a common pathological feature in both FUS-ALS and sporadic ALS, and they suggest a

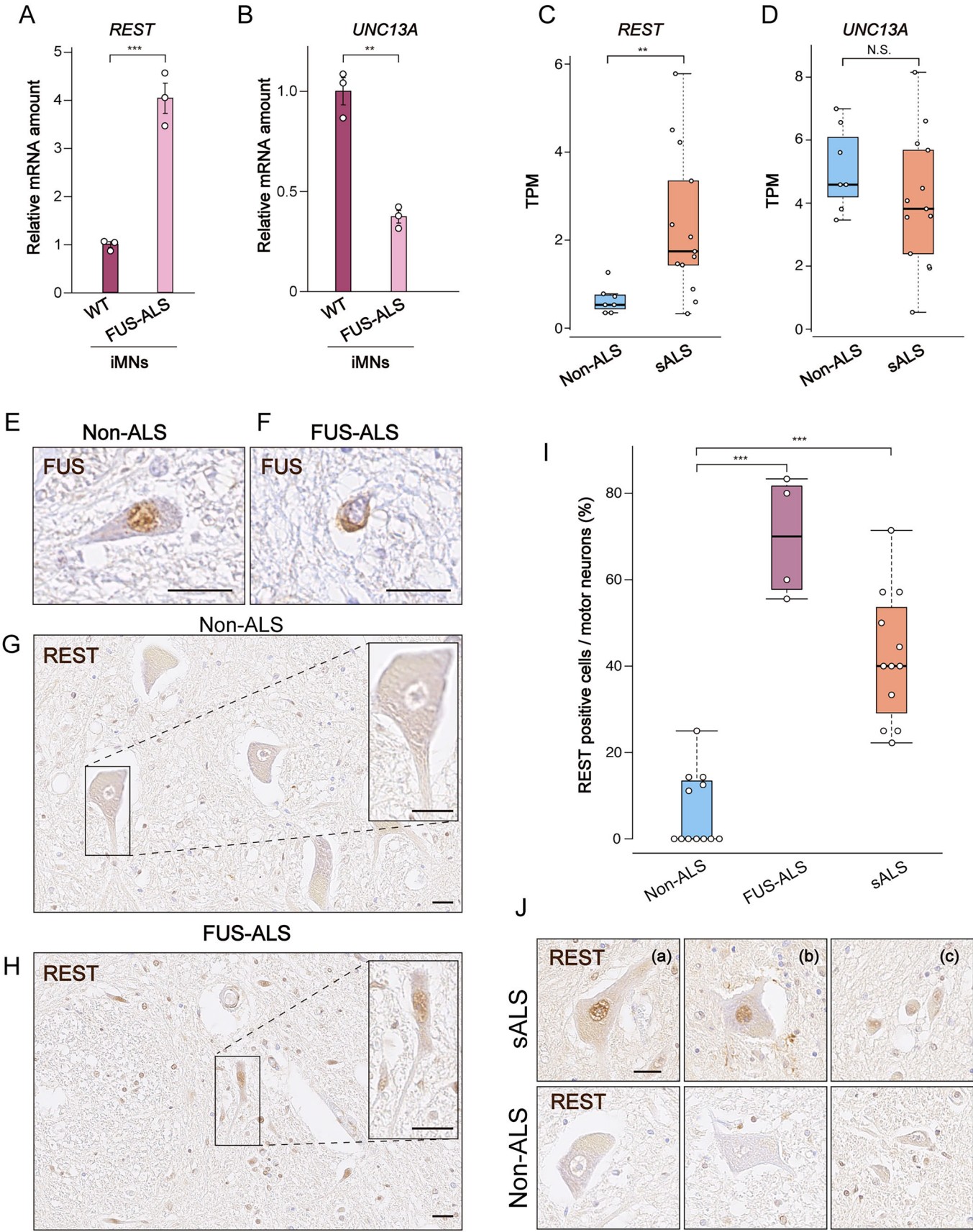

**Figure 7.   REST is overexpressed in spinal motor neurons of individuals with ALS.**

(A, B) RT-qPCR analysis of *REST* (A) and *UNC13A* (B) mRNA in WT and FUS P525L/+ (FUS-ALS) induced motor neurons (iMNs). Data are means ± SEM from three biological replicates. **$P < 0.01$, ***$P < 0.001$ (Student's *t* test); exact *P* values: *REST* (A), $P = 0.00071$; *UNC13A* (B), $P = 0.0011$. (C, D) RNA-seq analysis was previously performed for lumbar motor neurons isolated from control (non-ALS) individuals ($n = 7$) and patients with sporadic ALS (sALS, $n = 13$) by laser capture microdissection (Krach et al, 2018). The data are available under accession number GSE76220 in the GEO database. *REST* (C) and *UNC13A* (D) expression levels are shown separately for the non-ALS and sALS. TPM, transcripts per million. Data are presented as box plots, in which the boxes show the median and upper and lower quartile values, and the whiskers represent the range. **$P < 0.01$, N.S., not significant (Mann–Whitney *U* test); exact *p* value: *REST* (C), $P = 0.0070$; *UNC13A* (D), $P = 0.27$. (E, F) Immunohistochemical staining for FUS in spinal motor neurons of a control (non-ALS) individual with sporadic inclusion body myositis (sIBM) and an individual with familial ALS associated with a *FUS* mutation (R521C/+), respectively. Scale bars, 25 µm. (G, H) Immunohistochemical staining for REST in spinal motor neurons of a control individual with sIBM and an individual with familial ALS associated with a *FUS* mutation (R521C/+), respectively. The boxed regions in the main images are shown at higher magnification in the insets. Scale bars, 25 µm. (I) Quantification of REST-positive cells among anterior horn motor neurons by immunohistochemical analysis of spinal cord sections (four sections per individual) from three control individuals (with sIBM, carcinoma peritonitis, or multiple system atrophy; $n = 12$), one individual with familial ALS associated with a *FUS* mutation (R521C/+; $n = 4$), and three individuals with sALS ($n = 12$). The proportion of REST-positive cells was quantified in a 1-mm$^2$ area in each section and expressed as a percentage of anterior horn motor neurons. Box plots represent the median (center line), interquartile range (IQR; box), and whiskers indicating the most extreme data point within 1.5×IQR from the quartiles. **$P < 0.01$, ***$P < 0.001$ (Kruskal–Wallis test followed by Dunn's test); exact *P* value: Non-ALS vs FUS-ALS, $p = 0.00019$; Non-ALS vs sALS, $P = 0.00093$. See also Appendix Fig. S7D for the values for each individual. (J) Immunohistochemical staining for REST in spinal motor neurons of three individuals with sALS. TDP-43 pathology was apparent in patients (b) and (c), but not in patient (a), as is shown in Appendix Fig. S7D. Scale bar, 25 µm. See also Appendix Fig. S7. Source data are available online for this figure.

broad role for REST in ALS pathogenesis that is likely mediated by effects on the expression of synapse-related genes, including *UNC13A*.

## Discussion

Multiple RBPs are closely associated with ALS both genetically and pathologically. However, a unified downstream pathway by which these RBPs contribute to ALS pathogenesis has remained to be identified. We have now investigated *UNC13A* as a potential convergence point for mechanisms by which the loss of function of four ALS-associated RBPs gives rise to ALS. UNC13A plays a pivotal role in the packaging of neurotransmitters into synaptic vesicles and the subsequent transport of these vesicles to the presynaptic membrane (Betz et al, 2001). In addition, it facilitates the priming and docking of the vesicles, ensuring that they are properly positioned for rapid neurotransmitter release on neuronal activation (Augustin et al, 1999; Siksou et al, 2009). UNC13A therefore maintains effective synaptic transmission and overall neural function. The prevalence of intronic mutations in *UNC13A* that destabilize the mRNA in individuals with ALS suggests that a loss of the UNC13A protein can contribute to ALS pathogenesis. Furthermore, restoration of UNC13A expression in TDP-43–depleted neurons derived from induced pluripotent stem cells (iPSCs) was found to fully rescue impaired presynaptic function (Keuss et al, 2024), suggesting that the downregulation of UNC13A that results from TDP-43 loss underlies the disruption of synaptic integrity apparent in ALS. Our findings now extend this observation by showing that functional impairment of additional ALS-associated RBPs also results in suppression of *UNC13A* expression, suggesting that *UNC13A* is a convergence point for mechanisms underlying the disruption of presynaptic vesicle function. This pathophysiological convergence is consistent with evidence that synaptic dysfunction is a fundamental characteristic of ALS and is closely linked to the functional abnormalities of several ALS-related genes (Clayton et al, 2024; Nishimura and Arias, 2021). In the widely studied mouse model of ALS based on transgenic expression of the G93A mutant of superoxide dismutase 1 (SOD1), the fusion of synaptic vesicles at the neuromuscular junction has been found

to be impaired (Vinsant et al, 2013). Furthermore, synaptic loss has been detected in presymptomatic carriers of a mutation in *C9orf72*, which is the gene most strongly associated with familial ALS and frontotemporal dementia (Malpetti et al, 2021). Similarly, ultrastructural alterations of synaptic vesicles are evident at presynaptic terminals of the motor cortex in heterozygous *Fus* mutant mice (Scekic-Zahirovic et al, 2021). It is therefore plausible that synaptic terminal abnormalities can trigger pathogenesis in sporadic ALS.

Based on previous findings that *UNC13A* expression is reduced due to mRNA instability from noncoding region SNPs and CE inclusion following nuclear loss of TDP-43 (Brown et al, 2022; Ma et al, 2022), our study focused on *UNC13A* among REST target genes, proposing that the pathway from RBP dysfunction to *REST* upregulation and ultimately to *UNC13A* suppression plays a crucial role in ALS pathophysiology. Of note, we also identified multiple potential REST target genes other than *UNC13A* that were commonly downregulated in MATR3-, FUS-, and hnRNPA1-deficient cells. These findings raise the possibility that REST upregulation in ALS contributes to disease pathogenesis through mechanisms independent of UNC13A. REST is highly expressed in embryonic and neural stem cells, and it plays a key role in neuronal differentiation by repressing neuron-specific genes (Andrés et al, 1999; Johnson et al, 2008; Lunyak and Rosenfeld, 2005). Although it is expressed primarily in nonneuronal versus neuronal cells, REST has been implicated in neurodegenerative diseases. In Alzheimer's disease, the loss of REST from the nucleus of neurons is associated with a loss of its neuroprotective function and contributes to cognitive impairment (Lu et al, 2014). Conversely, in Huntington's disease, the mutated huntingtin protein induces abnormal REST accumulation in the nucleus and consequent suppression of the expression of genes such as that for brain-derived neurotrophic factor (BDNF) that are required for neuronal survival, thereby contributing to neurodegeneration (Soldati et al, 2013; Zuccato et al, 2007; Zuccato et al, 2003). Although the possibility of a direct link between REST overexpression and ALS has not been well explored, studies have suggested that such a link may exist. In a mouse model of spinal cord injury, for example, REST disrupts axon regeneration by repressing regeneration-associated genes (Cheng et al, 2022). Furthermore, neurite regrowth following axotomy was found to be impaired in motor neurons

derived from iPSCs harboring an ALS-associated *FUS* mutation (Stoklund Dittlau et al, 2021), suggesting the possibility that FUS dysfunction might lead to the overexpression or activation of REST, which in turn might be responsible for the inhibition of neuronal regeneration. In addition, REST overexpression in mice gives rise to impairment of spontaneous locomotion (Lu et al, 2018), a phenotype also seen in the SOD1(G93A) mouse model of ALS (Allodi et al, 2021), supporting the hypothesis that REST plays a role in ALS pathogenesis.

In this study, we primarily employed SH-SY5Y cell models with individual RBPs depleted, owing to their tractability and reproducibility, which are essential for systematic functional comparisons across multiple RBPs. Although iPSC-derived motor neurons offer certain physiological advantages, they are less suitable for our primary focus on the RBPs–REST–UNC13A regulatory axis, due to technical variability and the considerable challenges associated with generating multiple cell lines with individual RBP gene knockout and differentiating them into homogeneous, mature neuronal populations. Our models did not fully replicate the physiological properties of mature motor neurons, but they provided functional evidence supporting a pathological role of REST in ALS. Specifically, we observed that REST knockdown partially rescued the shortened cytoplasmic processes in SH-SY5Y cells depleted of MATR3, FUS, or hnRNPA1. These results support the idea that REST overexpression contributes to neurodegenerative phenotypes in models of ALS-related RBP loss of function. It should also be noted that REST may have a neuroprotective role in certain contexts, such as aging-related neurodegeneration and Parkinson's disease (Lu et al, 2014; Ryan et al, 2021). In our clinicopathological analysis, the postmortem spinal cords of ALS patients exhibited a greater number of motor neurons expressing REST compared to non-ALS cases. While this observation may indicate that elevated REST expression promotes neurodegeneration in ALS, it also raises the alternative possibility that neurons retaining REST expression are relatively more likely to survive. This suggests that REST activation may function as a compensatory response rather than serving as a primary pathogenic driver—a possibility that remains to be fully elucidated. Although REST may indeed serve context-dependent roles, our data support the notion that its over-expression contributes, at least in part, to ALS pathogenesis by suppressing neuronal gene expression. Clarifying whether REST predominantly exerts a protective or pathogenic role in ALS will require models such as iPSC-derived motor neurons and in vivo systems.

We have now shown that MATR3, FUS, and hnRNPA1 bind to *REST* mRNA and thereby attenuate *REST* expression at the posttranscriptional level. This finding is consistent with previous RNA-seq data showing that *REST* mRNA stability increased in response to FUS depletion in human neural progenitor cells in which RNA synthesis had been inhibited (Kapeli et al, 2016). Of note, we further show that the 27YS mutant of FUS is deficient in liquid-liquid phase separation (LLPS) and unable to downregulate *REST* mRNA in FUS-KO cells, underscoring the potential importance of LLPS in this regulatory mechanism. While FUS is well known for its role in stress granule formation, it also contributes to the assembly of nuclear LLPS-dependent condensates, such as paraspeckles, which regulate mRNA fate by sequestering particular transcripts (An et al, 2019; Reber et al, 2021). Although we did not identify specific condensates regulating

*REST* mRNA, the dysregulation in both FUS-depleted and 27YS-expressing cells under non-stress conditions suggests that FUS-dependent nuclear condensates formed under physiological conditions are involved in *REST* mRNA regulation. In line with this, both FUS and hnRNPA1 harbor large intrinsically disordered PrLDs that facilitate LLPS, with many ALS-associated mutations having been found to concentrate in these regions (Milicevic et al, 2022). Although MATR3 lacks a distinct PrLD, its $NH_2$-terminal region contains a disordered domain that mediates LLPS, and the S85C mutation of MATR3, which is strongly linked to ALS onset, is located in this region and influences the condensation process (Johnson et al, 2014). In addition to the RBPs studied here, others including ATXN2, TIA-1, hnRNPA2/B1, and TAF15 have been found to harbor ALS-related mutations in PrLDs (Mann and Donnelly, 2021; Milicevic et al, 2022). The structural commonalities among these ALS-associated RBPs suggest that they may share a common pathophysiological mechanism. We hypothesize that the LLPS potential shared by FUS, hnRNPA1, MATR3, and possibly other ALS-associated RBPs with disordered domains may play a key role in *REST* mRNA downregulation.

In our analysis, REST was dysregulated in iMNs harboring the ALS-associated FUS P525L mutation, which resides within the COOH-terminal NLS. This mutation interferes with the interaction between FUS and the nuclear import receptor Transportin, leading to cytoplasmic mislocalization (Dormann et al, 2010). The NLS region is also intrinsically disordered (Basu et al, 2022), suggesting that disruptions in intranuclear phase separation may additionally contribute to REST dysregulation. While we consider it likely that the resulting nuclear loss of function is the primary driver of *REST* mRNA dysregulation in our FUS P525L/+ iMNs, we cannot exclude the possibility that altered intranuclear LLPS properties of mutant FUS also contribute to this dysregulation. In particular, impaired LLPS, driven by altered FUS P525L binding to G-quadruplex-containing RNAs (Ishiguro et al, 2021), represents one potential mechanism. Further studies in disease-relevant neuronal models will be necessary to determine whether and how LLPS-dependent functions are involved in REST regulation.

We found that REST is overexpressed in motor neurons of individuals with sporadic ALS. A recent study showed that dysfunction of TDP-43 in sporadic ALS induces dysregulation of target RNA metabolism while the protein is still localized within the nucleus, before its mislocalization to the cytoplasm (Rothstein et al, 2023). Whereas mislocalization of RBPs other than TDP-43 is not frequently encountered in sporadic ALS (Honda et al, 2015; Tada et al, 2018; Tyzack et al, 2019), similar dysfunction of FUS, MATR3, and hnRNPA1 might also be evident while they remain in the nucleus. We speculate that such impaired function of these ALS-associated RBPs might lead to the upregulation of REST, potentially contributing to the repression of synapse-related genes such as *UNC13A*, in the motor neurons of individuals with sporadic ALS. With regard to the potential therapeutic approach of restoring UNC13A expression with a splice-correcting antisense oligonucleotide in individuals with ALS associated with TDP-43 pathology (Keuss et al, 2024), it may also be important to take into account the potential overexpression of REST in motor neurons of such patients.

In summary, we have revealed that the loss of each of the four RBPs may contribute to ALS pathogenesis by convergence on a common pathophysiological pathway initiated by the downregulation of

UNC13A expression. The mechanism by which the loss of TDP-43 gives rise to the attenuation of UNC13A expression is distinct from that by which the loss of MATR3, FUS, or hnRNPA1 does so, with the latter mechanism being mediated by transcriptional repression of *UNC13A* in a manner dependent on the upregulation of REST. The identification of this mechanism may provide a basis for the development of new therapeutic agents for ALS that target REST activity.

# Methods

### Reagents and tools table

| Reagent/resource | Reference or source | Identifier or catalog number |
|---|---|---|
| **Experimental models** | | |
| HEK293T | ATCC | CRL-3216 |
| SH-SY5Y | ATCC | CRL-2266 |
| U2OS | ATCC | HTB-96 |
| iPSC (201B7) | Takahashi et al, 2007 | RIKEN HPS0063 |
| iPSC (FUS-008-1-G2) | Setsu et al, 2025 | N/A |
| **Recombinant DNA** | | |
| Plasmid: pEN_TTGmiRc2 | Shin et al, 2006 | Addgene 25753 |
| Plasmid: pSLIK-neo | Shin et al, 2006 | Addgene 25735 |
| Plasmid: pSLIK-neo-FLAG-FUS (FL) | Nakaya, et al, 2020 | N/A |
| Plasmid: pSLIK-neo-FLAG-FUS-ΔGly | Nakaya, et al, 2020 | N/A |
| Plasmid: pSLIK-neo-FLAG-FUS-ΔRRM | Nakaya, et al, 2020 | N/A |
| Plasmid: pSLIK-neo-FLAG-FUS-ΔRGG1 | Nakaya, et al, 2020 | N/A |
| Plasmid: pSLIK-neo-FLAG-FUS-ΔZnf | Nakaya, et al, 2020 | N/A |
| Plasmid: pSLIK-neo-FLAG-FUS-ΔRGG2 | Nakaya, et al, 2020 | N/A |
| Plasmid: pSLIK-neo-3FLAG-FUS-27YS | This study | N/A |
| Plasmid: pSLIK-neo-3FLAG-FUS-27YS-SV40NLS | This study | N/A |
| Plasmid: pSLIK-neo- 3FLAG-hnRNPA1 | This study | N/A |
| Plasmid: psPAX2 | Didier Trono Lab | Addgene 12260 |
| Plasmid: pMD2.G | Didier Trono Lab | Addgene 12259 |
| Plasmid: PB-TA-ERN | Kim et al, 2013 | Addgene 80474 |
| Plasmid: PB-TA-MATR3 | This study | N/A |
| Plasmid: Super PiggyBac Transposase Expression Vector | System Biosciences | PB210PA-1 |
| Plasmid: pGL4.12 [luc2CP] | Promega | E6671 |
| Plasmid: pGL4.12-UNC13A-WT | This study | N/A |

| Reagent/resource | Reference or source | Identifier or catalog number |
|---|---|---|
| Plasmid: pGL4.12-UNC13A-ΔR | This study | N/A |
| Plasmid: pRL-CMV | Promega | E2261 |
| Plasmid: pSpCas9(BB)-2A-Puro(PX459) V2.0 | Ran et al, 2013 | Addgene 62988 |
| **Antibodies** | | |
| Rabbit anti-TDP-43 | Proteintech | 10782-2-AP; RRID AB_615042 |
| Rabbit anti-FUS | Bethyl | A300-293A; RRID AB_263409 |
| Rabbit anti-FUS | Bethyl | A300-302A; RRID AB_309445 |
| Mouse anti-FUS | Santa Cruz | sc-47711; RRID AB_2105208 |
| Mouse anti-hnRNPA1(9H10) | Santa Cruz | sc-56700; RRID AB_629651 |
| Rabbit anti-MATR3 | Abcam | ab151714; RRID AB_2491618 |
| Rabbit anti-MATR3 | Abcam | ab151739; RRID AB_2885091 |
| Rabbit anti-PARP (46D11) | Cell Signaling | 9532; RRID AB_659884 |
| Rabbit anti-Cleaved PARP (D64E10) | Cell Signaling | 5625; RRID AB_10699459 |
| Rabbit anti-UNC13A | Proteintech | 55053-1-AP; RRID AB_10804173 |
| Rabbit anti-UPF1 | Proteintech | 23379-1-AP; RRID AB_11232421 |
| Rabbit anti-REST (E3L2I) | Cell Signaling | 88188; RRID N/A |
| Rabbit anti-REST | Millipore | 09-019; RRID AB_1587469 |
| Rabbit anti-REST | Bethyl | IHC-00141; RRID AB_2285179 |
| Mouse anti-G3BP | BD Bioscience | 61126; RRID AB_398438 |
| Mouse anti-FLAG (M2, HRP conjugated) | Sigma | A8592; RRID AB_439702 |
| Rabbit anti-FLAG | Sigma | F7425; RRID AB_439687 |
| Mouse anti-β-Actin (8H10D10) | Cell Signaling | 3700; RRID AB_2242334 |
| Mouse anti-HSP70 | BD Biosciences | 610607; RRID AB_397941 |
| Mouse anti-HSP90 | BD Biosciences | 610418; RRID AB_397798 |
| Mouse anti-HB9 | DSHB | 81.5C10-C; RRID AB_2145209 |
| Mouse anti Tubulin β3 (TUBB3) | BioLegend | 801202; RRID AB_2313773 |
| Mouse anti-IgG | Santa Cruz | sc-2025; RRID AB_737182 |
| Rabbit anti-IgG | Sigma | I5006; RRID AB_1163659 |
| Mouse anti-IgG (HRP conjugated) | Promega | W4021; RRID AB_430834 |

| Reagent/resource | Reference or source | Identifier or catalog number |
|---|---|---|
| Rabbit anti-IgG (HRP conjugated) | Promega | W4011; RRID AB_430833 |
| Rabbit anti-IgG True Blot | eBioscience | 18-8816-33; RRID AB_2610848 |
| Rabbit IgG (H + L) Alexa Fluor 555 | Invitrogen | A21429; RRID AB_2535850 |
| Mouse IgG (H + L) Alexa Fluor 594 | Invitrogen | A11032; RRID AB_2534091 |
| Rabbit IgG (H + L) Alexa Fluor 488 | Invitrogen | A11008; RRID AB_143165 |
| Mouse IgG2a Alexa Fluor IgG2a 647 | Invitrogen | A21241; RRID AB_2535810 |
| Mouse IgG2b Alexa Fluor IgG2a 488 | Invitrogen | A21141; RRID AB_2535778 |
| **Oligonucleotides and other sequence-based reagents** | | |
| PCR Primers | This study | Table EV2 |
| siRNA targeting UPF1 | Thermo Fisher | 4392420; Assay ID s11927 |
| siRNA targeting UPF1 | Thermo Fisher | 4392420; Assay ID s11928 |
| siRNA targeting REST | Thermo Fisher | 4392420; Assay ID s11933 |
| siRNA targeting REST | Thermo Fisher | 4392420; Assay ID s11934 |
| Negative Control siRNA (medium GC) | Thermo Fisher | 12935300 |
| **Chemicals, enzymes, and other reagents** | | |
| Aprotinin | Sigma | A1153 |
| Leupeptin | Sigma | L-8511 |
| Phenylmethylsulfonyl fluoride | Wako | 162-12182 |
| Y27632 | Nakalai | 08945-84 |
| StemFit AK02N | Takara Bio | AJ100 |
| KBM neural stem cell medium | Koujin Bio | 1605100 |
| SB431542 | Sigma | S4317 |
| Dimethindene Maleate | Santa Cruz | sc-361329 |
| BDNF | R&D Systems | 248-BD |
| GDNF | Alomone Labs | G-240 |
| DAPT | Sigma | D5942 |
| PD0332991 | Sigma | PZ0199 |
| CHIR99021 | Cayman | 13122 |
| iMatrix-511 | Takara Bio | T304 |
| poly-L-lysin | Sigma | P8920 |
| Matrigel | Thermo Fisher | CB-40234 |
| Cycloheximide | Sigma | C-1988 |
| Doxycycline | Takara Bio | 631311 |
| Pierce Protein G Plus Agarose | Thermo Fisher | 22851 |
| Dynabeads Protein G | Invitrogen | 10004D |
| RNase OUT | Invitrogen | 10777019 |
| LR Clonase II | Invitrogen | 11791020 |

| Reagent/resource | Reference or source | Identifier or catalog number |
|---|---|---|
| Polybrene | Sigma | TR-1003 |
| RNAiMax (Lipofectamine) | Thermo Fisher | 13778030 |
| PEI MAX | Polysciences | 24765-1 |
| Fugene HD | Promega | E2311 |
| G-Block | Genostaff | GB-01 |
| Peroxidase-conjugated streptavidin | Nichirei | 426061 |
| Mayer's hematoxylin | Muto Pure Chemicals | 30002 |
| Malinol | Muto Pure Chemicals | 10781 |
| SeV-LHX3-NGN2-ISL1 | Setsu, et al, 2025 | N/A |
| **Software** | | |
| ImageJ | Schneider et al, 2012 | https://imagej.nih.gov/ij |
| R | R Core Team | https://cran.r-project.org |
| NDP.view2 Plus Viewing software | Hamamatsu Photonics | U12388-02 |
| **Other** | | |
| TruSeq Standard mRNA LT Sample Prep Kit | illumina | RS-122-9005DOC |
| PrimeSTAR Mutagenesis Basal Kit | Takara Bio | R046A |
| RIP-Assay Kit | MBL | RN1001 |
| Click-it Nascent RNA Capture Kit | Invitrogen | C10365 |
| SimpleChIP Enzymatic Chromatin IP Kit | Cell Signaling | 9003S |
| SV Total RNA isolation system | Promega | Z3100 |
| PrimeScript RT reagent kit | Takara Bio | RR037A |
| SuperScript VILO cDNA Synthesis Kit | Thermo Fisher | 11754050 |
| Fast SYBR Green Master Mix | Thermo Fisher | 4385610 |
| Dual-Glo Luciferase assay system | Promega | E2920 |
| Avidin/Biotin Blocking Kit | Vector | SP-2001 |
| illumina hiseq 2500 | illumina | |
| BZ-X800 | Keyence | |
| NanoZoomer S20 Digital slide scanner | Hamamatsu Photonics | C16300-01 |

## Plasmid construction

For construction of Cas9-sgRNA plasmids, sgRNAs were designed with the use of CRISPR direct (https://crispr.dbcls.jp) and subcloned into the pSpCas9(BB)-2A-Puro (PX459) V2.0 vector (Addgene) (Ran et al, 2013). Lentivirus vectors for doxycycline-inducible NH$_2$-terminally FLAG-tagged human FUS (WT and deletion mutants) were kindly provided by T Nakaya (Nakaya, 2020). For the construction of the doxycycline-inducible 3×FLAG-hnRNPA1 vector, cDNA encoding human hnRNPA1 was initially cloned into the pENTR vector (Thermo Fisher Scientific) and verified by sequencing. The hnRNPA1 sequence of the resulting

vector was then transferred by recombination with the use of LR Clonase II (Thermo Fisher Scientific) into a p3xFLAG vector that had been modified to include attR sites. The 3×FLAG-tagged hnRNPA1 cDNA was subsequently amplified from the p3xFLAG vector by PCR and cloned with the use of an In-Fusion Cloning Kit (Takara Bio) into the pEN_TTGmiRc2 vector (Addgene), from which the EGFP and miR-30a coding regions had previously been removed with restriction enzymes. The resulting pEN_TTG-3xFLAG-hnRNPA1 vector was finally subjected to recombination with the pSLIK-neo destination plasmid (Addgene). Human MATR3 cDNA was cloned into the pENTR vector and subjected to similar verification and recombination steps as for hnRNPA1 cDNA, yielding a recombind PB-TA-ERN vector (Addgene) with the use of LR Clonase II. In addition, doxycycline-inducible lentivirus vectors for LLPS-deficient FUS mutants were constructed from cDNAs synthesized by GeneArt Custom Gene Synthesis (Invitrogen) and assembled in a similar manner to that adopted for the doxycycline-inducible 3×FLAG-hnRNPA1 construct. For construction of the WT luciferase vector for the *UNC13A* promoter, the promoter region of human *UNC13A* as defined in the UCSC genome browser was amplified from genomic DNA by PCR and then inserted into the pGL4.12 vector, which had previously been digested with BamHI and HindIII. A construct lacking the REST binding motif was generated with the use of a PrimeSTAR Mutagenesis Basal Kit (Takara Bio).

## Cell culture and transfection

HEK293T, U2OS and SH-SY5Y cells were maintained in Dulbecco's modified Eagle's medium (DMEM) supplemented with 10% fetal bovine serum, penicillin (50 U/ml), streptomycin (50 μg/ml), 2 mM L-glutamine, 1% MEM–nonessential amino acids, and 1% sodium pyruvate. HEK293T cells were transiently transfected using PEI MAX (Polyscience), and SH-SY5Y cells were transfected using FuGENE HD (Promega). Feeder-free iPSCs were maintained in StemFit AK02N medium. Cells were dissociated using 0.5× TripLE Select (Thermo Fisher Scientific) and seeded at $0.3–1 \times 10^4$ cells/well in six-well plates treated with 2 μL/mL iMatrix-511 (Matrixome). Y27632 (10 μM; Nacalai) was added only for the first day. The medium was changed every other day.

## RNA-seq analysis

A TruSeq Standard mRNA LT Sample Prep Kit (Illumina) was used for library preparation. Sequencing was conducted on an Illumina HiSeq 2500 instrument to yield 151-nucleotide paired-end reads. Adapter sequences and low-quality bases were trimmed with the use of Trim_Galore. The resulting high-quality reads were aligned to the reference genome (GRCh38) with the use of STAR. Gene expression analysis, including k-means clustering and transcription factor motif analysis, was performed with iDEP (versions 2.01 and 96, respectively) (Ge et al, 2018).

## RNA interference

Cells were transfected twice with 20 nM siRNAs with the use of the RNAiMax reagent (Thermo Fisher Scientific). For morphological phenotypic analysis, cell images were captured 120 h after initial transfection. For gene expression analysis, the second transfection was performed 72 h after the onset of the first, and the cells were harvested for analysis 24 h after the second transfection.

## RT-PCR and RT-qPCR

For SH-SY5Y cells, RNA was isolated from cells with the use of an SV Total RNA Isolation System (Promega) and was subjected to RT with a PrimeScript RT Reagent Kit (Takara Bio). For qPCR, the resulting cDNA was amplified by real-time PCR analysis with the use of a StepOnePlus Real-Time PCR System (Life Technologies) and Fast SYBR Green Master Mix (Life Technologies). Data were analyzed with the 2-ΔΔCT method and normalized by the amount of human *GAPDH* mRNA. For PCR, cDNA was amplified with the use of PrimeSTAR Max DNA Polymerase (Takara Bio), and the amplification products were subjected to 2% agarose gel electrophoresis. For iMNs, total RNA was extracted on day 14 post-SeV infection using an RNeasy Mini Kit (QIAGEN) with DNase I treatment. cDNA was synthesized using an iScript cDNA Synthesis Kit (Bio-Rad). qPCR analysis was performed with the 2-ΔΔCT method, with normalization to human *ACTB* mRNA. The primer sequences for qPCR and PCR are listed in Table EV2.

## Nascent RNA purification

Cells were seeded at a density of $1 \times 10^6$ cells per well in six-well plates, cultured overnight, and labeled with 0.2 mM 4-EU for 1 h. Total RNA was extracted from the cells with the use of an SV Total RNA Isolation System (Promega), and nascent RNA was purified from the total RNA with a Click-iT Nascent RNA Capture Kit (Invitrogen). In brief, the extracted RNA was subjected to biotinylation by incubation with Click-iT Reaction Buffer, Biotin Azide, and $CuSO_4$ solution for 30 min at room temperature with vortex mixing. Biotinylated nascent RNA was captured with Dynabeads MyOne Streptavidin T1 magnetic beads (Invitrogen). The bead-bound RNA was washed and then immediately subjected to cDNA synthesis with a SuperScript VILO cDNA Synthesis Kit (Thermo Fisher). qPCR analysis was performed with the 2-ΔΔCT method, with normalization to human *18S rRNA*.

## Immunoprecipitation and immunoblot analysis

Cells were washed with phosphate-buffered saline and then lysed for 10 min at 4 °C in NP-40 lysis buffer (0.5% Nonidet P-40, 50 mM Tris-HCl [pH 7.5], 150 mM NaCl, 10% glycerol) supplemented with a protease inhibitor cocktail (aprotinin [10 μg/ml, Sigma], leupeptin [10 μg/ml, Peptide Institute], 1 mM phenylmethylsulfonyl fluoride [Wako]). The lysates were centrifuged at $20,000 \times g$ for 15 min at 4 °C, and the resulting supernatants were harvested for immunoblot analysis. For the preparation of an insoluble fraction, the pellet obtained by the centrifugation step was solubilized by ultrasonic treatment in urea buffer (7 M urea, 2 M thiourea, 1% CHAPS detergent, 30 mM Tris-HCl [pH 8.0], 25 mM imidazole). For immunoblot analysis, the samples were mixed with Laemmli buffer and fractionated by SDS-polyacrylamide gel electrophoresis. The separated proteins were transferred to a polyvinylidene difluoride membrane (Millipore), which was then incubated consecutively with primary antibodies, horseradish peroxidase-conjugated secondary antibodies, and chemiluminescence reagents. Signals were detected with a ChemiDoc Touch System (Bio-Rad). For immunoprecipitation, the lysate supernatants were incubated with Dynabeads Protein G (Life Technologies) conjugated with antibodies to FUS, hnRNPA1 or to IgG (negative control). The resulting immunoprecipitates were washed

three times with PBS containing 0.1% Triton X-100 and 10% glycerol and were then subjected to immunoblot analysis.

## Generation of RBP-KO cell lines

Cells were transiently transfected with Cas9-sgRNA plasmids and exposed to puromycin (5 μg/ml) for 2 days, and the surviving cells were cloned by the limiting dilution method. The single-cell-derived clones were expanded and then validated by genomic PCR analysis of extracted DNA followed by sequencing. The target sequences of the sgRNAs were as follows: 5′-CCCATG GAAAA-CAACCGAAC-3′ within *TARDBP*, 5′-CCAGCAGTCATCTCT-CAGTA-3′ within *MATR3*, 5′-CGGACATGGCCTCAAACGgt-3′ within *FUS*, and 5′-TGCCGTCATGTCTAAGTCAG-3′ within *HNRNPA1*.

## Restoration of RBP expression in RBP-KO cell lines

For doxycycline-inducible FLAG-FUS (WT and mutant) or 3 × FLAG-hnRNPA1 expression in corresponding KO cell lines, the FUS-KO or hnRNPA1-KO cells were infected with lentiviruses in the presence of polybrene (8 μg/ml) and then subjected to selection with G418 (400 μg/ml, Wako) for at least 7 days. The lentiviruses were produced by transfection of HEK293T cells with the pSLIK-neo vectors containing the rtTA-TRE–regulated FLAG-FUS or 3 × FLAG-hnRNPA1 cDNA sequences, as well as with the packaging plasmid psPAX2 and the envelope plasmid pMD2.G (Addgene). For doxycycline-inducible MATR3 expression in MATR3-KO cells, the cells were transfected with the corresponding PiggyBac vector and the Super PiggyBac Transposase Expression Vector (System Biosciences) and were then subjected to selection with G418 (400 μg/ml) for 2 weeks. For the induction of each RBP, cells were treated with doxycycline (1 μg/ml, LKT Laboratories) for 2 days.

## SG formation assay

U2OS cells expressing doxycycline-inducible FLAG-FUS (WT and mutants) were generated by lentiviral infection in the presence of polybrene (8 μg/mL), followed by G418 selection (800 μg/mL, Wako) for 7 days. Cells were seeded in six-well plates with a coverslip at the bottom and treated with doxycycline for 24 h before exposure to 1 mM sodium arsenite for 30 min. Cells were then fixed with 4% paraformaldehyde for 10 min, permeabilized with 0.5% Triton X-100 in PBS, and blocked with 1% bovine serum albumin (BSA). For immunostaining, cells were incubated overnight at 4 °C with FLAG (Sigma) and G3BP (BD Bioscience) antibodies, followed by Alexa Fluor 594-conjugated anti-mouse IgG and Alexa Fluor 488-conjugated anti-rabbit IgG at room temperature for 45 min. After PBS washes, nuclei were stained with Hoechst and mounted on glass slides. Imaging was performed using a BZ-X800 fluorescence microscope (Keyence). For quantification, at least 90 cells per field were analyzed using the Cell Counter plugin in ImageJ.

## Analysis of pre-mRNA

Total RNA was extracted from cells with the use of an SV Total RNA Isolation System (Promega) and was subjected to RT with a PrimeScript RT Reagent Kit (Takara Bio) but without an oligo(dT)

primer. The RT reaction was also performed with (+RT) or without (−RT) reverse transcriptase in order to control for genomic DNA contamination. The reaction products were subjected to qPCR analysis with primers designed to target an intronic region of the *UNC13A* gene. The abundance of *UNC13A* pre-mRNA was compared across cell lines by normalization of the cycle threshold (CT) values obtained from the +RT samples by those from the −RT samples.

## Luciferase reporter assay

The human *UNC13A* promoter region spanning nucleotides −200 to +121 relative to the TSS was amplified by PCR and ligated into the pGL4.12 luciferase reporter vector (Promega) to yield pGL4.12-UNC13A-WT. The pGL4.12-UNC13A-ΔR vector for the deletion mutant lacking an intact REST binding site was constructed with a PrimeSTAR Mutagenesis Basal Kit (Takara Bio). Luciferase activities of cell lysates were measured with a Dual-Glo Luciferase Assay System (Promega) and a Berthold Centro LB960 instrument. The ratio of firefly to *Renilla* luciferase activity was calculated. For knockdown of REST in HEK293T cells, the cells were transfected with Cas9-sgRNA vectors the day before transfection with the promoter-luciferase constructs. The target sequences of the sgRNAs were as follows: sgRNA pair #1, 5′-GTTATGGCCACCCAGG-TAAT-3′ and 5′-AGACATATGCGTACTCATTC-3′; sgRNA pair #2, 5′-CAACAGTGAGCGAGTATCAC-3′ and 5′-GTCTTCTGA-GAACTTGAGTA-3′.

## RIP assay

RIP was performed with a RIP-Assay Kit (MBL). In brief, cells were lysed in RIP lysis buffer supplemented with 1.5 mM dithiothreitol, RNase OUT (50 U/ml, Invitrogen), and protease inhibitors. The lysates were incubated for 3 h at 4 °C under gentle rotation with Pierce Protein G Plus Agarose (Thermo Fisher Scientific) conjugated to antibodies specific for TDP-43 (Proteintech, 10782-2-AP), FUS (Santa Cruz Biotechnology, sc-47711), MATR3 (Abcam, ab151739), or hnRNPA1 (Santa Cruz Biotechnology, sc-56700). Mouse IgG (Santa Cruz Biotechnology, sc-2025) and rabbit IgG (Sigma, I5006) were used as controls. The beads were then washed extensively to remove nonspecifically bound material, after which coprecipitated RNA was isolated from the beads with an SV Total RNA Isolation System (Promega). The isolated RNA was subjected to RT with a PrimeScript RT Reagent Kit (Takara Bio), and the resulting cDNA was subjected to PCR amplification followed by 2% agarose gel electrophoresis for detection of *REST* mRNA.

## ChIP-qPCR analysis

ChIP assays were performed with the use of a SimpleChIP Enzymatic Chromatin IP Kit (Cell Signaling). In brief, cells (~1 × 10^7 per IP) cultured in a 15-cm dish were fixed with 1% formaldehyde for 10 min at room temperature, subjected to quenching, and enzymatically digested for 20 min at 37 °C. Chromatin was sheared by ultrasonic treatment (five 30-s applications) and incubated overnight at 4 °C with 2 μg of antibodies to REST (Millipore) or IgG (negative control) per 10 μg of chromatin. Chromatin in immune complexes was then

precipitated by incubation with protein G-conjugated magnetic beads, washed, and eluted. After reversal of cross-links and purification, precipitated DNA was subjected to qPCR analysis with specific primers (Table EV2).

## Differentiation and immunocytochemistry of iMNs

iPSC lines, WT (201B7) and FUS P525L/+ (FUS-008-1-G2) were used with approval from the ethics committee of Keio University School of Medicine (approval no. 20080016) and in accordance with the Declaration of Helsinki. These cells were induced to differentiate into motor neurons (iMNs) as previously described (Setsu et al, 2025). In brief, iPSCs were cultured in StemFit AK02N with 10 μM Y27632 and 2 μL/mL iMatrix-511 for 5 days, followed by embryoid body (EB)-like state induction with a chemical induction medium. SeV vectors (SeV-LHX3-NGN2-ISL1, Repli-tech Co., Ltd.) were applied at MOI 5, and cells were maintained in KBM neural stem cell medium (KOHJIN BIO) supplemented with B27, antibiotics, ascorbic acid, BDNF, GDNF, DAPT, and Y27632. Y27632 was removed after 1 day. Medium was changed on days 1, 3, 4, 7, 10, and 13 post-SeV infection. DAPT was added until day 7, and PD0332991 was used on days 4 and 7 to remove proliferating cells. For immunocytochemistry, cells were fixed in 4% paraformaldehyde, permeabilized, blocked, and incubated on day 7 with primary antibodies against HB9 (DSHB), TUBB3 (BioLegend), and FUS (Bethyl, A300-293A), followed by species-specific secondary antibodies conjugated to Alexa Fluor 488, Alexa Fluor 555, or Alexa Fluor 647 (Invitrogen, Thermo Fisher Scientific) and Hoechst 33258 (Sigma) for nuclear counterstaining.

## Immunohistochemistry of human samples

Archived, de-identified spinal cord tissue from autopsies was used. Informed consent for autopsy and research use was obtained from legal representatives in accordance with institutional guidelines. The study was approved by the Ethics Committee of the Graduate School of Medicine, Tohoku University, and all procedures complied with the Declaration of Helsinki and the Belmont Report. Patients were diagnosed with ALS according to the revised El Escorial criteria (Brooks et al, 2000), with diagnoses being further confirmed pathologically by postmortem examination. Tissue from each level of the spinal cord was either immediately placed in 10% buffered formalin or embedded in paraffin for neuropathologic examination. Immunohistochemistry for FUS and TDP-43 was performed as previously described (Akiyama et al, 2019). For REST staining, tissue sections were depleted of paraffin with xylene, rehydrated with a graded series of ethanol solutions in phosphate-buffered saline, and subjected to antigen retrieval by microwave irradiation for 20 min in 10 mM Tris/1 mM EDTA (pH 9.0). Endogenous peroxidase activity was blocked by treatment with 0.3% $H_2O_2$ in methanol for 30 min, after which the sections were exposed to G-Block (Genostaff), and then Avidin/Biotin Blocking Kit (Vector), and incubated consecutively overnight at 4 °C with rabbit monoclonal antibodies to REST (Bethyl), for 30 min at room temperature with biotin-conjugated goat antibodies to rabbit IgG, and for 5 min at room temperature with peroxidase-conjugated streptavidin (Nichirei). Peroxidase activity was visualized by staining with diaminobenzidine (DAB), and the sections were

counterstained with Mayer's hematoxylin (Muto), dehydrated, and mounted with Malinol (Muto). Images were obtained using NanoZoomer S20 Digital slide scanner and analyzed with NDP.view2 Plus Viewing software. To quantify the proportion of REST-positive motor neurons, images were acquired with a field of view of 1.4 mm × 0.87 mm, including the anterior horn region. Motor neurons were defined as cells exhibiting a tapered morphology with clearly identifiable neurite origins. Cells with ambiguous morphology due to sectioning artifacts or staining heterogeneity were excluded from the analysis. ImageJ (version 2.16.0) was used to identify REST-positive motor neurons via the Color Deconvolution function, which separates DAB staining from background signals. REST-positivity was defined as a nuclear REST staining intensity greater than twice the background level. The percentage of REST-positive motor neurons was calculated as the proportion of total motor neurons within each field.

## Quantification and statistical analysis

Relative band intensities were quantified by densitometry with the use of ImageJ (Schneider et al, 2012). Statistical analysis was performed with R software. Data were compared between two groups with the unpaired two-tailed Student's $t$ test or Mann–Whitney $U$ test, and among three or more groups either by one-way analysis of variance (ANOVA) followed by Tukey's post hoc test, or by the Kruskal–Wallis test followed by Dunn's test. Data are presented as means ± SEM unless indicated otherwise, and a $P$ value of <0.05 was considered statistically significant.

# Data availability

RNA-seq data have been deposited in GEO under the accession number GSE292352.

The source data of this paper are collected in the following database record: biostudies:S-SCDT-10_1038-S44318-025-00506-0.

# Peer review information

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

## Acknowledgements

This work was supported by KAKENHI grants (JP22K15702 to YW, JP21K07411 to N.S., JP23H02821 to MA, JP21H02458, JP24K02300 to KN, JP21H05278, JP22K15736 to SM) from the Japan Society for the Promotion of Science (JSPS), by the Japan Agency for Medical Research and Development (AMED) (JP24ek0109631 to MA; JP23bm1123046, JP23kk0305024 to SM; JP21wm0425009, JP23bm1423002 to HO), by the Japan ALS Association (to YW), by the Yukihiko Miyata Memorial Trust for ALS Research (to YW), by the Daiichi Sankyo Foundation of Life Science (to SM), by the Kato Memorial Trust for Nambyo Research (to SM), and by the Inamori Foundation (to SM). This work was also supported by the Biomedical Research Core of Tohoku University Graduate School of Medicine and the Support System for Young Researchers to Use Research Equipment, Instruments, and Devices in Tohoku University. We thank K Kuroda, M Nakagawa, Y Nagasawa and M Kikuchi for general technical and secretarial assistance; S Nakamura and F Ozawa for assistance with iPSC-related experiments; H Aoyama for assistance with immunohistochemistry. We also appreciate discussions with R Harada, K Ikeda, M Shirota, and R Funayama, as well as A Gitler (Stanford University). We are also grateful to T Nakaya (International University of Health and Welfare) for providing pSLIK-neo-FLAG-FUS and the corresponding mutant plasmids; S Yamanaka, K Okita, and M Nakagawa (Kyoto University) for providing 201B7 iPSC line; Takeda Pharmaceutical Company for providing FUS-008-1-G2 iPSC line. We would like to thank the Answer ALS consortium for providing transcriptomics data used in this manuscript.

## Author contributions

**Yasuaki Watanabe**: Conceptualization; Resources; Data curation; Formal analysis; Funding acquisition; Validation; Investigation; Visualization; Methodology; Writing—original draft; Project administration; Writing—review and editing. **Naoki Suzuki**: Resources; Supervision; Funding acquisition; Writing—review and editing. **Tadashi Nakagawa**: Investigation; Methodology; Writing—review and editing. **Masaki Hosogane**: Methodology; Writing—review and editing. **Tetsuya Akiyama**: Data curation; Writing—review and editing. **Naotoshi Kageyama**: Investigation. **Yukino Funayama**: Investigation. **Hitoshi Warita**: Writing—review and editing. **Satoru Morimoto**: Resources; Funding acquisition; Investigation. **Hideyuki Okano**: Resources; Funding acquisition. **Masashi Aoki**: Resources; Funding acquisition; Writing—review and editing. **Keiko Nakayama**: Resources; Supervision; Funding acquisition; Project administration; Writing—review and editing.

Source data underlying figure panels in this paper may have individual authorship assigned. Where available, figure panel/source data authorship is listed in the following database record: biostudies:S-SCDT-10_1038-S44318-025-00506-0.

## Disclosure and competing interests statement

The authors declare no competing interests.

# Expanded View Figures

## A

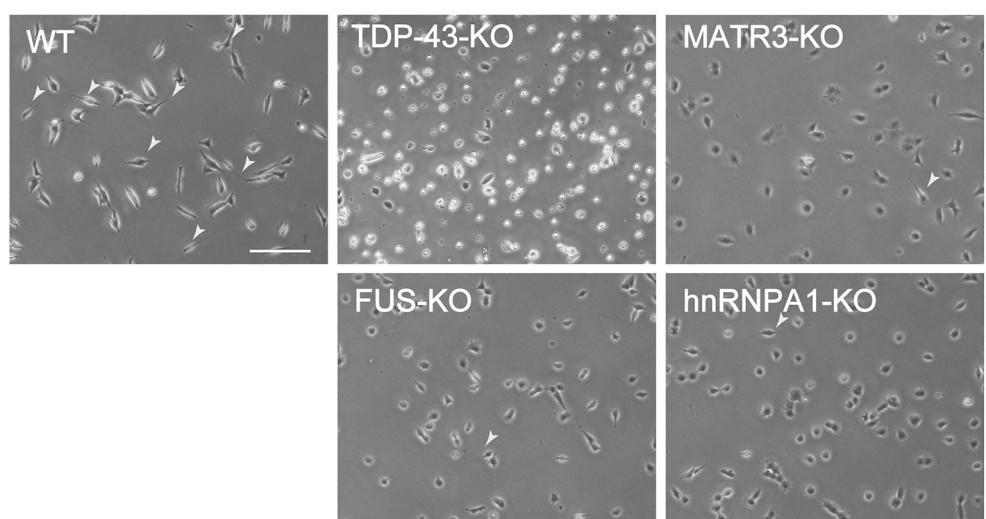

## B

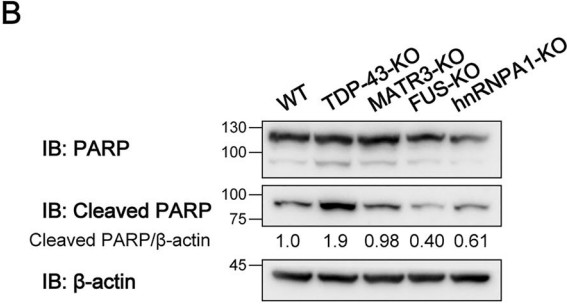

**Figure EV1. Characterization of RBP-KO cell lines, related to Fig. 1.**

(A) Bright-field images of WT cells and RBP-KO cell lines. Cells were seeded in iMatrix-coated wells, and images were captured the next day. Yellow arrowheads indicate cells with cytoplasmic processes. Scale bar = 200 μm. (B) Immunoblot analysis of PARP and cleaved PARP in WT cells and in RBP-KO cell lines. β-actin served as a loading control. The band intensity for cleaved PARP (normalized by that of β-actin) was quantified by densitometry.

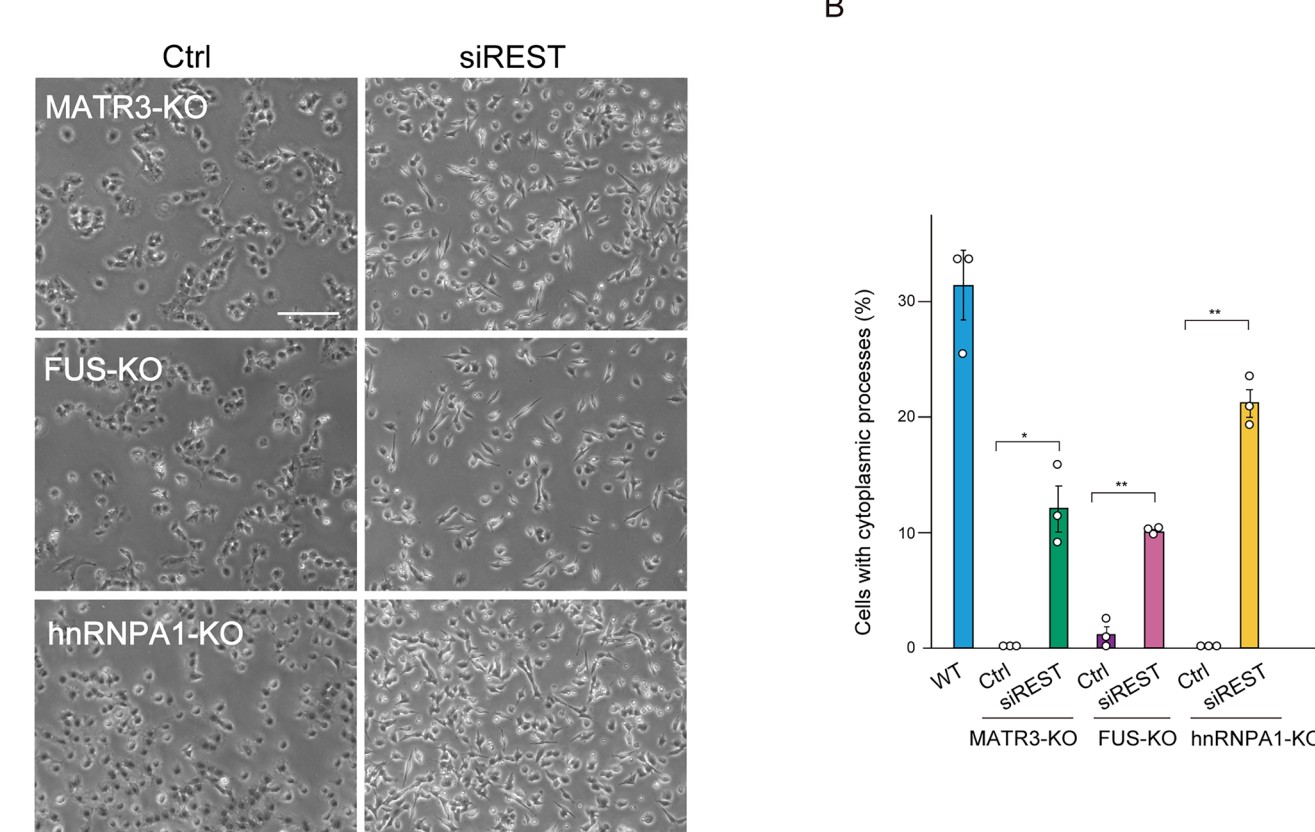

**Figure EV2.  Knockdown of REST rescues morphological defects in RBP-KO cell lines, related to Fig. 3.**

(A) Bright-field images of MATR3-, FUS-, and hnRNPA1-KO cell lines in iMatrix-coated 6-well plate, 5 days after transfection with either a GC duplex (negative control) or a *REST* siRNA. Scale bar = 200 µm. (B) Quantification of cytoplasmic process formation in MATR3-KO, FUS-KO, and hnRNPA1-KO cell lines shown in (A), with WT cells included as a reference. Cells were classified as having cytoplasmic processes if their extensions exceeded the length of the major axis of the cell body. Data are mean ± SEM from three independent experiments. **$P < 0.01$, ***$P < 0.001$, ****$P < 0.0001$ (Student's *t* test); exact *P* values: MATR3-KO, $P = 0.0036$; FUS-KO, $P = 0.00027$; hnRNPA1-KO, $P = 6.6e\text{-}05$.

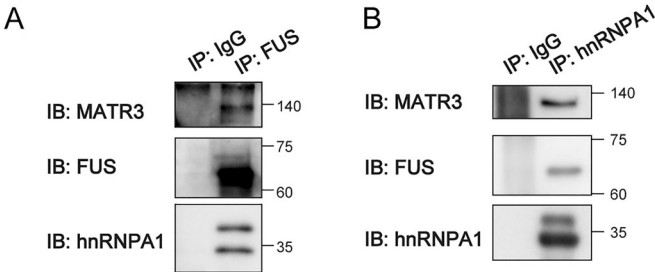

**Figure EV3. FUS, MATR3, and hnRNPA1 bind to each other, related to Fig. 5.**

(A) Immunoprecipitation of FUS from WT cell lysate. The immunoprecipitate (IP) obtained with antibodies to FUS or with control immunoglobulin G (IgG) were subjected to immunoblot analysis with antibodies to MATR3, FUS or hnRNPA1. (B) Immunoprecipitation of hnRNPA1 from WT cell lysate.

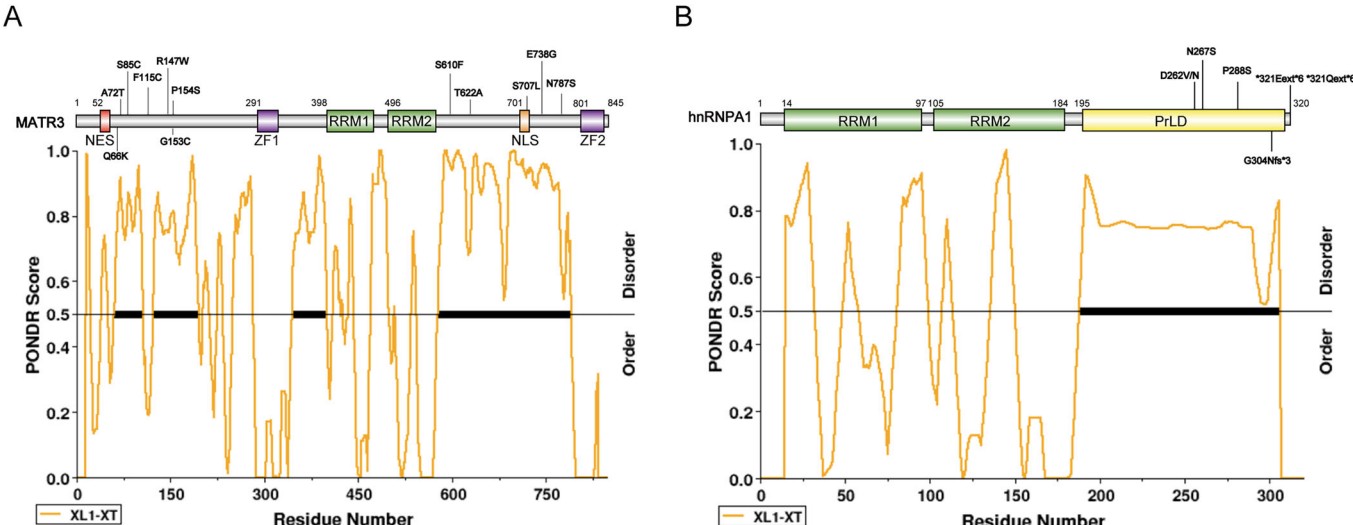

**Figure EV4.  Domain structures of MATR3 and hnRNPA1 with ALS-associated mutations, related to Fig. 6.**

(**A**) Schematic representation of the domain structure of MATR3 (top). NES, nuclear export signal; ZnF, zinc finger; RRM, RNA recognition motif; NLS, nuclear localization signal. The locations of ALS-associated mutations in MATR3 referenced from Malik and Barmada, 2021 are also shown. Disorder prediction for MATR3 residues by PONDR (http://www.pondr.com) is shown at the bottom. (**B**) Schematic representation of the domain structure of hnRNPA1 (top). RRM, RNA recognition motif; PrLD, prion-like domain. The locations of ALS-associated mutations in hnRNPA1 referenced from Beijer et al, 2021 are also shown. G304Nfs*3 is a frameshift mutation, while *321Eext*6 and *321Qext*6 are extension mutations. Disorder prediction for hnRNPA1 residues by PONDR is shown at the bottom.

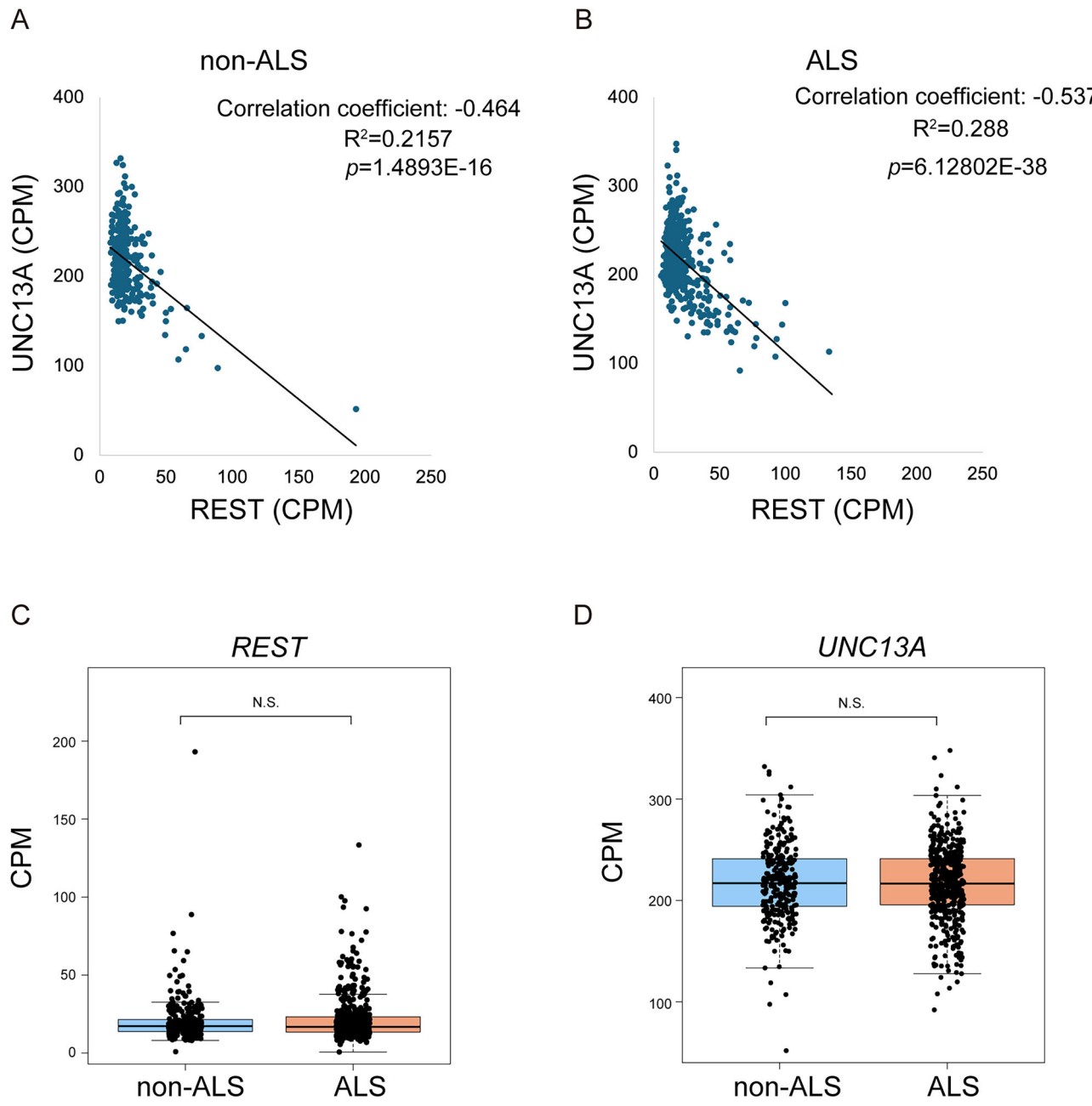

**Figure EV5.  *REST* and *UNC13A* expression in iPSC-derived motor neurons from the Answer ALS platform, related to Fig. 7.**

(A, B) Correlation analysis of *REST* and *UNC13A* expression using transcriptomics data from iPSC-derived motor neurons available on the Answer ALS platform. Each blue dot represents one sample. The line indicates the linear regression fit. The $R^2$ value represents the proportion of variance in *UNC13A* expression explained by REST expression, and the *p* value was calculated using Pearson's correlation test. (A) Non-ALS group ($n = 283$), (B) ALS group ($n = 490$). CPM, counts per million. (C, D) Comparison of *REST* (C) and *UNC13A* (D) expression between control (non-ALS, $n = 283$) and ALS (ALS, $n = 490$) using transcriptomics data from iPSC-derived motor neurons available on the Answer ALS platform. Box plots represent the median (center line), interquartile range (IQR; box), and whiskers indicating the most extreme data point within 1.5×IQR from the quartiles. CPM, counts per million. N.S., not significant (Mann–Whitney *U* test).

