## [Peer Review File · The EMBO Journal]

ALS-associated RNA-binding proteins promote UNC13A transcription through REST downregulation

Yasuaki Watanabe, Naoki Suzuki, Tadashi Nakagawa, Masaki Hosogane, Tetsuya Akiyama, Naotoshi Kageyama, Yukino Funayama, Hitoshi Warita, Satoru Morimoto, Hideyuki Okano, Masashi Aoki, and Keiko Nakayama

Corresponding author(s): Keiko Nakayama (keiko.nakayama.e4@tohoku.ac.jp) , Yasuaki Watanabe (yasuaki.watanabe.b8@tohoku.ac.jp)

Review Timeline:

Submission Date:	15th Oct 24
Editorial Decision:	25th Nov 24
Revision Received:	22nd Mar 25
Editorial Decision:	2nd May 25
Revision Received:	26th May 25
Accepted:	16th Jun 25

Editor: Ioannis Papaioannou

Transaction Report:

Dear Prof. Nakayama,

Thank you for submitting your manuscript EMBOJ-2024-119340 for consideration by The EMBO Journal, and for your patience during peer review. Your manuscript has now been seen by three experts in the field, and we have received the full set of their comments, which are included below.

As you will see, the referees recognize that the results presented in your manuscript are potentially interesting, significant, and relevant. Although they all find the work well-performed for the most part and the findings largely compelling, they also raise several concerns and list a number of points that should be addressed for strengthening the work and the manuscript further. These include (but are not limited to) concerns regarding the relevance of the findings to patient neurons, the lack of sufficient data to support the claims on synaptic function, the lack of proper controls for some experiments, as well as the clarity and reproducibility of some of the presented data. We find the referees' reports balanced and constructive, and we agree with the referees that addressing these points would significantly increase the solidity and impact of your manuscript.

Given the referees' comments and recommendations, I would like to invite you to submit a thoroughly revised version of the manuscript along with a detailed point-by-point response addressing all referees' comments. I should add that it is The EMBO Journal policy to allow only a single round of major revision, and acceptance of your manuscript will therefore depend on the completeness of your responses in this revised version. Please let me know if you have any questions or comments that you would like to discuss with me.

We generally allow three months as standard revision time (February 24, 2025). As a matter of policy, competing manuscripts published during this period will not negatively impact our assessment of the conceptual advance presented by your study. However, we request that you contact us as soon as possible upon publication of any related work, to discuss how to proceed. Should you foresee a problem in meeting this three-month deadline, please let us know in advance and we may be able to grant an extension.

Thank you for the opportunity to consider your work for publication in The EMBO Journal. I look forward to your revision.

Best regards,

Ioannis

Instructions for preparing your revised manuscript

1. When you are ready to submit the revision, please upload:

- A Word file of the manuscript text (including legends of main Figures, EV Figures and Tables). Please make sure that changes are highlighted (or "tracked") to be clearly visible.

- Individual production-quality figure files (one file per figure). When assembling your figures, please refer to our figure preparation guidelines in order to ensure proper formatting and readability in print as well as on screen:

If the data shown in a figure are obtained from n {less than or equal to} 2, please use scatter plots showing the individual data points.

- i. the name of the statistical test used to generate error bars and P values
- ii. the number (n) of independent experiments (please specify technical or biological replicates) underlying each data point (discussion of statistical methodology can be reported in the Materials and Methods section, but figure legends should contain a basic description of n , P , and the test applied)
- iii. the nature of the bars and error bars (s.d., s.e.m.).

- A point-by-point response to the referees' comments, with a detailed description of the changes made (as a word file). All referees' concerns must be fully addressed and their suggestions taken on board. When preparing your letter of response to the referees' comments, please bear in mind that this will form part of the Review Process File and will therefore be available online to the community. Please note that you have the possibility to opt out of the transparent process at any stage prior to publication by letting the editorial office know (contact@embojournal.org); if you do opt out, the Review Process File link will point to the following statement: "No Review Process File is available with this article, as the authors have chosen not to make the review process public in this case.". For more details on our Transparent Editorial Process, please visit our website: <https://www.embopress.org/page/journal/14602075/authorguide#transparentprocess>

- Expanded View (EV) files (replacing Supplementary Information) that are collapsible/expandable online. A maximum of 5 EV Figures can be typeset. EV Figures should be cited as "Figure EV1, Figure EV2" etc. in the text, and their respective legends should be included in the manuscript file after the legends of regular figures. See detailed instructions regarding Expanded View files here: <https://www.embopress.org/page/journal/14602075/authorguide#expandedview>

- For the figures that you do NOT wish to display as Expanded View figures, they should be bundled together with their legends in a single PDF file called "Appendix", which should start with a short Table of Contents (including page numbers). Appendix figures should be referred to in the main text as: "Appendix Figure S1, Appendix Figure S2" etc. Please see detailed instructions here: <https://www.embopress.org/page/journal/14602075/authorguide#expandedview>

- A complete author checklist, which you can download from our author guidelines (<https://www.embopress.org/page/journal/14602075/authorguide>). Please note that the checklist will also be part of the Review Process File.

2. Please note that no statistics should be calculated and shown in Figures if $n=2$. Please also note that each p value should be reported as an exact value.

3. Before submitting your revision, primary datasets (and computer code, where appropriate) produced in this study need to be deposited in appropriate public databases (see <https://www.embopress.org/page/journal/14602075/authorguide#dataavailability>).

In particular, we kindly request you to deposit your RNA-seq datasets to an appropriate database. The accession numbers, database, and the specific URLs (links) should be listed in a formal "Data availability" section (placed after Methods), following the example below:

"The RNA-seq datasets produced in this study are available in the following database:
Gene Expression Omnibus GSE46843 (<https://www.ncbi.nlm.nih.gov/geo/query/acc.cgi?acc=GSE46843>)"

*** All links should resolve to a page where the data can be accessed. ***

*** Please remember to provide in the Data availability section of your revised manuscript reviewer passwords if the datasets are not yet public. ***

*** The Data Availability Section is restricted to new primary data that are part of this study. In case you have no data that require deposition in a public database, please state so instead of referring to the database: "Our study includes no data deposited in public repositories." under the heading "Data availability". ***

4. Please check that the title and the abstract of the manuscript are brief, yet explicit, even to non-specialists. The length of the title should not exceed 100 characters, and the abstract should be a single paragraph not exceeding 175 words.

5. Please also note our reference format: <https://www.embopress.org/page/journal/14602075/authorguide#referencesformat>.

7. Please remember: digital image enhancement is acceptable practice, as long as it accurately represents the original data and conforms to community standards. If a figure has been subjected to significant electronic manipulation, this must be noted in the figure legend or in the "Materials and Methods" section. The editors reserve the right to request original versions of figures and the original images that were used to assemble the figure.

8. Our journal encourages inclusion of data citations in the reference list to directly cite datasets that were obtained from public databases. Data citations in the article text are distinct from normal bibliographical citations and should directly link to the database records from which the data can be accessed. In the main text, data citations are formatted as follows: "Data ref: Smith et al, 2001" or "Data ref: NCBI Sequence Read Archive PRJNA342805, 2017". In the Reference list, data citations must

be labeled with "[DATASET]". A data reference must provide the database name, accession number/identifiers, and a resolvable link to the landing page from which the data can be accessed at the end of the reference. Further instructions are available at: <https://www.embopress.org/page/journal/14602075/authorguide#referencesformat>.

9. We request authors to consider both actual and perceived competing interests. Please review our policy (<https://www.embopress.org/page/journal/14602075/authorguide#conflictsofinterest>) and update your competing interests statement if necessary. Please name this section 'Disclosure and competing interests statement' and place it after the Acknowledgements section.

10. Please note that all corresponding authors are required to provide an ORCID ID upon submission of a revised manuscript (<https://orcid.org/>). Please find instructions on how to link your ORCID ID to your account in our manuscript tracking system in our Author guidelines (<https://www.embopress.org/page/journal/14602075/authorguide#authorshipguidelines>).

11. We use CRediT to specify the contributions of each author in the journal submission system. CRediT replaces the author contribution section, which should be removed from the manuscript. Please use the free text box to provide more detailed descriptions. See also guide to authors: <https://www.embopress.org/page/journal/14602075/authorguide#authorshipguidelines>.

13. We would also welcome the submission of cover suggestions or motifs to be used by our Graphics Illustrator in designing a cover.

14. Please use the link below to submit your revision:
<https://emboj.msubmit.net/cgi-bin/main.plex>

Referee #1:

Summary: the paper by Watanabe et al., describes a mechanism of UNC13A regulation so far not described in the literature. They show that three ALS-related RBPs are responsible for maintaining its physiological levels. Indeed in absence of these proteins, UNC13A levels were reduced although through a mechanism independent from CE inclusion, which is characteristic of TDP-43 dysfunction. The study shows that these three RBPs are responsible also for maintaining REST levels and that REST up-regulation, caused by the absence of these RBPs, is responsible for UNC13A down-regulation. The work is well crafted and the results are solid. Nevertheless there are few major points that should be addressed for robustness and relevance and some minor technical point to address for clarity and completeness.

Major Points:

- The entire work is carried out in SH-SY5Y. Although this model is used for neuronal-related studies, it would be important to replicate some key results in IPS-neurons. Mechanisms of generegulation and CE inclusion are highly cell specific and thus the results produced in this paper are not informative about what happens in patients neurons. It would add relevance and importance the findings to add experiments in IPS-neurons.
- A follow up on my previous point is that throughout the paper the accent on synaptic function is very strong. Although no results are provided to show actual synaptic dysfunction following RBPs KO due to UNC13A reduction. This would be easy to measure in neurons either by measuring neuronal activity, or synaptic loading or the number of synapsis by microscopy or with synaptic isolation followed by WB. I would warrant the execution of at least one of these experiment to prove synaptic dysfunction.
- In figure 6 the result described in this figure is very correlative and it is not clear whether FUS phase separation has anything to do with REST regulation. The authors should try to modulate phase separation in the cells and study REST levels in their experimental conditions. Also, soluble and insoluble fractionation measures the solubility of a specific protein not its phase separation especially in live cells. The authors should show SGs formation by microscopy and the change in SGs assembly with the FUS construct examined in the paper.

Minor Ponts:

- Are TDP-43 levels affected by the KO of the others RBPs
- Do cells experience decreased viability when the RBPs are KO?
- Figure 2A: I suggest the use of at least another pair of primer to measure UNC13A-CE. Many papers are using different

primers for different CE inclusion. It adds completeness and robustness to measure CE with a second set of primers.

- Figure 5B, D: loading controls are missing

- Figure 7 B and C: since the data are quite spread, it would be informative to plot the data as REST TPM vs UNC13A TPM, to see if at higher levels of REST correspond low levels of UNC13A as done in Fig S6B with STMN.

- Figure 7 general comment: it is counterintuitive to see that the only neurons remained alive are the ones with REST. It is more suggestive of a protective mechanism rather than a toxic one. I understand the point the authors are trying to make but it would be correct to discuss this view in the discussion.

Referee #2:

In this study, the authors report that in cells with TDP-43, MATR3, FUS and hnRNPA1 KO, UNC13A expression level is greatly reduced. In contrast to TDP-43 whose loss causes aberrant cryptic exon splicing and degradation of UNC13A mRNA, loss of other three RNA-binding proteins decreases UNC13A transcription. The authors went on to show that the decreased UNC13A expression is due to transcriptional suppression of REST, whose mRNA becomes more stabilized in the absence of MATR3, FUS and hnRNPA1. Moreover, they claim that REST level is elevated in motor neurons of postmortem ALS patient tissues. Because all these 4 proteins are implicated in ALS pathogenesis and UNC13A has emerged as a key target of TDP-43 pathology, which occurs in 97% of ALS cases, these findings further highlight the importance of UNC13A and also suggest REST is another key player in ALS pathogenesis. However, the quality and rigor of a number of results are poor, thus the authors need to provide stronger experimental evidence to back up their claims.

1. Because it is already known that 97% of ALS cases exhibit TDP-43 pathology and UNC13A is a key target of TDP-43, the authors' claim that "a unified molecular basis for ALS has remained elusive" and their study identify UNC13A as a key common mechanism in ALS is overstated and should be toned down.
2. In several figures, the font size is too small for many words.
3. In Figure 1D and Figure 4H, UNC13A mRNA is almost completely gone in RBP KO cells. However, in Figure 2E, up to 30% is still expressed. Moreover, in Figure 2G, UNC13A transcription in hnRNPA1 KO cells is about 60% of that in WT cells. These results do not add up and this major discrepancy should be resolved.
4. In Figure 5, the authors did not provide a negative control, thus, the conclusion that MATR3, FUS and hnRNPA1 bind to REST mRNA is not well substantiated.
5. In Figure 6, the authors did structure-function analysis but these results do not provide any direct evidence that FUS phase separation underlies its regulation of REST mRNA, thus their claim is not well substantiated.
6. In Figure 7, Immunohistochemical staining for FUS in spinal motor neurons is not very convincing, it is difficult to tell the difference between controls and ALS cases.

Referee #3:

This manuscript by Watanabe et al. investigates the regulation of UNC13A by ALS-associated RNA-binding proteins (RBPs) - TDP-43, MATR3, FUS, and hnRNP A1 - through both convergent and divergent pathways. The study builds on previous findings that TDP-43 stabilizes UNC13A mRNA to prevent cryptic exon inclusion. The authors further propose that MATR3, FUS, and hnRNP A1 modulate REST mRNA levels, thereby implicating REST as a potential transcriptional regulator of UNC13A. This study highlights intriguing mechanisms through which ALS-associated RBPs may regulate UNC13A expression, a topic of significant interest in ALS pathology. While the findings are compelling, several aspects of the study require clarification and improvement. Key experiments appear preliminary and lack robust causative evidence to definitively establish the proposed RBP-REST-UNC13A axis as a contributor to ALS pathology. Additionally, many experiments lack statistical rigor, undermining confidence in the presented conclusions. Below, I provide specific comments to strengthen the manuscript:

Major point:

1. The manuscript does not clearly describe the phenotypes observed in the SH-SY5Y cell lines with individual RBP knockout (TDP-43, MATR3, FUS, and hnRNPA1). Are these cells exhibiting neurite degeneration or UNC13A-related impairments that are characteristic of ALS pathology? To strengthen the study, the authors should investigate whether the phenotypes induced by the knockouts of MATR3, FUS, and hnRNPA1 can be rescued through REST overexpression or modulation, via the binding to the promoters. This would provide critical insights into the functional relationships between these RBPs and REST.
2. The potential clinical relevance of this study lies in Figure 7. However, the REST immunostaining presented for ALS post-mortem tissues is unclear and of poor quality. The authors are encouraged to refine their staining protocol to improve clarity and reproducibility. Furthermore, while the manuscript references older data obtained using laser microdissection, the authors could strengthen their findings by leveraging recent and comprehensive datasets. Resources such as Answer ALS, which includes RNA-seq and proteomics data from over 1,000 ALS patients (spanning iPSC-derived motor neurons and post-mortem tissues), could provide valuable insights. Specifically, analyzing the correlation of the four RBPs with UNC13A and REST in these

datasets would help substantiate the proposed convergent and divergent pathways. If feasible, co-staining RBPs with REST in post-mortem tissue would further enhance the mechanistic understanding.

3. The selection of the 17 ALS genes in Figure 1C is unclear. To date, more than 65 ALS risk or mutated genes have been identified. The manuscript would benefit from explicitly detailing the criteria or rationale for selecting these specific genes, which would improve the transparency and relevance of the analysis. The selection of UNC13A appears quite arbitrary and biased.

4. Although the studied RBPs are implicated in ALS pathology, the overlap of differentially expressed genes (DEGs) across TDP-43, MATR3, FUS, and hnRNPA1 knockouts is limited, with only 267 shared genes reported in Figure 1B. The authors should discuss whether major DEG pathways or ALS-relevant genes emerge from the analysis. This discussion is essential to understand the broader implications of the limited functional convergence among these RBPs.

5. Several key experiments lack sufficient replicates and statistical validation. For instance, Figures 6D, 6E, and S4B require additional biological replicates (preferably triplicates) to ensure robust statistical analysis. Addressing this issue will improve the reliability and interpretability of the data.

6. The ChIP-qPCR data presented to confirm REST binding at the UNC13A promoter lack convincing evidence. The authors should enhance the experimental design, optimize the ChIP-qPCR protocol, and include additional biological replicates to achieve robust statistical validation. Improved data quality would significantly strengthen the study's conclusions regarding REST's direct role in UNC13A regulation.

7. The study emphasizes FUS-mediated regulation of REST mRNA but provides insufficient detail on why MATR3 and hnRNPA1 are not similarly examined. Have the authors attempted to validate interactions between these RBPs and REST mRNA? Additionally, the claim that "FUS phase separation underlies regulation of REST mRNA" is inadequately supported. To solidify this claim, the authors should include *in vitro* phase separation assays, such as LLPS formation studies, and fluorescence recovery after photobleaching (FRAP) experiments to demonstrate direct interactions between FUS and REST mRNA.

8. The manuscript suggests that the three RBPs selectively affect REST, and REST selectively affects UNC13A. This hypothesis appears biased without sufficient supporting data. Additional experiments or explanations are needed to justify this selective pathway.

Addressing these points would significantly strengthen the scientific rigor and impact of this manuscript. Incorporating additional experiments and leveraging large-scale datasets will enhance the mechanistic insights and relevance of the findings to ALS pathology.

Minor point:

1. The manuscript identifies REST as a known suppressor of neuronal gene expression and chooses to focus on REST for subsequent analysis of the REST/UNC13A pathway. However, other proteins such as YY1, JUN, GATA3, and SIN3A are also mentioned. Could the authors explore (or explain the rationale) potential links between these proteins and UNC13A? Addressing this would provide a more comprehensive perspective on potential regulatory networks.

2. The authors propose that TDP-43 stabilizes UNC13A splicing, while MATR3, FUS, and hnRNPA1 promote UNC13A transcription. However, Koike et al., 2023 (DOI: 10.1371/journal.pbio.3002228) report that hnRNPA1 also represses UNC13A cryptic exon splicing, and TDP-43 downregulation is critical for observing UNC13A cryptic RNA accumulation. This discrepancy requires careful reconciliation. The authors should address these differences and consider whether UNC13A cryptic exon regulation might vary across RBPs. Long-read sequencing technologies such as Nanopore could help resolve these ambiguities by providing a more complete view of UNC13A isoforms.

3. Could the authors clarify the binding or functional domains of MATR3 and hnRNPA1? Do these domains share similarities with those of FUS? Exploring this aspect would enhance the mechanistic understanding of these RBPs.

4. The layout of Figure 4G is confusing and requires reorganization for clarity. Improved annotations would also help readers interpret the data effectively.

5. Figures 5B-5D could be better organized to clearly depict the interactions of REST mRNA with MATR3, FUS, and hnRNPA1. Additionally, in Figure 5D, the manuscript implies an interaction between FUS and hnRNPA1. If such interactions are relevant, they should be clarified and supported with data.

6. Some figures lack statistical validation (e.g., Figures 1D and 4H), while others lack proper titles (e.g., Figure 1F). More specific and descriptive titles would enhance clarity. Statistical analyses should also be added where absent to ensure rigor.

7. There is a typographical error in Figure Legend 1D, where "UNC13A" is misspelled. Please correct this.

8. The dashed lines in Figure S1C and S1D are not explained.

9. Figures 2D and 2E could be combined to facilitate a direct comparison of UNC13A expression across different cell lines following CHX treatment. Figure 2H does not seem to have illustrate the conclusion well.

10. Figure 7D cartoon is probably not needed as it is data from other people.

11. The distinction between mRNA and nascent RNA in Figure 2 is unclear. The Y-axis annotations should be revised to clearly indicate the measured parameters.

12. The heatmap lacks a score legend or annotation. Adding this information would make the data more accessible to readers.

13. The manuscript mentions the noncanonical NRSE motif but does not describe its function. A brief explanation would provide valuable context for readers unfamiliar with this element.

14. The quantification in Figure 7H is not well-described. Providing details about the analysis methods and statistical validation would strengthen the conclusions drawn from this figure.

Response to Referee #1

Major Points:

1. *The entire work is carried out in SH-SY5Y. Although this model is used for neuronal-related studies, it would be important to replicate some key results in IPS-neurons. Mechanisms of gene regulation and CE inclusion are highly cell specific and thus the results produced in this paper are not informative about what happens in patients neurons. It would add relevance and importance the findings to add experiments in IPS-neurons.*

[**Response**] We appreciate the referee's insightful comment. We performed additional experiments using motor neurons differentiated from iPSCs carrying an ALS-associated *FUS* mutation (*FUS* P525L/+). In these *FUS* P525L/+ iMNs, we observed partial cytoplasmic mislocalization of FUS, suggesting a loss of its nuclear function (new **Appendix Fig. S7A**). Consistent with our hypothesis, *REST* mRNA levels were significantly elevated, while *UNC13A* mRNA levels were reduced in *FUS* P525L/+ iMNs compared to WT cells (new **Fig. 7A,B**). At the protein level, we also observed an increase in *REST* expression and a decrease in *UNC13A* expression in *FUS* P525L/+ iMNs, further supporting our findings (new **Appendix Fig. S7B**). These results align with our findings in SH-SY5Y cells, demonstrating that *REST* upregulation and subsequent *UNC13A* repression occur in a disease-relevant neuronal model.

Additionally, we analyzed the Answer ALS transcriptomic dataset, which includes iPSC-derived motor neurons from ALS and non-ALS individuals. While this dataset did not show significant differences in *REST* and *UNC13A* expression levels between the groups, it revealed a conserved inverse correlation between them, supporting the general relevance of *REST*-mediated *UNC13A* repression in motor neurons (new Fig. EV5A, B). (As these results are more directly relevant to the concerns raised by Referee #3, we have included the corresponding figure in that section of our response.)

We believe these additional data further strengthen the relevance of our findings while directly addressing the referee's request.

new **Appendix Fig. S7A**

new Fig. 7A,B

new Appendix Fig. S7B

2. A follow up on my previous point is that throughout the paper the accent on synaptic function is very strong. Although no results are provided to show actual synaptic dysfunction following RBPs KO due to UNC13A reduction. This would be easy to measure in neurons either by measuring neuronal activity, or synaptic loading or the number of synapses by microscopy or with synaptic isolation followed by WB. I would warrant the execution of at least one of these experiment to prove synaptic dysfunction.

[Response] We thank the referee for the insightful comment. Previous studies have demonstrated the role of *UNC13A* in synaptic function. In *UNC13A* (*Munc13-1*) knockout mice, synaptic structure remains intact, but postsynaptic currents are reduced (Augustin I, Nature, 1999). Similarly, in iPSC-derived neurons with TDP-43 knockdown, postsynaptic currents were reduced but restored by treatment with an antisense oligonucleotide (ASO) targeting *UNC13A* cryptic exon inclusion (Keuss *et al.*, bioRxiv, 2024). These findings suggest that detecting the synaptic phenotype caused by UNC13A reduction in our RBP-KO

cells would require measuring postsynaptic currents. However, SH-SY5Y cells inherently lack synaptic structures. To overcome this limitation, we attempted to induce synapse formation by differentiating the cells using retinoic acid (RA) and brain-derived neurotrophic factor (BDNF) (Forster JI. *et al.*, J Biomol Screen, 2016). Following RA-BDNF differentiation, we performed synaptosome isolation using a synaptosome isolation kit (Invent Biotechnologies, Inc.), as the referee suggested, to determine whether synapses had formed. However, in RBP-KO cell lines, synaptosomes failed to form a pellet from the soluble lysate fraction, suggesting unsuccessful synapse formation. Western blot analysis of the synaptosome-containing soluble fraction further confirmed a reduction in synaptophysin, a presynaptic marker, in MATR3-KO, FUS-KO, and hnRNPA1-KO cells. Moreover, β 3-tubulin, a neuronal marker, was reduced in TDP-43-KO and FUS-KO cells, indicating impaired neuronal differentiation (**Figure 1 for Referee #1**).

Figure 1 for Referee #1

These findings underscore the limitations of RBP-KO SH-SY5Y cell lines as models for analyzing UNC13A-related synaptic phenotypes, likely due to the widespread downregulation of nervous system process-related genes (Figure 1B). However, the primary objective of our study was to elucidate the molecular mechanisms through which ALS-associated RBPs regulate the expression of *UNC13A* and other synaptic genes, rather than emphasizing their direct effects on synaptic function. To ensure the accuracy of our claims, we have revised the Abstract and Introduction to clarify this point (Page 2, Lines 12-14 and Page 4, Lines 10-12, respectively). Additionally, we removed the following text from the Results section because this statement might give the impression that our study focuses on the influence of RBPs on synaptic phenotype.

Removed text: “*Synaptic dysfunction is thought to contribute to the early stages of ALS (Nishimura & Arias, 2021; Vinsant et al., 2013). Given that UNC13A plays a key role in synaptic vesicle recycling and neurotransmitter release (Willemse et al., 2023), we hypothesized that dysfunction or nuclear depletion of ALS-associated RBPs might lead to a loss of UNC13A mRNA and thereby precipitate synaptic dysfunction.*”

3a. In figure 6 the result described in this figure is very correlative and it is not clear whether FUS phase separation has anything to do with REST regulation. The authors should try to modulate phase separation in the cells and study REST levels in their experimental conditions.

[Response] We thank the referee's insightful suggestion. As suggested by the Referee, we disrupted LLPS in cells using 1,6-hexanediol and assessed REST expression levels under different conditions (no treatment, 1% for 30 min, 1% for 10 min, and 5% for 10 min). Our results indicate that LLPS disruption did not lead to significant changes in REST expression levels (Figure 2 for Referee #1).

Figure 2 for Referee #1

One possible explanation for this result is that 1,6-hexanediol is not a specific inhibitor of FUS LLPS but rather affects multiple LLPS-related pathways that collectively influence REST mRNA stability, potentially masking any specific effect of FUS LLPS on REST regulation. However, we could not obtain direct evidence supporting the involvement of LLPS in REST regulation. Therefore, we have modified the title and conclusion of Figure 6 to more cautiously suggest that FUS phase separation potentially contribute to REST mRNA regulation, rather than stating it as a definitive mechanism (Page 10, Lines 13-14). We believe this refined interpretation more accurately represents our findings while maintaining the significance of our study.

3b. Also, soluble and insoluble fractionation measures the solubility of a specific protein not its phase separation especially in live cells. The authors should show SGs formation by microscopy and the change in SGs assembly with the FUS construct examined in the paper.

[Response] In the previous Figure 6G, we performed soluble and insoluble fractionation

assays primarily to confirm the previous findings, thereby validating the biochemical characteristics of the 27YS mutants. To clarify this in the revised manuscript, we have included a statement indicating that our results are “*consistent with previous findings that phase separation precedes the formation of insoluble aggregates (Qamar et al., 2018; Reber et al., 2021)*” (Page 10, Lines 7-8).

Additionally, to address the referee’s concern, we established cell lines expressing WT FUS and the FUS-27YS mutants. To assess the extent of FUS localization to SGs, we treated the cells with arsenite and evaluated colocalization with the SG marker G3BP. Cytoplasmic FUS granules were observed to co-localize with G3BP-positive SGs. The number of cells with G3BP-colocalized cytoplasmic FUS granules formed by the FUS-27YS mutants were significantly lower than that observed for WT FUS. We have moved previous Figure 6G to Appendix Fig. 6A and replaced it with the results of the quantification data of SG formation assay (**new Fig. 6G**). The representative image of SG formation assay is presented in **new Appendix Fig. S6B**.

new Fig. 6G (left), **Appendix S6B** (right)

Minor Points:

1. Are TDP-43 levels affected by the KO of the others RBPs

[**Response**] TDP-43 protein levels in RBP-KO cells were assessed and are shown in Figure 1E, where TDP-43 remains present in all conditions. To clarify this point explicitly in the main text, we have revised the manuscript (Page 5, Lines 5-7).

2. Do cells experience decreased viability when the RBPs are KO?

[**Response**] When establishing RBP-KO cell lines, only TDP-43-KO cells exhibited a high incidence of dead cells during culture. Examination of the expression of cleaved PARP, an apoptosis marker, revealed strong expression in TDP-43-KO cells (new **Fig. EV1B**).

new Fig. EV1B

3. *Figure 2A: I suggest the use of at least another pair of primer to measure UNCI3A-CE. Many papers are using different primers for different CE inclusion. It adds completeness and robustness to measure CE with a second set of primers.*

[**Response**] To enhance robustness, we performed qPCR using an additional primer set referencing the CE primer sequences from Koike *et al.*, (2023) (new **Appendix Fig. S3A**). We found that this primer set may amplify a subtle amount of pre-mRNA and genomic contamination, so we have illustrated its position within the *UNCI3A* sequence. Additionally, we have updated the Figure 2A legend to specify that the qPCR was performed using the CE primers from Ma *et al.* (2022) and have illustrated their positions in the Figure 2A.

new Appendix Fig. S3A (left), **new Fig. 2A** (right)

4. Figure 5B, D: loading controls are missing

[Response] To demonstrate that an appropriate amount of IP product was loaded in all lanes where the IP product was applied in Figures 5B and 5D, we expanded the cropping area of the MATR3 and FUS blot images to include the denatured IgG-derived bands, which serve as loading controls for the IP products (new Fig. 5B, D).

5. Figure 7 B and C: since the data are quite spread, it would be informative to plot the data as *REST* TPM vs *UNC13* TPM, to see if at higher levels of *rest* correspond low levels of *UNC13A* as done in Fig S6B with *STMN*.

[Response] We performed the correlation analysis using sALS datasets; however, we did not observe a significant correlation between *REST* and *UNC13A* expression levels (correlation coefficient = 0.069, $p = 0.822$) (**Figure 3 for Referee #1**). As discussed in Figure 7C, although the dataset used in this analysis is enriched for motor neurons, it likely includes a mixture of cell types, potentially masking a significant association.

Figure 3 for Referee #1

6. *Figure 7 general comment: it is counterintuitive to see that the only neurons remained alive are the ones with REST. it is more suggestive of a protective mechanism rather than a toxic one. I understand the point the author are trying to make but it would be correct to discuss this view in the discussion.*

[Response] We appreciate the referee's insightful comment. We have incorporated a discussion on the neuroprotective role of REST based on the suggested perspective (Page 13, Lines 29-38).

Response to Referee #2

1. Because it is already known that 97% of ALS cases exhibit TDP-43 pathology and UNC13A is a key target of TDP-43, the authors' claim that "a unified molecular basis for ALS has remained elusive" and their study identify UNC13A as a key common mechanism in ALS is overstated and should be toned down.

[Response] We appreciate the referee's comments. To avoid any potential overstatement and ensure that our conclusions are accurately conveyed, we have revised the Abstract and Introduction (Page 2, Lines 3-4 and Page 3, Lines 6-8). We would also like to clarify that our study does not propose UNC13A downregulation as a universal mechanism of ALS pathogenesis. Rather, our findings demonstrate that multiple ALS-associated RBPs converge on the regulation of UNC13A expression, representing a significant step toward a more comprehensive understanding of ALS pathology. To clarify this point, we modified the last paragraph of Introduction section (Page 4, Lines 10-12). We appreciate the opportunity to clarify this point and thank the referee for the constructive feedback.

2. In several figures, the font size is too small for many words.

[Response] We appreciate the referee's feedback. We have increased the font size for several key terms that were previously too small to read clearly.

3. In Figure 1D and Figure 4H, UNC13A mRNA is almost completely gone in RBP KO cells. However, in Figure 2E, up to 30% is still expressed. Moreover, in Figure 2G, UNC13A transcription in hnRNPA1 KO cells is about 60% of that in WT cells. These results do not add up and this major discrepancy should be resolved.

[Response] We appreciate the referee's thoughtful comments. The differences in *UNC13A* mRNA levels observed in Figures 1D, 2E, and 2G can be explained by differences in experimental conditions and the biological processes being measured.

First, the variation between Figures 1D and 2E is due to the presence or absence of DMSO. DMSO is known to influence gene expression through epigenetic and transcriptional mechanisms (Iwatani M et al., 2006; Xing L et al., 2005; Nishimura M et al., 2008; Thaler R et al., 2012). Because CHX is dissolved in DMSO, we included a DMSO-treated control in Figure 2E to account for its potential effects. This ensures that any changes in *UNC13A* mRNA levels caused by CHX treatment reflect NMD inhibition rather than solvent effects. Given DMSO's known impact on gene expression, the differences in mRNA levels between Figures 1D and 2E are expected and do not indicate inconsistencies in our findings.

Second, the difference between Figs.1D and 2G seem to arise from the fundamental distinction between steady-state mRNA levels and nascent transcription rates. Steady-state

mRNA levels reflect the cumulative balance of transcription and degradation over time, whereas nascent transcription measurements capture RNA synthesis within a defined time window (Chappleboim A et al., *Nucleic Acids Res.*, 2022; Bryll AR et al., *Transcription*, 2023). Considering that our NMD inhibition assay (Figure 2E) showed no significant differences in degradation between these conditions, the near-complete loss of *UNC13A* mRNA in Figure 1D is likely due to prolonged transcriptional suppression over time, rather than an immediate reduction in transcription rate at the time point measured in Figure 2G. Even if nascent transcription remains partially active within the one-hour 4-EU labeling window, sustained transcriptional suppression combined with ongoing degradation would lead to a progressive decline in steady-state mRNA levels. Therefore, Figs. 1D and 2G are not contradictory but instead capture different aspects of mRNA regulation at different timescales. We hope this explanation clarifies the referee's concerns.

4. In Figure 5, the authors did not provide a negative control, thus, the conclusion that MATR3, FUS and hnRNPA1 bind to REST mRNA is not well substantiated.

[Response] We thank the referee for the insightful comment. Our study proposes that TDP-43 does not affect *REST* mRNA expression levels, whereas the other three RBPs bind to *REST* mRNA and regulate its stability. Therefore, we considered TDP-43 as a suitable negative control. Consistent with previous reports, immunoprecipitated TDP-43 was bound to its own mRNA (*TARDBP*) but did not associate with *REST* mRNA (new **Fig. 5B, C**). This result further reinforces our hypothesis that MATR3, FUS, and hnRNPA1 specifically bind to *REST* mRNA to regulate its stability.

new Fig. 5B, C

5. In Figure 6, the authors did structure-function analysis but these results do not provide any direct evidence that FUS phase separation underlies its regulation of REST mRNA, thus their claim is not well substituted.

[Response] We appreciate the referee's comment. As we responded to Referee #1, we were unable to establish a direct causal relationship between FUS phase separation and the

upregulation of *REST* mRNA, but our findings suggest a critical role for FUS IDRs in *REST* mRNA regulation. To reflect this, we have carefully adjusted the tone of our statements to acknowledge the limitations while maintaining the significance of our observations (Page 10, Lines 13-14).

6. In Figure 7, Immunohistochemical staining for FUS in spinal motor neurons is not very convincing, it is difficult to tell the difference between controls and ALS cases.

[Response] We appreciate the referee's comment. We have re-stained FUS and obtained clearer images that better distinguish controls from ALS cases (new **Fig. 7D, E**).

new **Fig. 7D** (left), **E** (right)

Response to Referee #3

Major point:

1a. The manuscript does not clearly describe the phenotypes observed in the SH-SY5Y cell lines with individual RBP knockout (TDP-43, MATR3, FUS, and hnRNPA1). Are these cells exhibiting neurite degeneration or UNC13A-related impairments that are characteristic of ALS pathology?

[**Response**] Compared to WT cells, RBP-KO cells tended to exhibit a rounded morphology and lacked cytoplasmic processes (new **Fig. EV1A**). In the manuscript, we intentionally avoided the term “neurites” and instead used “cytoplasmic processes” because SH-SY5Y cells are generally considered immature and retain their proliferative potential, meaning they are not strictly classified as neurons. Additionally, due to the limitations of our RBP-KO cell models, we were unable to assess UNC13A-related synaptic phenotypes. Please refer to our response to Referee #1 for further details.

new Fig. EV1A

(The yellow arrowheads indicate the cells with cytoplasmic processes.)

1b. To strengthen the study, the authors should investigate whether the phenotypes induced by the knockouts of MATR3, FUS, and hnRNPA1 can be rescued through REST overexpression or modulation, via the binding to the promoters. This would provide critical insights into the functional relationships between these RBPs and REST.

[**Response**] We appreciate the referee for the insightful comment. GO analysis of potential target genes of REST, which was upregulated in RBP-KO cells (Figure S4D, S4E), revealed an enrichment of genes related to “neuron projection.” As the referee pointed out, to

investigate the effect of REST upregulation in RBP-KO cells, we performed siRNA-mediated knockdown of REST. As a result, the proportion of rounded cells decreased, and more cells exhibited cytoplasmic processes (new **Fig. EV2A, B**). These findings indicate that overexpressed REST contributes to the morphological phenotype observed in RBP-KO cells.
new Fig. EV2A (left), **B** (right)

2a. The potential clinical relevance of this study lies in Figure 7. However, the REST immunostaining presented for ALS post-mortem tissues is unclear and of poor quality. The authors are encouraged to refine their staining protocol to improve clarity and reproducibility.

[Response] We appreciate the referee for the valuable comment. To improve the clarity and reproducibility of REST immunostaining, we optimized the antibody concentration. While 3 $\mu\text{g}/\text{mL}$ caused weak background signals with the negative control IgG, increasing the REST antibody from 1 $\mu\text{g}/\text{mL}$ to 2 $\mu\text{g}/\text{mL}$ provided clear nuclear staining with minimal background. We have updated Figure 7G with representative images, now including REST staining from three sporadic ALS cases and an equal number of controls (new **Fig. 7G, H, J**). Based on these optimized conditions, we also updated the quantitative data in new **Figure 7I**. We believe these improvements enhance the clarity and reproducibility of our findings.
new Fig. 7G (left), **H** (right)

new Fig. 7J (left), I (right)

2b. Furthermore, while the manuscript references older data obtained using laser microdissection, the authors could strengthen their findings by leveraging recent and comprehensive datasets. Resources such as Answer ALS, which includes RNA-seq and proteomics data from over 1,000 ALS patients (spanning iPSC-derived motor neurons and post-mortem tissues), could provide valuable insights. Specifically, analyzing the correlation of the four RBPs with *UNC13A* and *REST* in these datasets would help substantiate the proposed convergent and divergent pathways.

[**Response**] We appreciate the referee's insightful comment. As suggested by the referee, we reviewed the Answer ALS dataset and found that, in terms of transcriptomics and proteomics data, it primarily consists of iPSC-derived motor neurons from non-ALS individuals and ALS patients, rather than postmortem tissue. While platforms like Target ALS provide gene

expression data from postmortem ALS tissues, our study includes immunohistochemical analysis of autopsy-derived spinal cord tissues, demonstrating REST expression in motor neurons. If the referee's concern is whether REST is overexpressed in the spinal cord of ALS patients, we believe our analysis sufficiently addresses this issue.

The referee suggested examining the correlation between RBP expression and REST/UNC13A expression. In this regard, ALS pathology is primarily characterized by nuclear depletion and cytoplasmic mislocalization of RBPs rather than changes in overall transcript or protein levels. Therefore, transcriptomic correlation analyses may not fully capture RBP dysfunction in ALS. To illustrate this limitation, we analyzed TDP-43 and *STMN2* mRNA levels in the Answer ALS dataset. Despite prior evidence linking TDP-43 mislocalization to *STMN2* downregulation (Prudencio et al., 2020, and others), no significant correlation was observed between their mRNA levels, reinforcing the inadequacy of gene expression correlation analyses for capturing ALS-specific RBP dysfunction (**Figure 1 for Referee #3**).

Figure 1 for Referee #3

Taking this limitation into account, we further analyzed the Answer ALS dataset. Proteomic data showed that REST and UNC13A were below the detection threshold, and RNA-seq data revealed no expected correlation between the three RBPs mRNA levels and *REST* expression in iPSC-derived motor neurons from control or ALS samples (**Figure 2 for Referee #3**). These findings align with the established understanding that ALS is driven by RBP mislocalization and functional impairment rather than changes in mRNA levels.

Figure 2 for Referee #3

Notably, while our initial analysis of the Answer ALS dataset did not reveal a significant correlation between RBP mRNA levels and *REST* or *UNC13A* expression, we identified an inverse relationship between *REST* and *UNC13A* expression in both ALS and non-ALS groups (**new Fig. EV5A,B**). This suggests that REST-mediated repression of *UNC13A* is a conserved mechanism in motor neurons, independent of ALS status. Although

REST and *UNC13A* expression levels did not differ significantly between ALS and non-ALS groups (new Fig. EV5C,D), we speculate that ALS MNs in the Answer ALS transcriptomics data may not fully capture disease-associated RBP dysfunction. This includes the cytoplasmic mislocalization of key RBPs such as FUS, MATR3, and hnRNPA1, which regulate *REST* and *UNC13A* expression, as well as TDP-43, which influences *UNC13A* expression. This limitation may explain the absence of *REST* upregulation and *UNC13A* downregulation in our analysis.

We appreciate the reviewer's insightful suggestion to explore the Answer ALS dataset, which provided valuable perspectives on *REST* function in motor neurons.

new Fig.EV5A-D

2c. If feasible, co-staining RBPs with *REST* in post-mortem tissue would further enhance the mechanistic understanding.

[Response] We appreciate the referee’s insightful comment. We acknowledge that co-staining RBPs with REST in post-mortem tissue could provide further mechanistic insight. However, extensive condition optimization is not feasible due to the age and limited availability of the FFPE samples. Moreover, fluorescence-based detection, required for co-staining, is significantly hindered by the strong autofluorescence inherent to aged FFPE tissues, which substantially impairs signal specificity and makes reliable co-staining results unattainable.

3. The selection of the 17 ALS genes in Figure 1C is unclear. To date, more than 65 ALS risk or mutated genes have been identified. The manuscript would benefit from explicitly detailing the criteria or rationale for selecting these specific genes, which would improve the transparency and relevance of the analysis. The selection of UNC13A appears quite arbitrary and biased.

[Response] As stated in the figure legend, we initially used the 17 ALS-related genes classified as “Definitive” in the ALS Online Database (ALSoD) which represent the strongest genetic evidence for ALS onset. In response to the referee’s suggestion, we expanded our selection criteria to also include genes categorized as “Clinical Modifier,” “Strong Evidence,” and “Moderate Evidence,” resulting in a total of 48 genes (new **Fig. 1C**). To ensure transparency, we have provided the full list of these genes, categorized accordingly, in new **Appendix Table S1**. Although the ALSoD gene list does not encompass all known ALS-associated genes, we established clear selection criteria to ensure transparency in our analysis. Notably, *SETX*, *SIGMAR1*, and *HNRNPA2B1* have been reported as ALS-associated but are not included in the ALSoD list. The detailed results of our k-means clustering analysis of RNA-seq data from RBP-KO cell lines are provided in **Table EV1**, enabling readers to verify that these ALS-related genes are not among the commonly downregulated genes across the four RBP-KO cell lines. In this regard, our analysis remains transparent and reproducible.

new Fig. 1C

4. Although the studied RBPs are implicated in ALS pathology, the overlap of differentially expressed genes (DEGs) across TDP-43, MATR3, FUS, and hnRNPA1 knockouts is limited, with only 267 shared genes reported in Figure 1B. The authors should discuss whether major DEG pathways or ALS-relevant genes emerge from the analysis. This discussion is essential

to understand the broader implications of the limited functional convergence among these RBPs.

[Response] We thank the referee's insightful comment. We have now explicitly addressed the relevance of the GO analysis results (Fig.1B) to ALS pathogenesis by revising the explanation of the DEG pathway analysis in Figure 1B (Page 4, Lines 29-31). This revision highlights the alignment between our RNA-seq analysis findings and well-established ALS pathophysiology, reinforcing the biological significance of the downregulated synaptic genes in the four RBP-KO cell lines. We believe this addition strengthens the manuscript by explicitly linking the functional enrichment results to ALS-related processes.

5. Several key experiments lack sufficient replicates and statistical validation. For instance, Figures 6D, 6E, and S4B require additional biological replicates (preferably triplicates) to ensure robust statistical analysis. Addressing this issue will improve the reliability and interpretability of the data.

[Response] We have performed additional experiments and increased the number of replicates for several Figures, including 6D, 6E and S4B.

6. The ChIP-qPCR data presented to confirm REST binding at the UNC13A promoter lack convincing evidence. The authors should enhance the experimental design, optimize the ChIP-qPCR protocol, and include additional biological replicates to achieve robust statistical validation. Improved data quality would significantly strengthen the study's conclusions regarding REST's direct role in UNC13A regulation.

[Response] We have optimized the ChIP-qPCR protocol by using a different REST antibody and adjusting its concentration. This optimization resulted in a reduced % Input signal in the negative control region relative to the *UNC13A* promoter region, thereby improving specificity. Additionally, we performed ChIP using an IgG control antibody to assess potential non-specific enrichment. The IgG ChIP showed no enrichment bias across all tested genomic regions. In contrast, ChIP with the REST antibody demonstrated a significantly higher % Input at the *UNC13A* promoter compared to the negative control region. Moreover, there was no significant difference in % Input between the *UNC13A* promoter and the positive control region, the SynI promoter (**new Appendix Fig.S4B**).

new Appendix Fig. S4B

7a. The study emphasizes *FUS*-mediated regulation of *REST* mRNA but provides insufficient detail on why *MATR3* and *hnRNPA1* are not similarly examined. Have the authors attempted to validate interactions between these RBPs and *REST* mRNA?

[Response] As stated in the manuscript, we focused on *FUS* among the three RBPs influencing *REST* expression because its depletion had a greater impact on *UNC13A* transcription (Figures 2G, 4F, S3E) and *FUS* mutations are more frequently linked to ALS (Akiyama et al., 2016; Renton et al., 2014). Additionally, our access to FFPE spinal cord samples from *FUS*-ALS patients (and also *FUS*-mutated iPSCs in the revised manuscript) provided a strong foundation for disease-relevant analyses, further supporting our decision to prioritize *FUS* in this study.

7b. Additionally, the claim that "*FUS* phase separation underlies regulation of *REST* mRNA" is inadequately supported. To solidify this claim, the authors should include *in vitro* phase separation assays, such as LLPS formation studies, and fluorescence recovery after photobleaching (FRAP) experiments to demonstrate direct interactions between *FUS* and *REST* mRNA.

[Response] We appreciate the referee for the insightful comment. As we addressed in our responses to referees #1 and #2, we were unable to establish a direct causal relationship between *FUS* LLPS and the upregulation of *REST* mRNA, even with LLPS inhibition, likely due to experimental limitations. The *in vitro* phase separation assay suggested by the referee also presents challenges, as *REST* mRNA exceeds 3 kb, making its full-length synthesis and proper folding difficult. Moreover, given the complexity of RNA-protein interactions in LLPS, truncated RNA fragments may not accurately reflect physiological conditions. In light of these constraints, we have refined the manuscript as described in our response to referee #1 (Page 10, Lines 13-14).

8. The manuscript suggests that the three RBPs selectively affect *REST*, and *REST* selectively

affects UNC13A. This hypothesis appears biased without sufficient supporting data. Additional experiments or explanations are needed to justify this selective pathway.

[Response] We appreciate the referee's comment and the opportunity to clarify our findings. Our study does not claim that MATR3, FUS, and hnRNPA1 selectively regulate *REST*, nor that *REST* exclusively regulates *UNC13A*. Instead, our data suggest that these RBPs broadly impact multiple genes, with *REST* being one key mediator among many. As shown in Figure S3D, the loss of these three RBPs leads to the common downregulation of 251 genes, including *REST* target genes. Throughout the manuscript, we focused on the RBPs-*REST*-*UNC13A* axis because *UNC13A* is a key *REST* target gene implicated in ALS. Its downregulation has been independently linked to ALS through TDP-43-related mechanisms, and SNPs in *UNC13A*, which are known genetic risk factors for ALS, have been associated with reduced *UNC13A* expression.

Regarding the specificity of *REST*-mediated regulation of *UNC13A*, our study does not suggest that *REST* acts exclusively on *UNC13A*. As shown in **Figure S4D** and **Table EV2** (Previous Supplementary Table S4), we identified 38 potential *REST* target genes that were significantly downregulated in RBP-KO cells, supporting the broader impact of *REST* dysregulation in ALS. This evidence clearly demonstrates that *REST*-mediated transcriptional dysregulation extends beyond *UNC13A*, directly addressing the concern that we present an overly selective pathway. Additionally, we have discussed the potential pathological role of *REST* in ALS through *UNC13A*-independent mechanisms, highlighting its broader involvement in disease pathogenesis. Therefore, while we emphasize the importance of *UNC13A* in this pathway, this does not mean that *UNC13A* is the sole target of *REST*.

Although the original manuscript discussed the possibility that RBP dysfunction may contribute to ALS pathogenesis through *REST* in a *UNC13A*-independent pathway, our explanation may have been ambiguous. To better address the reviewer's concerns, we have now explicitly clarified the potential involvement of *REST* in ALS pathogenesis beyond its regulation of *UNC13A* by adding a more detailed explanation to the Discussion (Page 13, Lines 1-9). We are confident that these clarifications fully resolve the reviewer's concerns and further substantiate the robustness of our conclusions.

Minor point:

1. The manuscript identifies REST as a known suppressor of neuronal gene expression and chooses to focus on REST for subsequent analysis of the REST/UNC13A pathway. However, other proteins such as YY1, JUN, GATA3, and SIN3A are also mentioned. Could the authors explore (or explain the rationale) potential links between these proteins and UNC13A? Addressing this would provide a more comprehensive perspective on potential regulatory networks.

[Response] We thank the referee for the insightful comment. We conducted an additional literature search to explore direct connections between REST, UNC13A, and these transcription factors but found no evidence linking REST/UNC13A with YY1, JUN, or GATA3. In contrast, SIN3A is a well-established core component of the SIN3/HDAC complex, which is recruited by REST to silence neuronal genes. Based on this established interaction, we have now explicitly highlighted SIN3A's potential role in the REST/UNC13A regulatory (Page 6, Lines 30-33).

2. The authors propose that TDP-43 stabilizes UNC13A splicing, while MATR3, FUS, and hnRNPA1 promote UNC13A transcription. However, Koike et al., 2023 (DOI: 10.1371/journal.pbio.3002228) report that hnRNPA1 also represses UNC13A cryptic exon splicing, and TDP-43 downregulation is critical for observing UNC13A cryptic RNA accumulation. This discrepancy requires careful reconciliation. The authors should address these differences and consider whether UNC13A cryptic exon regulation might vary across RBPs. Long-read sequencing technologies such as Nanopore could help resolve these ambiguities by providing a more complete view of UNC13A isoforms.

[Response] Koike *et al.* reported that knockdown of hnRNPA1 in wild-type HeLa cells did not induce the inclusion of the *UNC13A* cryptic exon (CE) (Koike *et al.*, 2023, Fig. 5A). This observation is consistent with our findings shown in Figure 2A. Furthermore, their data indicate that hnRNPA1 suppresses the amount of *UNC13A* mRNA containing CE, but only under TDP-43 knockout conditions (Koike *et al.*, 2023, Fig. 5C). Additionally, their study did not assess the potential impact of hnRNPs on the stability of *UNC13A*-CE mRNA, leaving open the possibility that cryptic exon regulation may involve post-transcriptional mechanisms in the TDP-43-KO cells. Our study focuses on the physiological role of hnRNPA1 and does not examine its overexpression in the absence of TDP-43. Therefore, the apparent discrepancy raised by the referee likely reflects differences in the specific experimental contexts of each study.

3. Could the authors clarify the binding or functional domains of MATR3 and hnRNPA1? Do these domains share similarities with those of FUS? Exploring this aspect would enhance the mechanistic understanding of these RBPs.

[Response] We appreciated the insightful comment. As discussed in the Discussion, MATR3, hnRNPA1, and FUS all contain intrinsically disordered regions (IDRs). hnRNPA1, like FUS, possesses a prion-like domain (PrLD), while MATR3 lacks a distinct PrLD but contains an IDR-rich N-terminal region. In the revised manuscript, we have further clarified the distribution of ALS-associated mutations within these RBPs. Most ALS-associated mutations in FUS, MATR3, and hnRNPA1 are concentrated within IDRs (**new Fig. 6A and Fig. EV4A, B**), raising the possibility that these RBPs contribute to REST regulation through

IDR-dependent mechanisms.

new Fig. 6A

new EV. 4A (left), B (right)

4. The layout of Figure 4G is confusing and requires reorganization for clarity. Improved annotations would also help readers interpret the data effectively.

[Response] We appreciate the referee's comment. This experiment represents the most critical dataset in our series of promoter assays. In response to both the suggestion regarding Figure 4G and a major comment requesting additional replicates, we have revised the promoter assay protocol and repeated the experiment with $n = 5$. To enhance clarity, we have reorganized the figure layout by plotting the Δ R/Canonical ratio on the y-axis (previously on the x-axis) and labeling the x-axis with cell line names. Additionally, we have updated the statistical annotations to display p -values comparing the Δ R/Canonical activity ratio between

WT and each RBP-KO cell line, using asterisks to indicate statistical significance.

new Fig. 4G

5. Figures 5B-5D could be better organized to clearly depict the interactions of *REST* mRNA with *MATR3*, *FUS*, and *hnRNPA1*.

[Response] We have rearranged the layout of Figures 5B-5D to enhance visual clarity and better illustrate the interactions between *REST* mRNA and each RBP.

6. Additionally, in Figure 5D, the manuscript implies an interaction between *FUS* and *hnRNPA1*. If such interactions are relevant, they should be clarified and supported with data.

[Response] We appreciate the referee for the insightful comment. In the RIP experiments, dithiothreitol was included to prevent RBP oxidation, maintain RNA integrity, and reduce non-specific interactions by disrupting disulfide bonds. Because the use of dithiothreitol is thus not suitable for detecting protein-protein interactions, the interaction between *FUS* and *hnRNPA1* was not observed in Figure 5D. To further investigate whether *FUS*, *hnRNPA1*, and *MATR3* interact with each other, as the referee suggested, we performed IP under neutral conditions and confirmed interactions among these RBPs (new **Fig. EV4A, B**). These findings suggest that these proteins may cooperatively regulate the stability of *REST* mRNA. We have revised the manuscript based on this perspective.

new EV4A (left), B (right)

7. Some figures lack statistical validation (e.g., Figures 1D and 4H), while others lack proper

titles (e.g., Figure 1F). More specific and descriptive titles would enhance clarity. Statistical analyses should also be added where absent to ensure rigor.

[Response] We have now added statistical validation to several Figures including Figures 1D and 4H. Additionally, we have revised the titles of Figure 1F and other relevant figures.

8. There is a typographical error in Figure Legend 1D, where "UNC13A" is misspelled. Please correct this.

[Response] We appreciate the referee's attention to detail. The typographical error in Figure Legend 1D has been corrected.

9. The dashed lines in Figure S1C and S1D are not explained.

[Response] We added the following text into the legends of these Figures. "Dashed lines indicate where the original blot image was spliced to juxtapose lanes that were non-contiguous."

10. Figures 2D and 2E could be combined to facilitate a direct comparison of UNC13A expression across different cell lines following CHX treatment.

[Response] We thank the referee for the valuable feedback. While we appreciate the suggestion to combine Figures 2D and 2E, we have chosen to present them separately to maintain consistency in our figure layout, as seen in Figures 2F and 2G, as well as Figures S3F and S3G. This layout allows for a clearer visualization of the impact of NMD inhibition on *UNC13A* mRNA stability in MATR3-, FUS-, and hnRNPA1-KO cells, where no significant effect was observed. By keeping these figures separate, we aim to highlight the contrast between the regulatory mechanisms in TDP-43-KO cells and other RBP-KO cells, ensuring that the lack of *UNC13A* mRNA stabilization in the latter is clearly conveyed.

11. Figure 2H does not seem to have illustrate the conclusion well.

[Response] We thank the referee for the valuable feedback. We have revised Figure 2H to present the conclusion more visually and effectively.

new Figure 2H

12. Figure 7D cartoon is probably not needed as it is data from other people.

[Response] We have removed the Figure 7D cartoon as suggested.

13. The distinction between mRNA and nascent RNA in Figure 2 is unclear. The Y-axis annotations should be revised to clearly indicate the measured parameters.

[Response] To clarify the distinction between mRNA and nascent RNA, we have labeled the Y-axis as “Relative nascent mRNA amount” for panels F and G.

14. The heatmap lacks a score legend or annotation. Adding this information would make the data more accessible to readers.

[Response] We have labeled the color key as “Z-score” to indicate the normalization method used for visualization. In the legend, we have added the following statement to further clarify the normalization approach: “Heatmap color key represents Z-score normalized expression values.”

15. The manuscript mentions the noncanonical NRSE motif but does not describe its function. A brief explanation would provide valuable context for readers unfamiliar with this element.

[Response] We have added a brief explanation of the noncanonical NRSE motif, highlighting its ability to interact with REST and contribute to gene repression (Page 7, Lines 5-9).

16. The quantification in Figure 7H is not well-described. Providing details about the analysis methods and statistical validation would strengthen the conclusions drawn from this figure.

[Response] We have updated the Methods section and added a description of the quantification method for REST-positive motor neurons.

Dear Prof. Nakayama,

Thank you again for submitting your revised manuscript (EMBOJ-2024-119340R) to The EMBO Journal for our consideration, and for your patience during peer review. Your manuscript has been seen by the three original referees who had previously assessed the first version of your manuscript, and we have received their comments, which you can find below.

As you will see, the referees recognize that the revised version of the manuscript is improved and addresses the majority of the initially raised concerns. However, and in contrast to referee #2 who has no further comments, both referees #1 and #3 identify a number of remaining limitations that have not been sufficiently addressed in this revision. They point out a number of claims that are not fully supported by the available data and call for additional experimental work including quantification of certain experiments for these claims to be sufficiently supported. I would also like to draw your attention to the first point of referee #3 regarding the effectiveness of the used differentiation protocol, which we find essential to be fully addressed. The third point of referee #3 would also strengthen your work significantly and increase its impact on the field.

Although at The EMBO Journal we have a single major review round policy, in light of the supportive comments of the referees explaining that the work has been significantly improved and many of the initially raised concerns addressed, we would like to give you the opportunity to submit another revised version of your manuscript that should fully address all referees' points that have been made since the initial round of review. Any points that were only made in the second round would be optional, but still recommended for the improvement of your study and manuscript. The first point of referee #3 would be essential to be fully addressed, since it could bring a significant part of the study into question. Your revised manuscript will be sent back to referees #1 and #3 for re-review; acceptance of your manuscript will depend on the completeness of your responses in this final version.

Please include in your resubmission a point-by-point letter with detailed and complete responses to all comments, also describing and explaining any changes and additions to the manuscript. Please let me know if there are any points you would like to discuss further with me.

From the editorial side, there are also a number of changes and corrections we need you to make in the next version of your manuscript, before we can further proceed with its handling:

- All funding information provided in the Acknowledgements section of the manuscript should also be entered in our manuscript tracking system during your next submission. Currently, information regarding "Biomedical Research Core of Tohoku University Graduate School of Medicine and the Support System for Young Researchers to Use Research Equipment, Instruments, and Devices" is missing from the online system.
- The author contributions statement should be removed from the manuscript file. Instead, we use CRediT to specify the contributions of each author in the journal submission system. Please feel free to use the free text box to provide more detailed descriptions during submission. See also our guide to authors for more information: <https://www.embopress.org/page/journal/14602075/authorguide#authorshipguidelines>.
- All Figure panel callouts should be listed sequentially.
- There is no Figure S4C - this callout should be corrected.
- Please make sure to include all necessary information regarding the ethics approval of the experiments involving human samples in the Ethics section of your Author Checklist and also in the Methods of your revised manuscript.
- In the "Experimental study design and statistics" section of your Author Checklist, you have specified that information regarding statistical tests is provided in the "Correspondence" section of the manuscript; it is not clear where this information can be found, could you please revise this for better clarity?
- The source file name, title, legend, and manuscript callouts all need to be updated to "Dataset EV1" instead of "Table EV1"; this file should not be uploaded as a zip folder, but individually as a Dataset file with the legends in a separate tab/sheet in the same Excel file.
- The other two tables should be renamed to "Table EV1-EV2" with the corresponding callouts updated accordingly and their legends placed above the tables in each Excel file; these tables should also be uploaded as individual files (not as zip folders) as Expanded View Content.
- Please download and fill our Reagents and Tools Table template (.docx), which you can find in our author guide: <https://www.embopress.org/page/journal/14602075/authorguide#structuredmethods>. When submitting your revised manuscript, please do not include the Reagents and Tools Table in the Methods section of the revised manuscript but instead upload it as a separate file choosing the file type "Reagent Table".

- During our routine data checks, our data editors have raised the following queries regarding Figures, data, and legends. Please make sure that the following requests are fully addressed in the next version of your manuscript:
1. Please define the annotated p values ****/***/**/* as well as provide the exact p-values for the same in the legend of Figure 4H as appropriate.
 2. Please note that the exact p values are not provided in the legends of Figures 1D, F; 2A, C, D, G; 3E, 4C, D, F, G; 6C, D, E, G, H, I; 7A, B, C, I; EV2 B.
 3. Please note that the box plots need to be defined in terms of minima, maxima, percentile in the legend of Figure 7I.
 4. Please note that the box plots need to be defined in terms of minima, maxima, centre, bounds of box and whiskers, and percentile in the legends of Figures EV5 C, D.
 5. Please note that information related to "n" is missing in the legend of Figure 7I.

- Please remove the "Correspondence" statement from your revised manuscript; the contact information of the corresponding authors is already included on the title page of the manuscript.

- Please also make sure that the e-mail address of co-author Masashi Aoki is updated in their profile in our system to a valid one, as our previous notification sent to "okim@med.tohoku.ac.jp" could not be delivered.

Please also note that as part of the EMBO publications' Transparent Editorial Process, The EMBO Journal publishes online a Peer Review File along with each accepted manuscript. This File will be published in conjunction with your paper and will include the referee reports, your point-by-point response and all pertinent correspondence relating to the manuscript. You can opt out of this by letting the editorial office know (contact@embojournal.org). If you do opt out, the Peer Review File link will point to the following statement: "No Peer Review File is available with this article, as the authors have chosen not to make the review process public in this case."

We look forward to seeing a final version of your manuscript as soon as possible. Please let us know if you have any questions and use this link to submit your revision: <https://emboj.msubmit.net/cgi-bin/main.plex>.

Best regards,

Ioannis

Referee #1:

The authors addressed almost all of my comments. There are, however, a few remaining concerns and points that need to be addressed.

In the new Figure Appendix S7A, it does not appear that there is less FUS in the nucleus in the mutated lines. These results should be quantified if the authors wish to claim that this model can be associated with an RBP knockout (which I previously requested to be carried out in neurons). I still disagree that this model is comparable to an RBP knockout; however, in any case, the experiment must be quantified to support any claim.

In response to my comment regarding the relationship between FUS phase separation and REST expression, the authors performed an experiment in which they dissolved stress granules (SGs) at baseline, as I understand it. I do not see the point of this experiment, since there are not many stress granules at baseline. Instead, I suggest that the authors measure REST expression upon sodium arsenite treatment.

In the new Figure EV1B, the authors claim that only TDP-43 causes cell death due to the evident increase of cleaved PARP. I would argue that HNRNPA1 knockout also causes cell death. Indeed, the ratio between cleaved and uncleaved PARP is approximately 30:70, much higher than in the controls (~10:90). I suggest adding a quantification of this Western blot.

Referee #2:

The authors have addressed my concerns.

Referee #3:

1. The use of FUS P525L/+ iMNs and the observation of partial cytoplasmic mislocalization of FUS suggest a potential loss of nuclear function, which is an important and relevant finding. However, the current revision does not sufficiently address whether knockouts of other RNA-binding proteins (RBPs)-such as TDP-43, MATR3, and hnRNPA1-in SH-SY5Y cell lines lead to neurite degeneration. Based on our experience with SH-SY5Y differentiation, the morphology of the undifferentiated cells shown in this study appears suboptimal and does not adequately reflect neuronal features. Notably, even the control cells exhibit an undifferentiated morphology, raising concerns about the effectiveness of the differentiation protocol used. To convincingly support the proposed role of REST in rescuing cytoplasmic phenotypes, the author should either validate the phenotype using iPSC-derived motor neurons (iPSC-MNs) or optimize the SH-SY5Y differentiation conditions. Several published protocols (e.g., doi: 10.1007/s11626-024-00948-6), demonstrate that efficient and reproducible differentiation is achievable. Furthermore, immunostaining with a neurite marker such as β -III tubulin is recommended to assess neurite degeneration and strengthen the phenotypic conclusions.
2. The quantification presented in Figures 7E-7J requires further clarification. Please provide the number of postmortem samples analyzed (n) and clearly define the criteria used to identify REST-positive motor neurons to ensure reproducibility and interpretability.
3. The mechanistic role of REST remains insufficiently substantiated in the current revision. To convincingly support the proposed ALS-REST-IDR axis, additional experiments are necessary. In particular, it would be informative to examine whether 27YS-mediated rescue in the context of FUS mutation can modulate REST levels and thereby stabilize Unc13A expression.
4. The revised figure layout presents challenges for the reader in terms of clarity and navigation. It is strongly recommended that the author reorganize the figure panels, particularly by integrating the expanded view figures more effectively into the main figures, to improve visual coherence and readability.

Referee #1

In the new Figure Appendix S7A, it does not appear that there is less FUS in the nucleus in the mutated lines. These results should be quantified if the authors wish to claim that this model can be associated with an RBP knockout (which I previously requested to be carried out in neurons). I still disagree that this model is comparable to an RBP knockout; however, in any case, the experiment must be quantified to support any claim.

(Response) While we do not equate the FUS P525L/+ iPSC-derived motor neuron model with a complete RBP knockout, we consider it a more physiologically and disease-relevant system that recapitulates the nuclear loss of FUS function observed in FUS-ALS patients, likely due to cytoplasmic mislocalization of FUS, as previously reported (Marrone et al., 2019; Sun et al., 2015). As the referee correctly pointed out, our findings using RBP-KO SH-SY5Y cells provide limited insight into “what happens in patient neurons.” Therefore, we believe this model is better suited to investigate disease mechanisms in a disease-relevant context than using a KO model.

To address the referee’s request, we quantified FUS localization by calculating the cytoplasmic-to-nuclear intensity ratio (Appendix Figure S7A, right). We have also revised the manuscript (Page 10, Lines 28–36) to clarify the rationale for using this model.

If the referee’s concern is that factors beyond partial cytoplasmic mislocalization of FUS may contribute to REST dysregulation in FUS P525L/+ motor neurons, we fully agree this is a valid and important point. Indeed, although there are differences in cell types, compared to the ~2-fold increase in REST observed in our FUS-KO SH-SY5Y cells (Fig. 3E), REST levels in FUS P525L/+ iMNs show an approximately 4-fold increase relative to WT iMNs (Fig. 7A). This raises possibility that, in addition to the loss of nuclear FUS function, the aberrant LLPS properties of the FUS P525L protein may also be involved. We further discussed this point in the Discussion section (Page 15, Lines 8–20) and also address it in our response to Referee #3-3.

We are grateful to the referee for the thoughtful comments, which improved the clarity of our study.

Appendix Figure S7A (revised)
In response to my comment regarding the relationship between FUS phase separation and REST expression, the authors performed an experiment in which they dissolved stress granules (SGs) at baseline, as I understand it. I do not see the point of this experiment, since there are not many stress granules at baseline. Instead, I suggest that the authors measure REST expression upon sodium arsenite treatment.

(Response) We appreciate the referee's continued interest in the relationship between FUS phase separation and REST expression. The referee suggested measuring REST expression following sodium arsenite treatment, likely to test the involvement of SGs in this regulatory mechanism. In Fig. 6G, we assessed SG formation in FUS 27YS-expressing cells to demonstrate its impaired LLPS capacity. However, this experiment was intended for characterization purposes. The condensates potentially involved in REST mRNA regulation are not limited to SGs. FUS has been shown to participate in several LLPS-related compartments even under non-stress conditions, including nuclear condensates such as paraspeckles (An *et al.*, 2019; Reber *et al.*, 2021). Importantly, we observed increased REST mRNA levels in both FUS-KO and FUS 27YS-expressing cells under basal conditions, suggesting that REST regulation by FUS does not require stress-induced SGs. For this reason, in our previous response to the referee, we used 1,6-hexanediol to broadly disrupt LLPS under physiological (non-stress) conditions, rather than targeting SGs specifically. Therefore, we believe sodium arsenite treatment may not offer additional insight into the mechanism we are investigating.

We thank the referee for raising this point and acknowledge the limitation of our current analysis in not identifying the specific condensate responsible for REST mRNA regulation. We have clarified this point in the revised Discussion (Page 14, Lines 27–34).

In the new Figure EV1B, the authors claim that only TDP-43 causes cell death due to the evident increase of cleaved PARP. I would argue that HNRNPA1 knockout also causes cell death. Indeed, the ratio between cleaved and uncleaved PARP is approximately 30:70, much higher than in the controls (~10:90). I suggest adding a quantification of this Western blot.

(Response) We agree that the cleaved-to-full-length PARP ratio appears elevated in hnRNPA1-KO cells compared to controls. However, our assessment of apoptosis was based on cleaved PARP levels normalized to the loading control (β -actin), which we consider a more consistent measure of caspase activity across KO lines. In contrast, normalization to full-length PARP is sensitive to variation in total PARP levels, which were reduced in hnRNPA1-KO cells, potentially inflating the cleaved/full-length ratio without reflecting increased apoptosis. To clarify, we have now quantified cleaved PARP relative to β -actin and added this to the revised Figure EV1B. The normalized values were 1.0 in WT, 1.9 in TDP-43-KO, and 0.61 in hnRNPA1-KO cells.

Referee #3:

1. The use of FUS P525L/+ iMNs and the observation of partial cytoplasmic mislocalization of FUS suggest a potential loss of nuclear function, which is an important and relevant finding. However, the current revision does not sufficiently address whether knockouts of other RNA-binding proteins (RBPs)-such as TDP-43, MATR3, and hnRNPA1-in SH-SY5Y cell lines lead to neurite degeneration. Based on our experience with SH-SY5Y differentiation, the morphology of the undifferentiated cells shown in this study appears suboptimal and does not adequately reflect neuronal features. Notably, even the control cells exhibit an undifferentiated morphology, raising concerns about the effectiveness of the differentiation protocol used. To convincingly support the proposed role of REST in rescuing cytoplasmic phenotypes, the author should either validate the phenotype using iPSC-derived motor neurons (iPSC-MNs) or optimize the SH-SY5Y differentiation conditions. Several published protocols (e.g., doi: 10.1007/s11626-024-00948-6), demonstrate that efficient and reproducible differentiation is achievable. Furthermore, immunostaining with a neurite marker such as β -III tubulin is recommended to assess neurite degeneration and strengthen the phenotypic conclusions.

(Response) We appreciate the referee's thoughtful suggestion and agree that studying neuronal morphology in a differentiated setting could provide additional insights into REST-dependent rescue effects. We would like to explain the technical limitations we faced and clarify why we chose our current experimental approach.

All SH-SY5Y (WT and RBP-KO lines) cell morphologies shown in EV1A and EV2A were observed under undifferentiated conditions, without any differentiation stimuli. Similar to the referee's current concern, we initially considered using RA-BDNF to induce neuronal differentiation prior to performing the REST rescue, expecting that a more neuron-like state might yield more biologically relevant effects. Accordingly, we applied a well-established RA-BDNF differentiation protocol (as described in doi: 10.1177/1087057115625190) to our SH-SY5Y RBP-KO lines. While wild-type cells differentiated successfully under these conditions (see **Figure 1 for Referee #3**), the RBP-KO cells exhibited significant difficulty attaching to glass surfaces—even after iMatrix coating—making it technically unfeasible to perform immunofluorescence-based morphological analysis.

This poor adhesion seems to be an inherent problem in the knockout cells, rather than a technical issue. Supporting this, our RNA-seq data showed that genes involved in the "Cell adhesion molecules" KEGG pathway were significantly downregulated in all four RBP-KO lines (Fig. 1B), suggesting that RBP loss directly affects cell adhesion.

In addition, the study cited by Referee #3 used β III-tubulin and Synaptophysin as markers of differentiation. In our previous response to Referee #1, we tested these markers by

Western blot after RA-BDNF treatment (see **Figure 2 for Referee #3**). While wild-type cells showed strong expression of both markers, RBP-KO cells showed much weaker signals, indicating a failure to properly differentiate. Therefore, even if the adhesion problem could somehow be solved, these cells still would not acquire proper neuronal features and would not be a suitable model for assessing REST rescue in a differentiated state.

Given these issues, we chose to carry out REST knockdown experiments in undifferentiated SH-SY5Y cells. Although these cells are not fully differentiated, they still show gene expression patterns related to the neuronal lineage. Importantly, reducing REST levels clearly increased the number of cells with neurite-like cell processes (Fig. EV2A–B), giving useful insight into REST’s function.

While we agree that iPSC-derived neurons may offer a more physiologically relevant model in principle, applying this system to our current study is technically challenging. In particular, generating and characterizing iPSC-derived neurons from RBP-KO lines would require establishing new gene-edited iPSC models, which is outside the scope and feasibility of the present work. Furthermore, the severe differentiation defects we observed in SH-SY5Y RBP-KO cells raise the possibility that similar limitations may apply to RBP-deficient iPSC-derived neurons as well. Therefore, we believe that our current model provides a technically sound and biologically relevant system to address the referee’s original question regarding the RBP–REST relationship.

That said, we recognize this as a limitation of our study, and agree that additional validation will be necessary to determine whether REST upregulation has a pathogenic effect in neurons. A statement about this point has been added to the revised Discussion section, addressing both the significance and limitations of using undifferentiated SH-SY5Y cells for REST knockdown experiments in the current study (Page 13, Line 34 – Page 14, Line 7), as well as the need for future validation using iPSC-derived motor neurons and in vivo systems (Page 14, Lines 16–20).

Figure 1 for Referee #3

(Successful differentiation in wild-type cells under RA-BDNF treatment)

Figure 2 for Referee #3 (Figure 1 for Referee #1 in 1st revision)

(Expression level of differentiation marker, βIII-tubulin and Synaptophysin are suppressed in RBP-KO cells)

2. The quantification presented in Figures 7E-7J requires further clarification. Please provide the number of postmortem samples analyzed (n) and clearly define the criteria used to identify REST-positive motor neurons to ensure reproducibility and interpretability.

(Response) We thank the referee for this important comment regarding data clarity and reproducibility. As now stated in the figure legend, we analyzed 28 spinal cord sections: 12 from three individuals with sporadic ALS, 4 from one individual with familial ALS (FUS R521C/+), and 12 from three non-ALS disease controls. As described in the Methods, REST positivity was defined as nuclear DAB staining at least twice the background level. Quantification was performed using the Colour Deconvolution function in ImageJ. The original immunostaining images have been provided as source data to support transparency and reproducibility.

3. The mechanistic role of REST remains insufficiently substantiated in the current revision. To convincingly support the proposed ALS-REST-IDR axis, additional experiments are necessary. In particular, it would be informative to examine whether 27YS-mediated rescue in the context of FUS mutation can modulate REST levels and thereby stabilize Unc13A expression.

(Response) We thank the referee for raising this important point regarding the potential involvement of REST in the phenotypic rescue observed with 27YS treatment in the context of FUS mutation. We understand that the referee is suggesting an experiment in a disease-relevant model, such as iPSC-derived motor neurons carrying endogenous mutant FUS, to evaluate whether expression of FUS-WT or LLPS-deficient FUS (27YS) can differentially affect REST and UNC13A levels. We agree that such a comparison could

further support the role of FUS LLPS in modulating the REST–UNC13A axis.

While we have already employed FUS P525L/+ iPSC-derived neurons in this study, overexpression of 27YS in these cells poses substantial technical challenges. In our experience, iPSC-derived motor neurons exhibit markedly reduced transduction efficiency after differentiation, making it difficult to achieve stable and uniform expression using lentiviral or other delivery vectors. Furthermore, due to their post-mitotic and non-proliferative nature, standard selection-based approaches for generating stable expression lines are not feasible. As a result, it remains extremely difficult to reproducibly perform quantitative analyses under well-controlled expression conditions without compromising cell viability. Given these limitations, we consider that carrying out such experiments within the current revision period is not feasible.

We recognize that the referee may be asking whether the IDRs of ALS-associated RBPs, such as FUS, are directly involved in REST regulation in disease-relevant neuronal models. In our study, we observed *REST* mRNA dysregulation in iPSC-derived motor neurons carrying the FUS P525L mutation, which resides within the IDR and disrupts nuclear localization. Although we cannot currently provide experimental evidence to establish a causal link, we believe it is reasonable to consider that LLPS-related mechanisms, in addition to nuclear depletion of FUS, may contribute to this dysregulation.

To address this mechanistic possibility, we have revised the Discussion to more clearly articulate how our FUS P525L/+ iMN data support a potential role for LLPS-dependent regulation of REST. This point also relates to Referee #1-1's comment, and we kindly ask the referee to refer to our response there as well. We now explicitly state that future studies will be needed to dissect this mechanism in disease-relevant systems (Page 15, Lines 8–20).

4. The revised figure layout presents challenges for the reader in terms of clarity and navigation. It is strongly recommended that the author reorganize the figure panels, particularly by integrating the expanded view figures more effectively into the main figures, to improve visual coherence and readability.

(Response) We thank the referee for the helpful comment on the figure layout. While we agree that better integration of figures can improve readability, our main figures were designed to clearly present the central message of our study—how ALS-related RBPs regulate REST and UNC13A. The expanded view figures provide additional data that support our findings, but they are not directly part of the main mechanism we are focusing on. Therefore, we think it is clearer to keep them as expanded view figures instead of moving them into the main figures.

Dear Prof. Nakayama,

Thank you for submitting your revised manuscript EMBOJ-2024-119340R1 to The EMBO Journal for our consideration. It was sent back to the original referees #1 and #3 for re-review, and we have now received their comments, which you can find below.

As you will see, both referees find the manuscript sufficiently revised and are now supportive of publication in The EMBO Journal. In light of this input, I am very pleased to inform you that your manuscript has been accepted for publication in The EMBO Journal. Thank you very much for comprehensively addressing the initially raised referees' criticisms and concerns, and the editorial requests for changes and corrections.

If you have any questions, please do not hesitate to contact the Editorial Office. Thank you for your contribution to The EMBO Journal. Working with you has been a pleasure!

Yours sincerely,

Referee #1:

The author answered all my requests.

Referee #3:

The authors have made several attempts to address the concerns raised by both myself and the other reviewer, particularly surrounding two key issues: (1) whether the SH-SY5Y cell line is a suitable model to capture motor neuron degeneration phenotypes, and (2) whether the proposed stress granule/IDR-LLPS axis causally links REST and RBP dysregulation. The authors observed that SH-SY5Y RBP knockout (KO) cells showed impaired differentiation, likely due to reduced adhesion, as reflected by diminished β III-tubulin and synaptophysin expression following RA-BDNF induction. As a result, they chose to analyze undifferentiated SH-SY5Y RBP-KO cells to probe downstream neuronal functions. While this rationale is reasonable, the causative attribution of the observed phenotypes to RBP loss, and if this phenotype is mediated by REST/Unc13 remains insufficiently demonstrated. To better support a causal relationship, a complementary approach, such as transient RBP knockdown in wild-type SH-SY5Y cells using lentiviral shRNA would be informative. This would allow the authors to test whether acute RBP depletion recapitulates the differentiation impairment and neurite pathology observed in the KO cells, and rescued by REST KD further. At present, these additional experiments remain unaddressed. The authors have acknowledged these limitations and discussed future directions in the revised manuscript, which is appreciated. Although the study does not yet provide definitive causative evidence linking the RBP/LLPS/REST axis to neuronal degeneration, it nonetheless offers intriguing mechanistic insight, particularly the convergence of multiple RBPs on Unc13 regulation via distinct molecular routes. In summary, while the mechanistic clarity remains somewhat limited, I find the manuscript to offer novel and thought-provoking observations. Provided that the authors clearly frame these findings as exploratory and acknowledge the current limitations, I am supportive of publication. Further validation in more physiologically relevant systems will be essential to establish the broader relevance of this regulatory pathway in the ALS context.
